# Graph Scattering beyond Wavelet Shackles

**Christian Koke**
Technical University of Munich &
Ludwig Maximilian University Munich
`christian.koke@tum.de`

**Gitta Kutyniok**
Ludwig Maximilian University Munich &
University of Tromsø
`kutyniok@math.lmu.de`

## Abstract

This work develops a flexible and mathematically sound framework for the design and analysis of graph scattering networks with variable branching ratios and generic functional calculus filters. Spectrally-agnostic stability guarantees for node- and graph-level perturbations are derived; the vertex-set non-preserving case is treated by utilizing recently developed mathematical-physics based tools. Energy propagation through the network layers is investigated and related to truncation stability. New methods of graph-level feature aggregation are introduced and stability of the resulting composite scattering architectures is established. Finally, scattering transforms are extended to edge- and higher order tensorial input. Theoretical results are complemented by numerical investigations: Suitably chosen scattering networks conforming to the developed theory perform better than traditional graph-wavelet based scattering approaches in social network graph classification tasks and significantly outperform other graph-based learning approaches to regression of quantum-chemical energies on QM7.

## 1 Introduction

Euclidean wavelet scattering networks [22, 4] are deep convolutional architectures where output-features are generated in each layer. Employed filters are designed rather than learned and derive from a fixed (tight) wavelet frame, resulting in a tree structured network with constant branching ratio. Such networks provide state of the art methods in settings with limited data availability and serve as a mathematically tractable model of standard convolutional neural networks (CNNs). Rigorous investigations — establishing remarkable invariance- and stability properties of wavelet scattering networks — were initially carried out in [22]. The extensive mathematical analysis [38] generalized the term 'scattering network' to include tree structured networks with varying branching rations and frames of convolutional filters, thus significantly narrowing the conceptual gap to general CNNs.

With increasing interest in data on graph-structured domains, well performing networks generalizing Euclidean CNNs to this geometric setting emerged [18, 5, 9]. If efficiently implemented, such graph convolutional networks (GCNs) replace Euclidean convolutional filters by functional calculus filters; i.e. scalar functions applied to a suitably chosen graph-shift-oprator capturing the geometry of the underlying graph [18, 14, 9]. Almost immediately, proposals aimed at extending the success story of Euclidean scattering networks to the graph convolutional setting began appearing: In [48], the authors utilize dyadic graph wavelets (see e.g. [14]) based on the non-normalized graph Laplacian resulting in a norm preserving graph wavelet scattering transform. In [10], diffusion wavelets (see e.g. [8]) are used to construct a graph scattering transform enjoying spectrum-dependent stability guarantees to graph level perturbations. For scattering transforms with $N$ layers and $K$ distinct functional calculus filters, the work [11] derives node-level stability bounds of $\mathcal{O}(K^{N/2})$ and conducts corresponding numerical experiments choosing diffusion wavelets, monic cubic wavelets [14] and tight Hann wavelets [35] as filters. In [12] the authors, following [8], construct so called geometric wavelets and establish the expressivity of a scattering transform based on such a frame through extensive numerical

experiments. A theoretical analysis of this and a closely related wavelet based scattering transform is the main focus of [28]. Additionally, graph-wavelet based scattering transforms have been extended to the spatio-temporal domain [27], utilized to overcome the problem of oversmoothing in GCNs [25] and pruned to deal with their exponential (in network depth) increase in needed resources [15].

Common among all these contributions is the focus on graph wavelets, which are generically understood to derive in a scale-sampling procedure from a common wavelet generating kernel function $g : \mathbb{R} \to \mathbb{R}$ satisfying various properties [14]. Established stability or expressivity properties — especially to structural perturbations — are then generally linked to the specific choice of the wavelet kernel $g$ and utilized graph shift operator [10, 28]. This severely limits the diversity of available filter banks in the design of scattering networks and draws into question their validity as models for more general GCNs whose filters generically do not derive from a wavelet kernel.

A primary focus of this work is to provide alleviation in this situation: After reviewing the graph signal processing setting in Section 2, we introduce a general framework for the construction of (generalized) graph scattering transforms beyond the wavelet setting in Section 3. Section 4 establishes spectrum-agnostic stability guarantees on the node signal level and for the first time also for graph-level perturbations. To handle the vertex-set non-preserving case, a new 'distance measure' for operators capturing the geometry of varying graphs is utilized. After providing conditions for energy decay (with the layers) and relating it to truncation stability, we consider graph level feature aggregation and higher order inputs in Sections 5 and 6 respectively. In Section 7 we then provide numerical results indicating that general functional calculus filter based scattering is at least as expressive as standard wavelet based scattering in graph classification tasks and outperforms leading graph neural network approaches to regression of quantum chemical energies on QM7.

## 2 Graph Signal Processing

Taking a signal processing approach, we consider signals on graphs as opposed to graph embeddings:

**Node-Signals:** Given a graph $(G, E)$, we are primarily interested in node-signals, which are functions from the node-set $G$ to the complex numbers, modelled as elements of $\mathbb{C}^{|G|}$. We equip this space with an inner product according to $\langle f, g \rangle = \sum_{i=1}^{|G|} \overline{f_i} g_i \mu_i$ (with all vertex weights $\mu_i \geqslant 1$) and denote the resulting inner product space by $\ell^2(G)$. We forego considering arbitrary inner products on $\mathbb{C}^{|G|}$ solely in the interest of increased readability.

**Functional Calculus Filters:** Our fundamental objects in investigating node-signals will be functional calculus filters based on a normal operator $\Delta : \ell^2(G) \to \ell^2(G)$. Prominent examples include the adjacency matrix $W$, the degree matrix $D$, normalized $(\mathbb{1} - D^{-\frac{1}{2}} W D^{-\frac{1}{2}})$ or un-normalized $(\mathcal{L} := D - W)$ graph Laplacians Writing normalized eigenvalue-eigenvector pairs of $\Delta$ as $(\lambda_i, \phi_i)_{i=1}^{|G|}$, the filter obtained from applying $g : \mathbb{C} \to \mathbb{C}$ is given by $g(\Delta) f = \sum_{i=1}^{|G|} g(\lambda_i) \langle \phi_i, f \rangle_{\ell^2(V)} \phi_i$. The operator we utilize in our numerical investigations of Section 6, is given by $\mathscr{L} := \mathcal{L}/\lambda_{\max}(\mathcal{L})$. We divide by the largest eigenvalue to ensure that the spectrum $\sigma(\mathscr{L})$ is contained in the interval $[0, 1]$, which aids in the choice of functions from which filters are derived.

**Generalized Frames:** We are most interested in filters that arise from a collection of functions adequately covering the spectrum of the operator to which they are applied. To this end we call a collection $\{g_i(\cdot)\}_{i \in I}$ of functions a **generalized frame** if it satisfies the **generalized frame condition** $A \leqslant \sum_{i \in I} |g_i(c)|^2 \leqslant B$ for any $c$ in $\mathbb{C}$ for constants $A; B > 0$. As proved in Appendix B, this condition is sufficient to guarantee that the associated operators form a frame:

**Theorem 2.1.** Let $\Delta : \ell^2(G) \to \ell^2(G)$ be normal. If the family $\{g_i(\cdot)\}_{i \in I}$ of bounded functions satisfies $A \leqslant \sum_{i \in I} |g_i(c)|^2 \leqslant B$ for all $c$ in the spectrum $\sigma(\Delta)$, we have $(\forall f \in \ell^2(G))$

$$A \|f\|_{\ell^2(G)}^2 \leqslant \sum_{i \in I} \|g_i(\Delta) f\|_{\ell^2(G)}^2 \leqslant B \|f\|_{\ell^2(G)}^2.$$

Notably, the functions $\{g_i\}_{i \in I}$ need not be continuous: In fact, in our numerical implementations, we will – among other mappings – utilize the function $\delta_0(\cdot)$, defined by $\delta_0(0) = 1$ and $\delta_0(c) = 0$ for $c \neq 0$ as well as a modified cosine, defined by $\overline{\cos}(0) = 0$ and $\overline{\cos}(c) = \cos(c)$ for $c \neq 0$.

# 3 The Generalized Graph Scattering Transform

A generalized graph scattering transform is a non-linear map $\Phi$ based on a tree structured multilayer graph convolutional network with constant branching factor in each layer. For an input signal $f \in \ell^2(G)$, outputs are generated in each layer of such a scattering network, and then concatenated to form a feature vector in a feature space $\mathscr{F}$. The network is built up from three ingredients:

**Connecting Operators:** To allow intermediate signal representations in the 'hidden' network layers to be further processed with functional calculus filters based on varying operators, which might not all be normal for the same choice of node-weights, we allow these intermediate representations to live in varying graph signal spaces. In fact, we do not even assume that these signal spaces are based on a common vertex set. This is done to allow for modelling of recently proposed networks where input- and 'processing' graphs are decoupled (see e.g. [1, 36]), as well as architectures incorporating graph pooling [20]. Instead, we associate one signal space $\ell^2(G_n)$ to each layer $n$. Connecting operators are then (not necessarily linear) operators $P_n : \ell^2(G_{n-1}) \to \ell^2(G_n)$ connecting the signal spaces of subsequent layers. We assume them to be Lipschitz continuous $(\|P(f) - P(g)\|_{\ell^2(G_{n-1})} \leqslant R^+\|f - g\|_{\ell^2(G_n)})$ and triviality preserving $(P(0) = 0)$. For our original node-signal space we also write $\ell^2(G) \equiv \ell^2(G_0)$ .

**Non-Linearities:** To each layer, we also associate a (possibly) non-linear function $\rho_n : \mathbb{C} \to \mathbb{C}$ acting poinwise on signals in $\ell^2(G_n)$. Similar to connecting operators, we assume $\rho_n$ preserves zero and is Lipschitz-continuous with Lipschitz constant denoted by $L_n^+$. This definition allows for the absolute value non-linearity, but also ReLu or – trivially – the identity function.

**Operator Frames:** Beyond these ingredients, the central building block of our scattering architecture is comprised of a family of functional calculus filters in each layer. That is, we assume that in each layer, the node signal space $\ell^2(G_n)$ carries a normal operator $\Delta_n$ and an associated collection of functions comprised of an **output generating function** $\chi_n(\cdot)$ as well as a **filter bank** $\{g_{\gamma_n}(\cdot)\}_{\gamma_n \in \Gamma_n}$ indexed by an index set $\Gamma_n$. As the network layer $n$ varies (and in contrast to wavelet-scattering networks) we allow the index set $\Gamma_n$ as well as the collection $\{\chi_n(\cdot)\} \bigcup \{g_{\gamma_n}(\cdot)\}_{\gamma_n \in \Gamma_n}$ of functions to vary. We only demand that in each layer the functions in the filter bank together with the output generating function constitute a generalized frame with frame constants $A_n, B_n \geqslant 0$.

We refer to the collection of functions $\Omega_N := (\rho_n, \{\chi_n(\cdot)\} \bigcup \{g_{\gamma_n}(\cdot)\}_{\gamma_n \in \Gamma_n})_{n=1}^N$ as a **module sequence** and call $\mathscr{D}_N := (P_n, \Delta_n)_{n=1}^N$ our **operator collection**. The generalized scattering transform is then constructed iteratively:

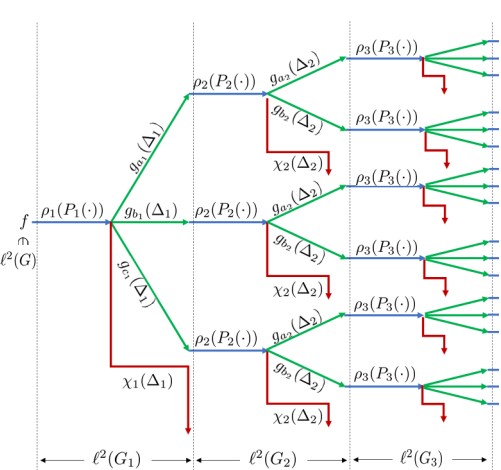

Figure 1: Schematic Scattering Architecture

To our initial signal $f \in \ell^2(G)$ we first apply the connecting operator $P_1$, yielding a signal representation in $\ell^2(G_1)$. Subsequently, we apply the pointwise non-linearity $\rho_1$. Then we apply our graph filters $\{\chi_1(\Delta_1)\} \bigcup \{g_{\gamma_1}(\Delta_1)\}_{\gamma_1 \in \Gamma_1}$ to $\rho_1(P_1(f))$ yielding the output $V_1(f) := \chi_1(\Delta_1)\rho_1(P_1(f))$ as well as the intermediate hidden representations $\{U_1[\gamma_1](f) := g_{\gamma_1}(\Delta_1)\rho_1(P_1(f))\}_{\gamma_1 \in \Gamma_1}$ obtained in the first layer. Here we have introduced the **one-step scattering propagator** $U_n[\gamma_n] : \ell^2(G_{n-1}) \to \ell^2(G_n)$ mapping $f \mapsto g_{\gamma_n}(\Delta_n)\rho_n(P_n(f))$ as well as the **output generating operator** $V_n : \ell^2(G_{n-1}) \to \ell^2(G_n)$ mapping $f$ to $\chi_n(\Delta_n)\rho_n(P_n(f))$. Upon defining the **set** $\Gamma^{N-1} := \Gamma_{N-1} \times ... \times \Gamma_1$ **of paths** of length $(N - 1)$ terminating in layer $N - 1$ (with $\Gamma^0$ taken to be the one-element set) and iterating the above procedure, we see that the outputs generated in the $N^{\text{th}}$-layer are indexed by paths $\Gamma^{N-1}$ terminating in the previous layer.

Outputs generated in the $N^{th}$ layer are thus given by $\{V_N \circ U[\gamma_{N-1}] \circ ... \circ U[\gamma_1](f)\}_{(\gamma_{N-1},...,\gamma_1) \in \Gamma^{N-1}}$. Concatenating the features obtained in the various layers of a network with depth $N$, our full feature vectors thus live in the feature space

$$\mathscr{F}_N = \oplus_{n=1}^N \left( \ell^2(G_n) \right)^{|\Gamma^{n-1}|}. \tag{1}$$

The associated canonical norm is denoted $\| \cdot \|_{\mathscr{F}_N}$. For convenience, a brief review of direct sums of spaces, their associated norms and a discussion of corresponding direct sums of maps is provided in Appendix A. We denote the hence constructed generalized scattering transform of length $N$, based on a module sequence $\Omega_N$ and operator collection $\mathscr{D}_N$ by $\Phi_N$.

In our numerical experiments in Section 7, we consider two particular instantiations of the above general architecture. In both cases the utilized shift-operator is $\mathscr{L} := \mathcal{L}/\lambda_{\max}(\mathcal{L})$, node weights satisfy $\mu_i = 1$, the branching ratio in each layer is chosen as 4 and the depth is set to $N = 4$ as well. The connecting operators are set to the identity and non-linearities are set to the modulus ($|\cdot|$). The two architectures differ in the utilized filters, which are repeated in each layer and depicted in Fig. 2. Postponing a discussion of other parameter-choices, we note here that the filters $\{\sin(\pi/2\cdot), \cos(\pi/2\cdot)\}$ provide a high and a low pass filter on the spectrum $\sigma(\mathscr{L}) \subseteq [0,1]$, while $\{\sin(\pi\cdot), \cos(\pi\cdot)\}$ provides a spectral refinement of the former two filters. The inner two elements of the filter bank in Architecture II thus separate an input signal into high- and low-lying spectral

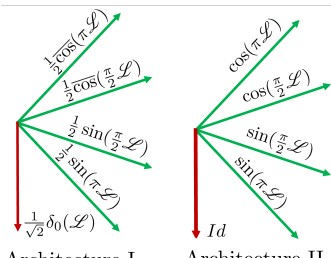

Figure 2: Filters of tested Architectures

components. The outer two act similarly at a higher spectral scale. Additionally Architecture I – utilizing $\overline{\cos}$ and $\delta_0$ as introduced Section 2 – prevents the lowest lying spectral information from propagating. Instead it is extracted via $\delta_0(\cdot)$ in each layer. Note that $Id$ arises from applying the constant-1 function to $\mathscr{L}$. Normalizations are chosen to generate frames with upper bounds $B \lessgtr 1$.

## 4  Stability Guarantees

In order to produce meaningful signal representations, a small change in input signal should produce a small change in the output of our generalized scattering transforms. This property is captured in the result below, which is proved in Appendix C.

**Theorem 4.1.** With the notation of Section 3, we have for all $f, h \in \ell^2(G)$:

$$\|\Phi_N(f) - \Phi_N(h)\|_{\mathscr{F}_N} \leqslant \left( 1 + \sum_{n=1}^N \max\{[B_n - 1], [B_n(L_n^+ R_n^+)^2 - 1], 0\} \prod_{k=1}^{n-1} B_k \right)^{\frac{1}{2}} \|f - h\|_{\ell^2(G)}$$

In the case where upper frame bounds $B_n$ and Lipschitz constants $L_n^+$ and $R_n^+$ are all smaller than or equal one, this statement reduces to the much nicer inequality:

$$\|\Phi_N(f) - \Phi_N(h)\|_{\mathscr{F}_N} \leqslant \|f - h\|_{\ell^2(G)}. \tag{2}$$

Below, we always assume $R_n^+, L_n^+ \leqslant 1$ as this easily achievable through rescaling. We will keep $B_n$ variable to demonstrate how filter size influences stability results. As for our experimentally tested architectures (cf. Fig. 2), we note for Architecture I that $B_n = 1/2$ for all $n$, so that (2) applies. For Architecture II we have $B_n = 3$, which yields a stability constant of $\sqrt{1 + 2 \cdot 3 + 2 \cdot 3^2 + 2 \cdot 3^3} = 9$. Similar to other constants derived in this section, this bound is however not necessarily tight.

Operators capturing graph geometries might only be known approximately in real world tasks; e.g. if edge weights are only known to a certain level of precision. Hence it is important that our scattering representation be insensitive to small perturbations in the underlying normal operators in each layer, which is captured by our next result, proved in Appendix D. Smallness here is measured in Frobenius norm $\| \cdot \|_F$, which for convenience is briefly reviewed in Appendix A).

**Theorem 4.2.** Let $\Phi_N$ and $\widetilde{\Phi}_N$ be two scattering transforms based on the same module sequence $\Omega_N$ and operator sequences $\mathscr{D}_N, \widetilde{\mathscr{D}}_N$ with the same connecting operators ($P_n = \widetilde{P}_n$) in each layer. Assume $R_n^+, L_n^+ \leqslant 1$ and $B_n \leqslant B$ for some $B$ and $n \leqslant N$. Assume that the respective normal operators satisfy $\|\Delta_n - \widetilde{\Delta}_n\|_F \leqslant \delta$ for some $\delta > 0$. Further assume that the functions

$\{g_{\gamma_n}\}_{\gamma_n \in \Gamma_n}$ and $\chi_n$ in each layer are Lipschitz continuous with associated Lipschitz constants satisfying $L_{\chi_n}^2 + \sum_{\gamma_n \in \Gamma_n} L_{g_{\gamma_n}}^2 \leqslant D^2$ for all $n \leqslant N$ and some $D > 0$. Then we have

$$\|\widetilde{\Phi}_N(f) - \Phi_N(f)\|_{\mathscr{F}_N} \leqslant \sqrt{2(2^N - 1)} \cdot \sqrt{(\max\{B, 1/2\})^{N-1}} \cdot D \cdot \delta \cdot \|f\|_{\ell^2(G)}$$

for all $f \in \ell^2(G)$. If $B \leqslant 1/2$, the stability constant improves to $\sqrt{2(1 - B^N)/(1 - B)} \cdot D \leqslant 2 \cdot D$.

The condition $B \leqslant \frac{1}{2}$ is e.g. satisfied by our Architecture I, but –strictly speaking– we may not apply Theorem 4.2, since not all utilized filters are Lipschitz continuous. Remark D.3 in Appendix D however shows, that the above stability result remains applicable for this architecture as long as we demand that $\Delta$ and $\widetilde{\Delta}$ are (potentially rescaled) graph Laplacians. For Architecture II we note that $D = \pi\sqrt{10}/2$ and thus the stability constant is given by $\sqrt{2(2^4 - 1)} \cdot \sqrt{3^3} \cdot \pi\sqrt{10}/2 = 45\pi$.

We are also interested in perturbations that change the vertex set of the graphs in our architecture. This is important for example in the context of social networks, when passing from nodes representing individuals to nodes representing (close knit) groups of individuals. To investigate this setting, we utilize tools originally developed within the mathematical physics community [29]:

**Definition 4.3.** Let $\mathcal{H}$ and $\widetilde{\mathcal{H}}$ be two finite dimensional Hilbert spaces. Let $\Delta$ and $\widetilde{\Delta}$ be normal operators on these spaces. Let $J : \mathcal{H} \to \widetilde{\mathcal{H}}$ and $\widetilde{J} : \widetilde{\mathcal{H}} \to \mathcal{H}$ be linear maps — called **identification operators**. We call the two spaces $\delta$-**quasi-unitarily-equivalent** (with $\delta \geqslant 0$) if for any $f \in \mathcal{H}$ and $u \in \widetilde{\mathcal{H}}$ we have

$$\|Jf\|_{\widetilde{\mathcal{H}}} \leqslant 2\|f\|_{\mathcal{H}}, \quad \|(J - \widetilde{J}^*)f\|_{\widetilde{\mathcal{H}}} \leqslant \delta\|f\|_{\mathcal{H}},$$

$$\|f - \widetilde{J}Jf\|_{\mathcal{H}} \leqslant \delta\sqrt{\|f\|_{\mathcal{H}}^2 + \langle f, |\Delta| \, f \rangle_{\mathcal{H}}}, \quad \|u - J\widetilde{J}u\|_{\widetilde{\mathcal{H}}} \leqslant \delta\sqrt{\|u\|_{\widetilde{\mathcal{H}}}^2 + \langle u, |\widetilde{\Delta}| \, u \rangle_{\widetilde{\mathcal{H}}}}.$$

If, for some $w \in \mathbb{C}$ the resolvent $R := (\Delta - \omega)^{-1}$ satisfies $\|(\widetilde{R}J - JR)f\|_{\widetilde{\mathcal{H}}} \leqslant \delta\|f\|_{\mathcal{H}}$ for all $f \in \mathcal{H}$, we say that $\Delta$ and $\widetilde{\Delta}$ are $\omega$-$\delta$-**close** with identification operator $J$.

Absolute value $|\Delta|$ and adjoint $\widetilde{J}^*$ of operators are briefly reviewed in Appendix A. While the above definition might seem fairly abstract at first, it is in fact a natural setting to investigate structural perturbations as Figure 3 exemplifies. In our current setting, the Hilbert spaces in Definition 4.3 are node-signal spaces $\mathcal{H} = \ell^2(G)$, $\widetilde{\mathcal{H}} = \ell^2(\widetilde{G})$ of different graphs. The notion of $\omega$-$\delta$-closeness is then useful, as it allows to compare filters defined on different graphs but obtained from applying the same function to the respective graph-operators:

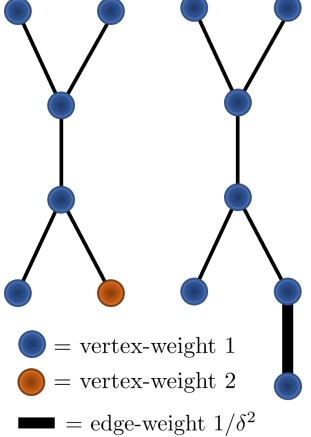

**Lemma 4.4.** In the setting of Definition 4.3 let $\Delta$ and $\widetilde{\Delta}$ be $\omega$-$\delta$-close and satisfy $\|\Delta\|_{op}, \|\widetilde{\Delta}\|_{op} \leqslant K$ for some $K > 0$. If $g : \mathbb{C} \to \mathbb{C}$ is holomorphic on the disk $B_{K+1}(0)$ of radius $(K + 1)$, there is a constant $C_g \geqslant 0$ so that

$$\|g(\widetilde{\Delta})J - Jg(\Delta)\|_{op} \leqslant C_g \cdot \delta$$

with $C_g$ depending on $g$, $\omega$ and $K$.

An explicit characterization of $C_g$ together with a proof of this result is presented in Appendix F. Lemma 4.4 is our main tool in establishing our next result, proved in Appendix G, which captures stability under vertex-set non-preserving perturbations:

= vertex-weight 1

= vertex-weight 2

= edge-weight $1/\delta^2$

Figure 3: Prototypical Example of $\delta$-unitary-equivalent Node Signal Spaces with $(-1)$-$12\delta$-close Laplacians. Details in Appendix E.

**Theorem 4.5.** Let $\Phi_N, \widetilde{\Phi}_N$ be scattering transforms based on a common module sequence $\Omega_N$ and differing operator sequences $\mathscr{D}_N, \widetilde{\mathscr{D}}_N$. Assume $R_n^+, L_n^+ \leqslant 1$ and $B_n \leqslant B$ for some $B$ and $n \geqslant 0$. Assume that there are identification operators $J_n : \ell^2(G_n) \to \ell^2(\widetilde{G}_n)$, $\widetilde{J}_n : \ell^2(\widetilde{G}_n) \to \ell^2(G_n)$ $(0 \leqslant n \leqslant N)$ so that the respective signal spaces are $\delta$-unitarily equivalent, the respective normal operators $\Delta_n, \widetilde{\Delta}_n$ are $\omega$-$\delta$-close as well as bounded (in norm) by $K > 0$ and the connecting operators satisfy $\|\widetilde{P}_n J_{n-1}f - J_n P_n f\|_{\ell^2(\widetilde{G}_n)} = 0$. For the common module sequence $\Omega_N$ assume that the non-linearities satisfy $\|\rho_n(J_n f) - J_n \rho_n(f)\|_{\ell^2(\widetilde{G}_n)} = 0$ and that the constants $C_{\chi_n}$ and

$\{C_{g_{\gamma_n}}\}_{\gamma_n \in \Gamma_N}$ associated through Lemma 4.4 to the functions of the generalized frames in each layer satisfy $C_{\chi_n}^2 + \sum_{\gamma_n \in \Gamma_N} C_{g_{\gamma_n}}^2 \leqslant D^2$ for some $D > 0$. Denote the operator that the family $\{J_n\}_n$ of identification operators induce on $\mathscr{F}_N$ through concatenation by $\mathscr{J}_N : \mathscr{F}_N \to \widetilde{\mathscr{F}}_N$. Then, with $K_N = \sqrt{(2^N - 1)2D^2 \cdot B^{N-1}}$ if $B > 1/2$ and $K_N = \sqrt{2D^2 \cdot (1 - B^N)/(1 - B)}$ if $B \leqslant 1/2$:

$$\|\widetilde{\Phi}_N(J_0 f) - \mathscr{J}_N \Phi_N(f)\|_{\widetilde{\mathscr{F}}_N} \leqslant K_N \cdot \delta \cdot \|f\|_{\ell^2(G}, \quad \forall f \in \ell^2(G).$$

The stability result persists with slightly altered stability constants, if identification operators only *almost* commute with non-linearities and/or connecting operators, as Appendix G further elucidates. Theorem 4.5 is not applicable to Architecture I, where filters are not all holomorphic, but is directly applicable to Architecture II. Stability constants can be calculated in terms of $D$ and $B$ as before.

Beyond these results, stability under truncation of the scattering transform is equally desirable: Given the energy $W_N := \sum_{(\gamma_N, \ldots, \gamma_1) \in \Gamma^N} \|U[\gamma_N] \circ \ldots \circ U[\gamma_1](f)\|_{\ell^2(G_N)}^2$ stored in the network at layer $N$, it is not hard to see that after extending $\Phi_N(f)$ by zero to match dimensions with $\Phi_{N+1}(f)$ we have $\|\Phi_N(f) - \Phi_{N+1}(f)\|_{\mathscr{F}_{N+1}}^2 \leqslant (R_{N+1}^+ L_{N+1}^+)^2 B_{N+1} \cdot W_N$ (see Appendix H for more details). A bound for $W_N$ is then given as follows:

**Theorem 4.6.** Let $\Phi_\infty$ be a generalized graph scattering transform based on a an operator sequence $\mathscr{D}_\infty = (P_n, \Delta_n)_{n=1}^\infty$ and a module sequence $\Omega_\infty$ with each $\rho_n(\cdot) \geqslant 0$. Assume in each layer $n \geqslant 1$ that there is an eigenvector $\psi_n$ of $\Delta_n$ with solely positive entries; denote the smallest entry by $m_n := \min_{i \in G_n} \psi_n[i]$ and the eigenvalue corresponding to $\psi_n$ by $\lambda_n$. Quantify the 'spectral-gap' opened up at this eigenvalue through neglecting the output-generating function by $\eta_n := \sum_{\gamma_n \in \Gamma_n} |g_{\gamma_n}(\lambda_n)|^2$ and assume $B_n m_n \geqslant \eta_n$. We then have (with $C_N^+ := \prod_{i=1}^N \max\{1, B_i(L_i^+ R_i^+)^2\}$)

$$W_N(f) \leqslant C_N^+ \cdot \left[ \prod_{n=1}^N \left(1 - \left(m_n - \frac{\eta_n}{B_n}\right)\right) \right] \cdot \|f\|_{\ell^2(G)}^2. \tag{3}$$

The product in (3) decays if $C_N^+ \to C^+$ converges and $\sum_{n=1}^N (m_n - \eta_n/B_n) \to \infty$ diverges as $N \to \infty$. The positivity-assumptions on the eigenvectors $\psi_n$ can e.g. always be ensured if they are chosen to lie in the lowest lying eigenspace of a graph Laplacian or normalized graph Laplacian (irrespective of the connectedness of the underlying graphs). As an example, we note that if we extend our Architecture I to infinite depth (recall from Section 3 that we are using the same filters, operators, etc. in each layer) we have upon choosing $\lambda_n = 0$ and $\psi_n$ to be the constant normalized vector that $\eta_n = 0$, $C_N = 1$ and $m_n = 1/\sqrt{|G|}$, for a graph with $|G|$ vertices. On a graph with 16 vertices, we then e.g. have $W_N \leqslant (3/4)^N \|f\|_{\ell^2(G)}^2$ and thus $\|\Phi_N(f) - \Phi_{N+1}(f)\|_{\mathscr{F}_{N+1}} \leqslant (3/4)^N \cdot \|f\|_{\ell^2(G)}^2/2$. As detailed in Appendix H, Theorem 4.6 also implies that under the given assumptions the scattering transform has trivial 'kernel' for $N \to \infty$, mapping only 0 to 0.

## 5 Graph-Level Feature Aggregation

To solve tasks such as graph classification or regression over multiple graphs, we need to represent graphs of varying sizes in a common feature space. Given a scattering transform $\Phi_N$, we thus need to find a stability preserving map from the feature space $\mathscr{F}_N$ to some Euclidean space that is independent of any vertex set cardinalities. Since $\mathscr{F}_N$ is a large direct sum of smaller spaces (cf. (1)), we simply construct such maps on each summand independently and then concatenate them.

**General non-linear feature aggregation:** Our main tool in passing to graph-level features is a non-linear map $N_p^G : \ell^2(G) \to \mathbb{R}^p$ given as

$$N_p^G(f) = \frac{1}{\sqrt{p}} (\|f\|_{\ell^1(G)}/\sqrt{\mu_G}, \|f\|_{\ell^2(G)}, \|f\|_{\ell^3(G)}, \ldots, \|f\|_{\ell^p(G)})^\top, \tag{4}$$

with $\mu_G := \sum_{i \in G} \mu_i$ and $\|f\|_{\ell^q(G)} := (\sum_{i \in G} |f_i|^q \mu_i)^{1/q}$. Our inspiration to use this map stems from the standard case where all $\mu_i = 1$: For $p \geqslant |G|$, the vector $|f| = ((|f_1|, \ldots, |f_G|)^\top$ can then be recovered from $N_p^G(f)$ up to permutation of indices [23]. Hence, employing $N_p^G$ (with $p \geqslant |G|$) to aggregate node-information into graph-level information, we lose the minimal necessary information

about node permutation (clearly $N_p^G(f) = N_p^G(\Pi f)$ for any permutation matrix $\Pi$) and beyond that only information about the complex phase (respectively the sign in the real case) in each entry of $f$.

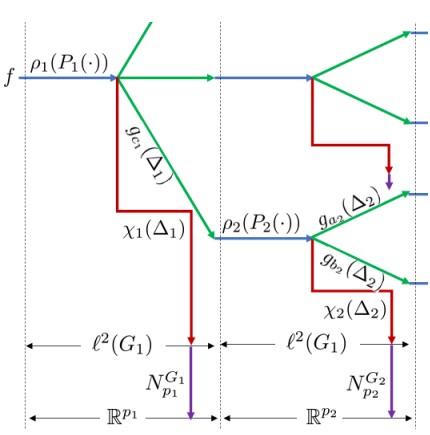

Figure 4: Graph Level Scattering

Given a scattering transform $\Phi_N$ mapping from $\ell^2(G)$ to the feature space $\mathscr{F}_N = \oplus_{n=1}^N \left( \ell^2(G_n) \right)^{|\Gamma^{n-1}|}$, we obtain a corresponding map $\Psi_N$ mapping from $\ell^2(G)$ to $\mathscr{R}_N = \oplus_{n=1}^N \left( \mathbb{R}^{p_n} \right)^{|\Gamma^{n-1}|}$ by concatenating the feature map $\Phi_N$ with the operator that the family of non-linear maps $\{N_{G_n}^{p_n}\}_{n=1}^N$ induces on $\mathscr{F}_N$ by concatenation. Similarly we obtain the map $\widetilde{\Psi}_N : \ell^2(\widetilde{G}) \to \mathscr{R}_N$ by concatenating the map $\widetilde{\Phi}_N : \ell^2(\widetilde{G}) \to \oplus_{n=1}^N \left( \ell^2(\widetilde{G}_n) \right)^{|\Gamma^{n-1}|}$ with the operator induced by the family $\{N_{\widetilde{G}_n}^{p_n}\}_{n=1}^N$. The feature space $\mathscr{R}_N$ is completely determined by path-sets $\Gamma^N$ and used maximal $p$-norm indices $p_n$. It no longer depends on cardinalities of vertex sets of any graphs, allowing to compare (signals on) varying graphs with each other. Most of the results of the previous sections then readily transfer to the graph-level-feature setting (c.f. Appendix I.1).

**Low-pass feature aggregation:** The spectrum-free aggregation scheme of the previous paragraph is especially adapted to settings where there are no high-level spectral properties remaining constant under graph perturbations. However, many commonly utilized operators, such as normalized and un-normalized graph Laplacians, have a somewhat 'stable' spectral theory: Eigenvalues are always real, non-negative, the lowest-lying eigenvalue equals zero and simple (if the graph is connected). In this section we shall thus assume that each mentioned normal operator $\Delta_n$ $(\widetilde{\Delta}_n)$ has these spectral properties. We denote the lowest lying normalized eigenvector (which is generically determined up to a complex phase) by $\psi_{\Delta_n}$ and denote by $M_{G_n}^{|\langle\cdot,\cdot\rangle|} : \ell^2(G_n) \to \mathbb{C}$ the map given by $M_{G_n}^{|\langle\cdot,\cdot\rangle|}(f) = |\langle\psi_{\Delta_n}, f\rangle_{\ell^2(G_n)}|$. The absolute value around the inner product is introduced to absorb the phase-ambiguity in the choice of $\psi_{\Delta_n}$. Given a scattering transform $\Phi_N$ mapping from $\ell^2(G)$ to the feature space $\mathscr{F}_N$, we obtain a corresponding map $\Psi_N^{|\langle\cdot,\cdot\rangle|}$ mapping from $\ell^2(G)$ to $\mathscr{C}_N = \oplus_{n=1}^N \mathbb{C}^{|\Gamma^{n-1}|}$ by concatenating the feature map $\Phi_N$ with the operator that the family of maps $\{M_{G_n}^{|\langle\cdot,\cdot\rangle|}\}_{n=1}^N$ induces on $\mathscr{F}_N$ by concatenation. As detailed in Appendix I.2, this map inherits stability properties in complete analogy to the discussion of Section 4.

## 6 Higher Order Scattering

Node signals capture information about nodes in isolation. However, one might be interested in binary, ternary or even higher order relations between nodes such as distances or angles in graphs representing molecules. In this section we focus on binary relations – i.e. edge level input – as this is the instantiation we also test in our regression experiment in Section 7. Appendix J.2 provides more details and extends these considerations beyond the binary setting. We equip the space of edge inputs with an inner product according to $\langle f, g \rangle = \sum_{i,j=1}^{|G|} \overline{f_{ij}} g_{ij} \mu_{ij}$ and denote the resulting inner-product space by $\ell^2(E)$ with $E = G \times G$ the set of edges. Setting e.g. node-weights $\mu_i$ and edge weights $\mu_{ik}$ to one, the adjacency matrix $W$ as well as normalized or un-normalized graph Laplacians constitute self-adjoint operators on $\ell^2(E)$, where they act by matrix multiplication. Replacing the $G_n$ of Section 3 by $E_n$, we can then follow the recipe laid out there in constructing $2^{nd}$-order scattering transforms; all that we need are a module sequence $\Omega_N$ and an operator sequence $\mathscr{D}_N^2 := (P_n^2, \Delta_n^2)_{n=1}^N$, where now $P_n^2 : \ell^2(E_{n-1}) \to \ell^2(E_n)$ and $\Delta_n^2 : \ell^2(E_n) \to \ell^2(E_n)$. We denote the resulting feature map by $\Phi_N^2$ and write $\mathscr{F}_N^2$ for the corresponding feature space. The map $N_p^G$ introduced in (4) can also be adapted to aggregate higher-order features into graph level features: With $\|f\|_q := (\sum_{ij \in G} |f_{ij}|^q \mu_{ij})^{1/q}$ and $\mu_E := \sum_{ij=1}^{|G|} \mu_{ij}$, we define $N_p^E(f) = (\|f\|_{\ell^1(E)}/\sqrt{\mu_E}, \|f\|_{\ell^2(E)}, \|f\|_{\ell^3(E)}, ..., \|f\|_{\ell^p(E)})^\top/\sqrt{p}$. Given a feature map $\Phi_N^2$ with feature space $\mathscr{F}_N^2 = \oplus_{n=1}^N \left( \ell^2(E_n) \right)^{|\Gamma^{n-1}|}$, we obtain a corresponding

map $\Psi_N^2$ mapping from $\ell^2(E)$ to $\mathscr{R}_N = \oplus_{n=1}^N (\mathbb{R}^{p_n})^{|\Gamma^{n-1}|}$ by concatenating $\Phi_N^E$ with the map that the family of non-linear maps $\{N_{p_n}^{E_n}\}_{n=1}^N$ induces on $\mathscr{F}_N$ by concatenation. The stability results of the preceding sections then readily translate to $\Phi_N^2$ and $\Psi_N^2$ (c.f. Appendix J).

## 7 Experimental Results

We showcase that even upon selecting the fairly simple Architectures I and II introduced in Section 3 (c.f. also Fig. 2), our generalized graph scattering networks are able to outperform both wavelet-based scattering transforms and leading graph-networks under different circumstances. To aid visual clarity when comparing results, we colour-code the best-performing method in green, the second-best performing in yellow and the third-best performing method in orange respectively.

**Social Network Graph Classification:** To facilitate contact between our generalized graph scattering networks, and the wider literature, we combine a network conforming to our general theory namely Architecture I in Fig. 2 (as discussed in Section 3 with depth $N = 4$, identity as connecting operators and $|\cdot|$-non-linearities) with the low pass aggregation scheme of Section 5 and a Euclidean support vector machine with RBF-kernel (GGSN+EK). The choice $N = 4$ was made to keep computation-time palatable, while aggregation scheme and non-linearities were chosen to facilitate comparison with standard *wavelet*-scattering approaches. For this hybrid architecture (GGSN+EK), classification accuracies under the standard choice of 10-fold cross validation on five common social network graph datasets are compared with performances of popular graph kernel approaches, leading deep learning methods as well as geometric wavelet scattering (GS-SVM) [12]. More details are provided in Appendix K. As evident from Table 1, our network consistently achieves higher accuracies than the geometric wavelet scattering transform of [12], with the performance gap becoming significant on the more complex REDDIT datasets, reaching a relative mean performance increase of more than $10\%$ on REDDIT-12K. This indicates the liberating power of transcending the graph wavelet setting. While on comparatively smaller and somewhat simpler datasets there is a performance gap between our static architecture and fully trainable networks, this gap closes on more complex datasets: While P-Poinc e.g. outperforms our method on IMDB datasets, the roles are reversed on REDDIT datasets. On REDDIT-B our approach trails only GIN; with difference in accuracies insignificant. On REDDIT-5K our method comes in third, with the gap to the second best method (GIN) being statistically insignificant. On REDDIT-12K we generate state of the art results.

Table 1: Classification Accuracies on Social Network Datasets

| Method | Classification Accuracies [%] | | | | | |
|---|---|---|---|---|---|---|
| | COLLAB | IMDB-B | IMDB-M | REDDIT-B | REDDIT-5K | REDDIT-12K |
| WL [33] | 77.82±1.45 | 71.60±5.16 | N/A | 78.52±2.01 | 50.77 ± 2.02 | 34.57 ± 1.32 |
| Graphlet [34] | 73.42±2.43 | 65.40±5.95 | N/A | 77.26±2.34 | 39.75 ± 1.36 | 25.98 ± 1.29 |
| DGK [42] | 73.00±0.20 | 66.90±0.50 | 44.50±0.50 | 78.00±0.30 | 41.20 ± 0.10 | 32.20 ± 0.10 |
| DGCNN [46] | 73.76±0.49 | 70.03±0.86 | 47.83±0.85 | N/A | 48.70 ± 4.54 | N/A |
| PSCN [26] | 72.60±2.15 | 71.00±2.29 | 45.23±2.84 | 86.30±1.58 | 49.10 ± 0.70 | 41.32 ± 0.42 |
| P-Poinc [19] | N/A | 81.86±4.26 | 57.31±4.27 | 79.78±3.21 | 51.71 ± 3.01 | 42.16 ± 3.41 |
| S2S-N2N-PP [16] | 81.75±0.80 | 73.80±0.70 | 51.19±0.50 | 86.50±0.80 | 52.28 ± 0.50 | 42.47 ± 0.10 |
| GSN-e [3] | 85.5 ± 1.2 | 77.8 ± 3.3 | 54.3 ± 3.3 | N/A | N/A | N/A |
| WKPI-kC[47] | N/A | 75.1 ± 1.1 | 49.5 ± 0.4 | N/A | 59.5 ± 0.6 | 48.4 ± 0.5 |
| GIN [41] | 80.20±1.90 | 75.10±5.10 | 52.30±2.80 | 92.40±2.50 | 57.50 ± 1.50 | N/A |
| GS-SVM [12] | 79.94±1.61 | 71.20±3.25 | 48.73±2.32 | 89.65±1.94 | 53.33 ± 1.37 | 45.23 ± 1.25 |
| GGSN+EK [OURS] | 80.34±1.68 | 73.20±3.76 | 49.47±2.27 | 91.60±1.97 | 56.89 ± 2.24 | 49.03 ± 1.58 |

**Regression of Quantum Chemical Energies:** In order to showcase the prowess of both our higher order scattering scheme and our spectrum-agnostic aggregation method of Section 5, we combine these building blocks into a hybrid architecture which we then apply in combination with kernel methods (2GGST + EK) to the task of atomization energy regression on QM7. This is a comparatively small dataset of 7165 molecular graphs, taken from the 970 million strong molecular database GDB-13 [2]. Each graph in QM7 represents an organic molecule, with nodes corresponding to individual atoms. Beyond the node-level information of atomic charge, there is also edge level information characterising interaction strengths between individual nodes/atoms available. This is encoded into so called Coulomb matrices (see e.g. [31] or Appendix K) of molecular graphs, which for us serve a dual purpose: On the one hand we consider a Coulomb matrix as an edge-level input signal on a given graph.

On the other hand, we also treat it as an adjacency matrix from which we build up a graph Laplacian $\mathcal{L}$. Our normal operator is then chosen as $\mathscr{L} = \mathcal{L}/\lambda_{max}(\mathcal{L})$ again. Connecting operators are set to the identity, while non-linearities are fixed to $\rho_{n \geqslant 1}(\cdot) = |\cdot|$. Filters are chosen as $(\sin(\pi/2 \cdot \mathscr{L}), \cos(\pi/2 \cdot \mathscr{L}), \sin(\pi \cdot \mathscr{L}), \cos(\pi \cdot \mathscr{L}))$ acting through matrix multiplication. Output generating functions are set to the identity and depth is $N = 4$, so that we essentially recover Architecture II of

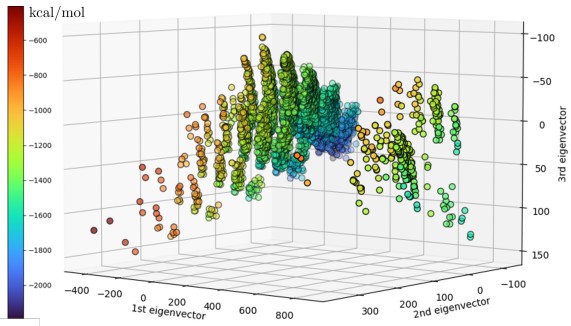

Fig. 2; now applied to edge-level input. Graph level features are aggregated via the map $N_5^E$ of Section 6. We chose $p = 5$ (and not $p \gg 5$) for $N_p^E$ to avoid overfitting. Generated feature vectors are combined with node level scattering features obtained from applying Architecture II of Fig. 2 to the input signal of atomic charge into composite feature vectors; plotted in Figure 5. As is visually evident, even when reduced to the low-dimensional subspace of their first three principal components, the generated scattering features are able to aptly resolve the atomization energy of the molecules. This aptitude is also reflected in Table 2, comparing our

Figure 5: Atomization Energy as a Function of primary Principal Components of Scattering Features

approach with leading graph-based learning methods trained with ten-fold cross validation on node

and (depending on the model) edge level information. Our method is the best performing. We significantly outperform the next best model (DTNN), producing less than half of its mean absolute error (MAE). Errors of other methods are at least one — sometimes two — orders of magnitude greater. In part, this performance discrepancy might be explained by the hightened suitability of our scattering transform for environments with somewhat limited training-data availability. Here we speculate that the additional performance gap might be explained by the fact that our graph shift operator $\Delta$ carries the same information as the Coulomb matrix (a proven molecular graph descriptor in itself [31]). Additionally, our filters being infinite series' in powers of the underlying normal operator allows for rapid dispersion of information across underlying molecular graphs, as opposed to e.g. the filters in GraphConv

Table 2: Comparison of Methods

| Method | MAE [kcal/mol] |
|---|---|
| AttentiveFP [40] | $66.2 \pm 2.8$ |
| DMPNN [44] | $105.8 \pm 13.2$ |
| DTNN [39] | $8.2 \pm 3.9$ |
| GraphConv [18] | $118.9 \pm 20.2$ |
| GROVER (base)[30] | $72.5 \pm 5.9$ |
| MPNN [13] | $113.0 \pm 17.2$ |
| N-GRAM[21] | $125.6 \pm 1.5$ |
| PAGTN (global) [6] | $47.8 \pm 3.0$ |
| PhysChem [45] | $59.6 \pm 2.3$ |
| SchNet [32] | $74.2 \pm 6.0$ |
| Weave [17] | $59.6 \pm 2.3$ |
| GGST+EK [OURS] | $11.3 \pm 0.6$ |
| 2GGST+EK [OURS] | $3.4 \pm 0.3$ |

or SchNet, which do not incorporate such higher powers. To quantify the effect of including second order scattering coefficients, we also include the result of performing kernel-regression solely on first order features generated through Architecture II of Fig. 2 (GGST + EK). While results are still better than those of all but one leading approach, incorporating higher order scattering improves performance significantly.

## 8    Discussion

Leaving behind the traditional reliance on graph wavelets, we developed a theoretically well founded framework for the design and analysis of (generalized) graph scattering networks; allowing for varying branching rations, non-linearities and filter banks. We provided spectrum independent stability guarantees, covering changes in input signals and for the first time also arbitrary normal perturbations in the underlying graph-shift-operators. After introducing a new framework to quantify vertex-set non-preserving changes in graph domains, we obtained spectrum-independent stability guarantees for this setting too. We provided conditions for energy decay and discussed implications for truncation stability. Then we introduced a new method of graph-level feature aggregation and extended scattering networks to higher order input data. Our numerical experiments showed that a simple scattering transform conforming to our framework is able to outperform the traditional graph-wavelet based approach to graph scattering in social network graph classification tasks. On complex datasets our method is also competitive with current fully trainable methods, ouperforming all competitors on REDDIT-12K. Additionally, higher order graph scattering transforms significantly outperform current leading graph-based learning methods in predicting atomization energies on QM7. A reasonable critique of scattering networks as tractable models for general graph convolutional

networks is their inability to emulate non-tree-structured network topologies. While transcending the wavelet setting has arguably diminished the conceptual gap between the two architectures, this structural difference persists. Additionally we note that despite a provided promising example, it is not yet clear whether the newly introduced graph-perturbation framework can aptly provide stability guarantees to all reasonable coarse-graining procedures. Exploring this question is the subject of ongoing work.

## Broader Impact

We caution against an over-interpretation of established mathematical guarantees: Such guarantees do not negate biases that may be inherent to utilized datasets.

## Disclosure of Funding

Christian Koke acknowledges support from the German Research Foundation through the MIMO II-project (DFG SPP 1798, KU 1446/21-2). Gitta Kutyniok acknowledges support from the ONE Munich Strategy Forum (LMU Munich, TU Munich, and the Bavarian Ministery for Science and Art), the Konrad Zuse School of Excellence in Reliable AI (DAAD), the Munich Center for Machine Learning (BMBF) as well as the German Research Foundation under Grants DFG-SPP-2298, KU 1446/31-1 and KU 1446/32-1 and under Grant DFG-SFB/TR 109, Project C09 and the Federal Ministry of Education and Research under Grant MaGriDo.

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
