# A  Some Concepts in Linear Algebra

In the interest of self-containedness, we provide a brief review of some concepts from linear algebra utilized in this work that might potentially be considered more advanced. Presented results are all standard; a very thorough reference is [24].

**Hilbert Spaces:**  To us, a Hilbert space — often denoted by $\mathcal{H}$ — is a vector space over the complex numbers which also has an inner product — often denoted by $\langle \cdot, \cdot \rangle_{\mathcal{H}}$. Prototypical examples are given by the Euclidean spaces $\mathbb{C}^d$ with inner product $\langle x, y \rangle_{\mathbb{C}^d} := \sum_{i=1}^d \overline{x}_i y_i$. Associated to an inner product is a norm, denoted by $\| \cdot \|_{\mathcal{H}}$ and defined by $\|x\|_{\mathcal{H}} := \sqrt{\langle x, x \rangle_{\mathcal{H}}}$ for $x \in \mathcal{H}$.

**Direct Sums of Spaces:**  Given two potentially different Hilbert spaces $\mathcal{H}$ and $\widehat{\mathcal{H}}$, one can form their direct sum $\mathcal{H} \oplus \widehat{\mathcal{H}}$. Elements of $\mathcal{H} \oplus \widehat{\mathcal{H}}$ are vectors of the form $(a, b)$, with $a \in \mathcal{H}$ and $b \in \widehat{\mathcal{H}}$. Addition and scalar multiplication are defined in the obvious way by

$$(a, b) + \lambda(c, d) := (a + \lambda c, b + \lambda d)$$

for $a, c \in \mathcal{H}$, $b, d \in \widehat{\mathcal{H}}$ and $\lambda \in \mathbb{C}$. The inner product on the direct sum is defined by

$$\langle (a, b), (c, d) \rangle_{\mathcal{H} \oplus \widehat{\mathcal{H}}} := \langle a, c \rangle_{\mathcal{H}} + \langle b, d \rangle_{\widehat{\mathcal{H}}}.$$

As is readily checked, this implies that the norm $\| \cdot \|_{\mathcal{H} \oplus \widehat{\mathcal{H}}}$ on the direct sum is given by

$$\|(a, b)\|_{\mathcal{H} \oplus \widehat{\mathcal{H}}}^2 := \|a\|_{\mathcal{H}}^2 + \|b\|_{\widehat{\mathcal{H}}}^2.$$

Standard examples of direct sums are again the Euclidean spaces, where one has $\mathbb{C}^d = \mathbb{C}^n \oplus \mathbb{C}^m$ if $m + n = d$, as is easily checked. One might also consider direct sums with more than two summands, writing $\mathbb{C}^d = \oplus_{i=1}^d \mathbb{C}$ for example. In fact, one might also consider infinite sums of Hilbert spaces: The space $\oplus_{i=1}^\infty \mathcal{H}_i$ is made up of those elements $a = (a_1, a_2, a_3, ...)$ with $a_i \in \mathcal{H}_i$ for which the norm

$$\|a\|_{\oplus_{i=1}^\infty \mathcal{H}_i}^2 := \sum_{i=1}^\infty \|a_i\|_{\mathcal{H}_i}^2$$

is finite. This means for example that the vector $(1, 0, 0, 0, ...)$ is in $\oplus_{i=1}^\infty \mathbb{C}$, while $(1, 1, 1, 1, ...)$ is not.

**Direct Sums of Maps:**  Suppose we have two collections of Hilbert spaces $\{\mathcal{H}_i\}_{i=1}^{\Gamma}$, $\{\widetilde{\mathcal{H}}_i\}_{i=1}^{\Gamma}$ with $\Gamma \in \mathbb{N}$ or $\Gamma = \infty$. Suppose further that for each $i \leqslant \Gamma$ (resp. $i < \Gamma$) we have a (not necessarily linear) map $J_i : \mathcal{H}_i \to \widetilde{\mathcal{H}}_i$. Then the collection $\{J_i\}_{i=1}^{\Gamma}$ of these 'component' maps induce a 'composite' map

$$\mathscr{J} : \oplus_{i=1}^{\Gamma} \mathcal{H}_i \longrightarrow \oplus_{i=1}^{\Gamma} \widetilde{\mathcal{H}}_i$$

between the direct sums. Its value on an element $a = (a_1, a_2, a_3, ...) \in \oplus_{i=1}^{\Gamma} \mathcal{H}_i$ is defined by

$$\mathscr{J}(a) = (J_1(a_1), J_2(a_2), J_3(a_3), ...) \in \oplus_{i=1}^{\Gamma} \widetilde{\mathcal{H}}_i.$$

Strictly speaking, one has to be a bit more careful in the case where $\Gamma = \infty$ to ensure that $\|\mathscr{J}(a)\|_{\oplus_{i=1}^\infty \widetilde{\mathcal{H}}_i} \neq \infty$. This can however be ensured if we have $\|J_i(a_i)\|_{\widetilde{\mathcal{H}}_i} \leqslant C \|a_i\|_{\mathcal{H}_i}$ for all $1 \leqslant i$ and some $C$ independent of all $i$, since then $\|\mathscr{J}(a)\|_{\oplus_{i=1}^\infty \widetilde{\mathcal{H}}_i} \leqslant C \|a\|_{\oplus_{i=1}^\infty \mathcal{H}_i} \leqslant \infty$. If each $J_i$ is a linear operator, such a $C$ exists precisely if the operator norms (defined below) of all $J_i$ are smaller than some constant.

**Operator Norm:**  Let $J : \mathcal{H} \to \widetilde{\mathcal{H}}$ be a linear operator between Hilbert spaces. We measure its 'size' by what is called the operator norm, denoted by $\| \cdot \|_{op}$ and defined by

$$\|J\|_{op} := \sup_{\psi \in \mathcal{H}, \|\psi\|_{\mathcal{H}} = 1} \frac{\|A\psi\|_{\widetilde{\mathcal{H}}}}{\|\psi\|_{\mathcal{H}}}.$$

**Adjoint Operators**   Let $J : \mathcal{H} \to \widetilde{\mathcal{H}}$ be a linear operator from the Hilbert space $\mathcal{H}$ to the Hilbert space $\widetilde{\mathcal{H}}$. Its adjoint $J^* : \widetilde{\mathcal{H}} \to \mathcal{H}$ is an operator mapping in the opposite direction. It is uniquely determined by demanding that

$$\langle Jf, u \rangle_{\widetilde{\mathcal{H}}} = \langle f, J^*u \rangle_{\mathcal{H}}$$

holds true for arbitrary $f \in \mathcal{H}$ and $u \in \widetilde{\mathcal{H}}$.

**Normal Operators:**   If a linear operator $\Delta : \mathcal{H} \to \mathcal{H}$ maps from and to the same Hilbert space, we can compare it directly with its adjoint. If $\Delta\Delta^* = \Delta^*\Delta$, we say that the operator $\Delta$ is normal. Special instances of normal operators are self-adjoint operators, for which we have the stronger property $\Delta = \Delta^*$. If an operator is normal, there are unitary maps $U : \mathcal{H} \to \mathcal{H}$ diagonalizing $\Delta$ as

$$U^*\Delta U = \mathrm{diag}(\lambda_1, ... \lambda_n),$$

with eigenvalues in $\mathbb{C}$. We call the collection of eigenvalues the spectrum $\sigma(\Delta)$ of $\Delta$. If $\dim \mathcal{H} = d$, we may write $\sigma(\Delta) = \{\lambda\}_{i=1}^d$. It is a standard exercise to verify that each eigenvalue satisfies $|\lambda_i| \leqslant \|\Delta\|_{op}$. Associated to each eigenvalue is an eigenvector $\phi_i$. The collection of all (normalized) eigenvectors forms an orthonormal basis of $\mathcal{H}$. We may then write

$$\Delta f = \sum_{i=1}^d \lambda_i \langle \phi_i, f \rangle_{\mathcal{H}} \phi_i.$$

**Resolvent of a (normal) Operator:**   Given a normal operator $\Delta$ on some Hilbert space $\mathcal{H}$, we have that the operator $(\Delta - z) : \mathcal{H} \to \mathcal{H}$ is invertible precisely if $z \neq \sigma(\Delta)$. In this case we write

$$R(z, \Delta) = (\Delta - z)^{-1}$$

and call this operator the **resolvent** of $\Delta$ at $z$. It can be proved that the norm of the resolvent satisfies

$$\|R(z, \Delta)\|_{op} = \frac{1}{dist(z, \sigma(\Delta))},$$

where $dist(z, \sigma(\Delta))$ denotes the minimal distance between $z$ and any eigenvalue of $\Delta$.

**Functional Calculus:**   Given a normal operator $\Delta : \mathcal{H} \to \mathcal{H}$ on a Hilbert space of dimension $d$ and a complex function $g : \mathbb{C} \to \mathbb{C}$, we can define another normal operator obtained from applying the function $g$ to $\Delta$ by

$$g(\Delta)f = \sum_{i=1}^f g(\lambda_i)\langle \phi_i, f \rangle_{\mathcal{H}} \phi_i.$$

For example if $g(\cdot) = |\cdot|$, we obtain the absolute value $|\Delta|$ of $\Delta$ by specifying for all $f \in \mathcal{H}$ that

$$|\Delta|f = \sum_{i=1}^d |\lambda_i|\langle \phi_i, f \rangle_{\mathcal{H}} \phi_i.$$

Similarly we find (if $z \notin \sigma(\Delta)$ and for $f \in \mathcal{H}$)

$$\frac{1}{\Delta - z} = \sum_{i=1}^d \frac{1}{\lambda_i - z}\langle \phi_i, f \rangle_{\mathcal{H}} \phi_i = (\Delta - z)^{-1} = R(z, \Delta)$$

where we think of the left-hand-side as applying a function to $\Delta$, while we think of the right-hand-side as inverting the operator $(\Delta - z)$. This now allows us to apply tools from complex analysis also to operators: If a function $g$ is analytic (i.e. can be expanded into a power series), we have

$$g(\lambda) = -\frac{1}{2\pi i} \oint_S \frac{g(z)}{\lambda - z} dz$$

for any circle $S \subseteq \mathbb{C}$ encircling $\lambda$ by Cauchy's integral formula. Thus, if we chose $S$ large enough to encircle the entire spectrum $\sigma(\Delta)$, we have

$$g(\Delta)f = -\sum_{i=1}^d \frac{1}{2\pi i} \oint_S \frac{g(z)}{\lambda_i - z} dz \langle \phi_i, f \rangle_{\mathcal{H}} \phi_i = -\frac{1}{2\pi i} \oint_S g(z)R(z, \lambda)dz.$$

**Frobenius Norm:** Given a finite dimensional Hilbert space $\mathcal{H}$ with inner product $\langle \cdot, \cdot \rangle_{\mathcal{H}}$, and an orthonormal basis $\{\phi_i\}_{i=1}^d$, we define the trace of an operator $A : \mathcal{H} \to \mathcal{H}$ as

$$Tr(A) := \sum_{k=1}^d \langle \phi_k, A\phi_k \rangle_{\mathcal{H}}.$$

It is a standard exercise to show that this is independent of the choice of orthonormal basis. The associated Frobenius inner product on the space of operators is then given as

$$\langle B, A \rangle_F := Tr(B^*A) \sum_{k=1}^d \langle \phi_k, B^*A\phi_k \rangle_{\mathcal{H}}.$$

Hence the Frobenius norm of an operator is determined by

$$\|A\|_F^2 = Tr(A^*A) = \sum_{k=1}^d \langle \phi_k, A^*A\phi_k \rangle_{\mathcal{H}}.$$

It is a standard exercise to verify that we have $\|A\|_{op} \leq \|A\|_F$. Since the trace is independent of the choice of orthonormal basis, the Frobenius norm is invariant under unitary transformations. More precisely, if $U, V : \mathcal{H} \to \mathcal{H}$ are unitary, we have

$$\|UAV\|_F^2 = \|A\|_F^2.$$

Frobenius norms can be used to transfer Lipschitz continuity properties of complex functions to the setting of functions applied to normal operators:

**Lemma A.1.** Let $g : \mathbb{C} \to \mathbb{C}$ be Lipschitz continuous with Lipschitz constant $D_g$. This implies

$$\|g(X) - g(Y)\|_F \leq D_g \cdot \|X - Y\|_F.$$

for normal operators $X, Y$ on $\mathcal{H}$.

*Proof.* This proof is taken (almost) verbatim from [37]. For an operator $A : \mathcal{H} \to \mathcal{H}$ denote by $A_{ij}$ its matrix representation with respect to the orthonormal basis $\{\phi_i\}_{i=1}^d$:

$$A_{ij} := \langle \phi_i, A\phi_j \rangle_{\mathcal{H}}.$$

We then have

$$\|A\|_F^2 = \sum_{i,j=1}^d |A_{ij}|^2$$

as a quick calculation shows. Let now $U, W$ be unitary (with respect to the inner product $\langle \cdot, \cdot \rangle_{\mathcal{H}}$) operators diagonalizing the normal operators $X$ and $Y$ as

$$V^*XV = \mathrm{diag}(\lambda_1, ...\lambda_n) =: D(X)$$
$$W^*YW = \mathrm{diag}(\mu_1, ...\mu_n) =: D(Y).$$

Since the Frobenius norm is invariant under unitary transformations we find

$$
\begin{aligned}
\|g(X) - g(Y)\|_F^2 &= \|g(VD(X)V^*) - g(WD(Y)W^*)\|_F^2 \\
&= \|Vg(D(X))V^* - Wg(D(Y))W^*\|_F^2 \\
&= \|W^*Vg(D(X)) - g(D(Y))W^*V\|_F^2 \\
&= \sum_{i,j=1}^{d} |(W^*Vg(D(X)) - g(D(Y))W^*V)_{ij}|^2 \\
&= \sum_{i,j=1}^{d} \left| \sum_{k=1}^{n} [W^*V]_{ik}[g(D(X))]_{kj} - [g(D(Y))]_{ik}[W^*V]_{kj} \right|^2 \\
&= \sum_{i,j=1}^{d} |[W^*V]_{ij}|^2 \, |g(\lambda_j) - g(\mu_i)|^2 \\
&\leqslant \sum_{i,j=1}^{d} |[W^*V]_{ij}|^2 \, D_g^2 |\lambda_j - \mu_i|^2 \\
&= D_g^2 \sum_{i,j=1}^{d} \left| \sum_{k=1}^{n} [W^*V]_{ik}[D(X)]_{kj} - [D(Y)]_{ik}[W^*V]_{kj} \right|^2 \\
&= D_g^2 \|X - Y\|_F^2.
\end{aligned}
$$

$\square$

# B   Proof of Theorem 2.1

**Theorem B.1.** Let $\Delta : \ell^2(G) \to \ell^2(G)$ be normal. If the family $\{g_i(\cdot)\}_{i \in I}$ of bounded functions satisfies $A \leqslant \sum_{i \in I} |g_i(c)|^2 \leqslant B$ for all $c$ in the spectrum $\sigma(\Delta)$, we have ($\forall f \in \ell^2(G)$)

$$
A\|f\|_{\ell^2(G)}^2 \leqslant \sum_{i \in I} \|g_i(\Delta)f\|_{\ell^2(G)}^2 \leqslant B\|f\|_{\ell^2(G)}^2.
$$

*Proof.* Writing the normalized eigenvalue-eigenvector sequence of $\Delta$ as $(\lambda_i, \phi_i)_{i=1}^{|G|}$, we simply note

$$
\sum_{i \in I} \sum_{k=1}^{|G|} |\langle g_i(\lambda_k)\phi_k, f \rangle_{\ell^2(G)}|^2 = \sum_{k=1}^{|G|} \left( \sum_{i \in I} |g_i(\lambda_k)|^2 \right) |\langle \phi_k, f \rangle_{\ell^2(G)}|^2.
$$

Now under the assumption, we can estimate the sum in brackets by $A$ from below and by $B$ from above. Then we need only use Bessel's (in)equality to prove

$$
A\|f\|^2 \leqslant \sum_{i \in \hat{I}} \sum_{k=1}^{|G|} |\langle g_i(\lambda_k)\phi_k, f \rangle_{\ell^2(G)}|^2 \leqslant B\|f\|^2.
$$

$\square$

# C   Proof of Theorem 4.1

**Theorem C.1.** With the notation of Section 3 and setting $B_0 = 1$, we have:

$$
\|\Phi_N(f) - \Phi_N(h)\|_{\mathscr{F}_N}^2 \leqslant \left( 1 + \sum_{n=1}^{N} \max\{[B_n(L_n^+R_n^+)^2 - 1], 0\} \prod_{k=0}^{n-1} B_k(R_k^+L_k^+)^2 \right) \|f - h\|_{\ell^2(G)}^2
$$

To streamline the argumentation let us first introduce some notation:

**Notation C.2.** Let us denote paths in $\Gamma^N$ as $q := (\gamma_N, ..., \gamma_1)$. For $f \in \ell^2(G)$ let us write

$$f_q := U[\gamma_N] \circ ... \circ U[\gamma_1](f).$$

*Proof.* By Definition, we have

$$\|\Phi_N(f) - \Phi_N(g)\|^2_{\mathscr{F}_N} = \sum_{n=1}^{N} \left( \sum_{q \in \Gamma^{n-1}} \|V_n(f_q) - V_n(h_q)\|^2_{\ell^2(G_n)} \right)$$

$$= \sum_{n=1}^{N} \underbrace{\left( \sum_{q \in \Gamma^{n-1}} \|\chi_n(\Delta_n)\rho_n(P_n(f_q)) - \chi_n(\Delta_n)\rho_n(P_n(h_q))\|^2_{\ell^2(G_n)} \right)}_{=:a_n}.$$

We proceed in two steps:
Our initial goal is to upper bound $a_n$ as

$$a_n \leqslant B_n(L_n^+ R_n^+)^2 \cdot b_{n-1} - b_n \equiv (b_{n-1} - b_n) + \left[ B_n(L_n^+ R_n^+)^2 - 1 \right] \cdot b_{n-1} \tag{5}$$

for $b_n := \sum_{q \in \Gamma^n} \|f_q - h_q\|^2_{\ell^2(G_n)}$ with $b_0 = \|f - h\|^2_{\ell^2(G)}$. To achieve this we note that (5) is equivalent to

$$a_n + b_n \leqslant B_n(L_n^+ R_n^+)^2 \cdot b_{n-1}$$

which upon unraveling definitions may be written as

$$\sum_{q \in \Gamma^{n-1}} \|\chi_n(\Delta_n)\rho_n(P_n((f_q))) - \chi_n(\Delta_n)\rho_n(P_n(h_q)\|^2_{\ell^2(G_n)} + \sum_{\hat{q} \in \Gamma^n} \|f_{\hat{q}} - h_{\hat{q}}\|^2_{\ell^2(G_n)}$$

$$\leqslant B_n(L_n^+ R_n^+)^2 \sum_{q \in \Gamma^{n-1}} \|f_q - h_q\|^2_{\ell^2(G_{n-1})}. \tag{6}$$

To establish (6), we note, that in the sum over paths of length $n$, any $\hat{q} \in \Gamma^n$ can uniquely be written as $\hat{q} = (\gamma_n, q)$, with the path $q \in \Gamma^{n-1}$ of length $(n-1)$ determined by

$$\hat{q} = (\gamma_n, \underbrace{\gamma_{n-1}, ..., \gamma_1}_{=:q}).$$

With this we find

$$\sum_{\hat{q} \in \Gamma^n} \|f_{\hat{q}} - h_{\hat{q}}\|^2_{\ell^2(G_n)} = \sum_{\gamma_n \in \Gamma_n} \sum_{q \in \Gamma^{n-1}} \|g_{\gamma_n}(\Delta_n)\rho_n(P_n((f_q))) - g_{\gamma_n}(\Delta_n)\rho_n(P_n(h_q))\|^2_{\ell^2(G_n)}.$$

Thus we can rewrite the left hand side of (6) as

$$\sum_{q \in \Gamma^{n-1}} \|\chi_n(\Delta_n)\rho_n(P_n((f_q))) - \chi_n(\Delta_n)\rho_n(P_n(h_q)\|^2_{\ell^2(G_n)} + \sum_{\hat{q} \in \Gamma^n} \|f_{\hat{q}} - h_{\hat{q}}\|^2_{\ell^2(G_n)}$$

$$= \sum_{q \in \Gamma^{n-1}} \left( \|\chi_n(\Delta_n)\rho_n(P_n(f_q) - \chi_n(\Delta_n)\rho_n(P_n(h_q)\|^2_{\ell^2(G_n)} \right.$$

$$\left. + \sum_{\gamma_n \in \Gamma_n} \|g_{\gamma_n}(\Delta_n)\rho_n(P_n((f_q))) - g_{\gamma_n}(\Delta_n)\rho_n(P_n(h_q))\|^2_{\ell^2(G_n)} \right)$$

$$=: \star$$

The fact that in each layer the function $\{\chi_n(\cdot)\} \bigcup \{g_{\gamma_n}(\cdot)\}_{\gamma_n \in \Gamma_n}$ form a generalized frame with upper frame constant $B_n$ implies by Theorem 2.1, that we can further bound this as

$$\star \leqslant B_n \sum_{q \in \Gamma^{n-1}} \|\rho_n(P_n(f_q)) - \rho_n(P_n(h_q))\|^2_{\ell^2(G_n)}.$$

Using the Lipschitz continuity of $\rho_n$ and $P_n$, we arrive at the desired expression (6).

Having established that

$$a_n \leqslant (b_{n-1} - b_n) + \left[ B_n (L_n^+ R_n^+)^2 - 1 \right] \cdot b_{n-1}$$

holds true, we note that we can establish

$$b_{n-1} \leqslant \prod_{k=1}^{n-1} B_k (L_k^+ R_k^+)^2 b_{n-2}$$

arguing similarly as in the case of (6) by using (for $f \in \ell^2(G_{n-1})$)

$$\sum_{\gamma_{n-1} \in \Gamma_{n-1}} \|g_{\gamma_{n-1}}(\Delta_{n-1}) f\|_{\ell^2(G_{n-1})}^2 \leqslant \|\chi_{n-1}(\Delta_{n-1}) f\|_{\ell^2(G_{n-1})}^2 + \sum_{\gamma \in \Gamma} \|g_{\gamma_{n-1}}(\Delta_{n-1}) f\|_{\ell^2(G_{n-1})}^2$$

together with the frame property and Lipschitz continuities. We then iterate this inequality and recall that $b_0 = \|f - h\|_{\ell^2(G)}^2$. Using the fact that

$$\sum_{n=1}^{N} (b_{n-1} - b_n) = b_0 - b_N \leqslant b_0,$$

we finally find

$$\|\Phi_N(f) - \Phi_N(h)\|_{\mathscr{F}_N}^2 \leqslant \left( 1 + \sum_{n=1}^{N} \max\{[B_n(L_n^+ R_n^+)^2 - 1], 0\} \prod_{k=0}^{n-1} B_k (R_k^+ L_k^+)^2 \right) \|f - h\|_{\ell^2(G)}^2.$$

$\square$

## D   Proof or Theorem 4.2

**Theorem D.1.** Let $\Phi_N$ and $\widetilde{\Phi}_N$ be two scattering transforms based on the same module sequence $\Omega_N$ and operator sequences $\mathscr{D}_N, \widetilde{\mathscr{D}}_N$ with the same connecting operators ($P_n = \widetilde{P}_n$) in each layer. Assume $R_n^+, L_n^+ \leqslant 1$ and $B_n \leqslant B$ for some $B$ and $n \leqslant N$. Assume that the respective normal operators satisfy $\|\Delta_n - \widetilde{\Delta}_n\|_F \leqslant \delta$ for some $\delta > 0$. Further assume that the functions $\{g_{\gamma_n}\}_{\gamma_n \in \Gamma_n}$ and $\chi_n$ in each layer are Lipschitz continuous with associated Lipschitz constants satisfying $L_{\chi_n}^2 + \sum_{\gamma_n \in \Gamma_n} L_{g_{\gamma_n}}^2 \leqslant D^2$ for all $n \leqslant N$ and some $D > 0$. Then we have

$$\|\widetilde{\Phi}_N(f) - \Phi_N(f)\|_{\mathscr{F}_N} \leqslant \sqrt{2(2^N - 1)} \cdot \sqrt{(\max\{B, 1/2\})^{N-1}} \cdot D \cdot \delta \cdot \|f\|_{\ell^2(G)}$$

for all $f \in \ell^2(G)$. If $B \leqslant 1/2$, the stability constant improves to $\sqrt{2(1 - B^N)/(1 - B)} \cdot D \leqslant 2 \cdot D$.

**Notation D.2.** Let us denote scattering propagators based on operators $\Delta_n$ and connecting operators $P_n$ by $U_n$ and scattering propagators based on operators $\widetilde{\Delta}_n$ by $\widetilde{U}_n$. Similarly, to Notation C.2, let us then write (with $q = (\gamma_N, ..., \gamma_1)$)

$$\widetilde{f}_q := \widetilde{U}_n[\gamma_n] \circ ... \circ \widetilde{U}_1[\gamma_1](f).$$

*Proof.* By definition we have

$$\|\Phi_N(f) - \widetilde{\Phi}_N\|_{\mathscr{F}_N}^2 = \sum_{n=1}^{N} \underbrace{\left( \sum_{q \in \Gamma^{n-1}} \|\chi_n(\Delta_n)\rho_n(P_n((f_q))) - \chi_n(\widetilde{\Delta}_n)\rho_n(P_n(\widetilde{f}_q))\|_{\ell^2(G_n)}^2 \right)}_{=:a_n}.$$

We define $b_n := \sum_{q \in \Gamma^n} \|f_q - \widetilde{f}_q\|_{\ell^2(G_n)}^2$, with $b_0 = \|f - h\|_{\ell^2(G)}^2 = 0$ and note

$$a_n + b_n = \sum_{q \in \Gamma^{n-1}} \left( \|\chi_n(\Delta_n)\rho_n(P_n(f_q)) - \chi_n(\widetilde{\Delta}_n)\rho_n(P_n(\widetilde{f}_q))\|_{\ell^2(G_n)}^2 \right.$$

$$\left. + \sum_{\gamma_n \in \Gamma_n} \|g_{\gamma_n}(\Delta_n)\rho_n(P_n((f_q))) - g_{\gamma_n}(\widetilde{\Delta}_n)\rho_n(P_n(\widetilde{f}_q))\|_{\ell^2(G_n)}^2 \right).$$

Using (with $|a + b|^2 \leqslant 2(|a|^2 + |b|^2)$)

$$\frac{1}{2}\|g_{\gamma_n}(\Delta_n)\rho_n(P_n(f_q)) - g_{\gamma_n}(\widetilde{\Delta}_n)\rho_n(P_n(\widetilde{f}_q))\|^2_{\ell^2(G_n)}$$

$$\leqslant \|[g_{\gamma_n}(\Delta_n) - g_{\gamma_n}(\widetilde{\Delta}_n)]\rho_n(P_n(f_q))\|^2_{\ell^2(G_n)}$$

$$+ \|g_{\gamma_n}(\widetilde{\Delta}_n)[\rho_n(P_n((f_q))) - \rho_n(P_n(\widetilde{f}_q))]\|^2_{\ell^2(G_n)}$$

$$\leqslant \|[g_{\gamma_n}(\Delta_n) - g_{\gamma_n}(\widetilde{\Delta}_n)]\|^2_\infty \cdot \|\rho_n(P_n(f_q))\|^2_{\ell^2(G_n)}$$

$$+ \|g_{\gamma_n}(\widetilde{\Delta}_n)[\rho_n(P_n(f_q)) - \rho_n(P_n(\widetilde{f}_q))]\|^2_{\ell^2(G_n)},$$

and

$$\|[g_{\gamma_n}(\Delta_n) - g_{\gamma_n}(\widetilde{\Delta}_n)]\|^2_\infty \leqslant \|[g_{\gamma_n}(\Delta_n) - g_{\gamma_n}(\widetilde{\Delta}_n)]\|^2_F \leqslant L^2_{g_\gamma} \cdot \delta^2$$

(c.f. Lemma A.1 ), we find

$$a_n + b_n \leqslant 2 \sum_{q \in \Gamma^{n-1}} \left( L^2_{\chi_n} + \sum_{\gamma_n \in \Gamma_n} L_{g^2_{\gamma_n}} \right) (L^+_n R^+_n)^2 \delta^2 \|\rho_n(P_n(f_q))\|^2_{\ell^2(G_n)}$$

$$+ 2 \sum_{q \in \Gamma^{n-1}} B_n \|\rho_n(P_n(f_q)) - \rho_n(P_n(\widetilde{f}_q))\|^2_{\ell^2(G_n)}.$$

Using $L^2_{\chi_n} + \sum_{\gamma_n \in \Gamma_n} L^2_{\gamma_n} \leqslant D^2$, we then infer (using the assumption $L^+_n, R^+_n \leqslant 1$)

$$a_n \leqslant (b_{n-1} - b_n) + [2B - 1]b_{n-1} + B^{n-1}2D^2\delta^2 \|f\|_{\ell^2(G)}.$$

Now if $B \leqslant \frac{1}{2}$, we have

$$a_n \leqslant (b_{n-1} - b_n) + B^{n-1}2D^2\delta^2 \|f\|_{\ell^2(G)}$$

and results of geometric sums leads to the desired bound after summing over $n$.
Hence let us assume $B > \frac{1}{2}$. Using similar arguments as before, we find

$$b_{n-1} \leqslant B^{n-2}2D^2\delta^2 \|f\|^2_{\ell^2(G)} + 2Bb_{n-2} \leqslant B^{n-2}2D^2\delta^2 \|f\|^2_{\ell^2(G)} + B^{n-2}4D^2\delta^2 \|f\|^2_{\ell^2(G)} + 4b_{n-3}$$

$$\leqslant B^{n-2} \left( \sum_{k=1}^{n-1} 2^k \right) D^2\delta^2 \|f\|^2_{\ell^2(G)} = B^{n-2}(2^n - 2)D^2\delta^2 \|f\|^2_{\ell^2(G)}.$$

Thus we now know

$$a_n \leqslant 2D^2\delta^2 B^{n-1} \|f\|^2_{\ell^2(G)} + [2B - 1](2^n - 2)D^2\delta^2 B^{n-2} \|f\|^2_{\ell^2(G)} + (b_{n-1} - b_n)$$

In total we find

$$\sqrt{\sum_{n=1}^N a_n} \leqslant \sqrt{2(2^N - 1)} \cdot \sqrt{B^{N-1}} \cdot D \cdot \delta \cdot \|f\|_{\ell^2(G)},$$

where we have estimated the sum over $(b_{n-1} - b_n)$ by zero from above again. This establishes the claim. $\qquad \square$

**Remark D.3.** To see that this also holds for our Architecture I of Fig. 2, we note that the critical step is establishing that Lemma A.1 also applies to $\delta_0$ and $\overline{\cos}$, as defined in Section 2. Here we establish that

$$\|\delta_0(\Delta) - \delta_0(\widetilde{\Delta})\|_F = 0$$

and

$$\|\overline{\cos}(\Delta) - \overline{\cos}(\widetilde{\Delta})\|_F \leqslant D_{\cos} \|\Delta - \widetilde{\Delta}\|_F.$$

Indeed, since $\Delta$ and $\widetilde{\Delta}$ are (possibly) rescaled graph Laplacians on the same graph, the spectral projections to their lowest lying eigen space, associated to the eigenvalue $\lambda_{\min} = 0$ agree. Denoting this spectral projection by $\mathcal{P}$, we have

$$\overline{\cos}(\Delta) - \overline{\cos}(\widetilde{\Delta}) = [\cos(\Delta) - \mathcal{P}] - [\cos(\widetilde{\Delta}) - \mathcal{P}] = \cos(\Delta) - \cos(\widetilde{\Delta})$$

and we can apply Lemma A.1. Similar considerations apply to $\delta_0$.

## E Prototypical Example illustrating $\omega$-$\delta$ Closeness and $\delta$-Unitary Equivalence

To investigate the example of Figure 3, we label the vertices of the respective graphs as depicted in Figure 6. We denote the left graph by $G$ and the right graph by $\widetilde{G}$. The node-weights on $\widetilde{G}$ are given as $\widetilde{\mu}_i = 1$ for $1 \leqslant i \leqslant 7$, while on $G$ the weights are given as $\mu_i = 1$ for $1 \leqslant i \leqslant 5$ while $\mu_6 = 2$. We then consider the respective un-normalized graph Laplacians $\Delta : \ell^2(G) \to \ell^2(G)$ and $\widetilde{\Delta} : \ell^2(\widetilde{G}) \to \ell^2(\widetilde{G})$, which for a given adjacency matrix $W$ on a graph signal space $\ell^2(G)$ with node weights $\{\mu_i\}_i$ is given as

$$(\Delta f)_i = \frac{1}{\mu_i} \sum_j W_{ij}(f_i - f_j).$$

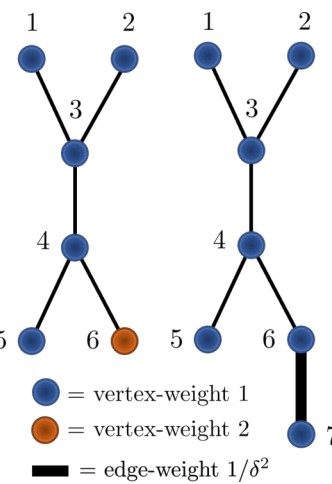

Such operators are positive and hence $|\Delta| = \Delta$ (similarly for $\widetilde{\Delta}$). We now need to find operators $J : \ell^2(G) \to \ell^2(\widetilde{G})$ and $\widetilde{J} : \ell^2(\widetilde{G}) \to \ell^2(G)$ satisfying the conditions of Definition 4.3. To construct $J$, we define a family $\{\psi_i\}_{i=1}^6$ of vectors on $\ell^2(\widetilde{G})$ as

$$\begin{aligned}
\psi_1 &= (1,0,0,0,0,0,0), & \psi_2 &= (0,1,0,0,0,0,0), \\
\psi_3 &= (0,0,1,0,0,0,0), & \psi_4 &= (0,0,0,1,0,0,0), \\
\psi_5 &= (0,0,0,0,1,0,0), & \psi_6 &= (0,0,0,0,0,1,1).
\end{aligned}$$

Figure 6: Indexing on the respective graphs

The map $J : \ell^2(G) \to \ell^2(\widetilde{G})$ is then defined as

$$Jf := \sum_{i=1}^6 f_i \psi_i,$$

for any $f \in \ell^2(G)$. We take $\widetilde{J} : \ell^2(\widetilde{G}) \to \ell^2(G)$ to be its adjoint ($\widetilde{J} := J^*$), which determined explicitly by

$$(\widetilde{J}u)_i = \frac{1}{\mu_i}\langle \psi_i, u\rangle_{\ell^2(\widetilde{G})}$$

for any $u \in \ell^2(\widetilde{G})$ We shall now first check the conditions for $\delta$-quasi unitary equivalence, which we list again for convenience; now adapted to our current setting:

$$\|Jf\|_{\ell^2(\widetilde{G})} \leqslant 2\|f\|_{\ell^2(G)}, \quad \|(J - \widetilde{J}^*)f\|_{\ell^2(\widetilde{G})} \leqslant \delta\|f\|_{\ell^2(G)},$$

$$\|f - \widetilde{J}Jf\|^2_{\ell^2(G)} \leqslant \delta^2\left(\|f\|^2_{\ell^2(G)} + \langle f, \Delta\, f\rangle_{\ell^2(G)}\right), \quad \|u - J\widetilde{J}u\|^2_{\ell^2(\widetilde{G})} \leqslant \delta^2\left(\|u\|^2_{\ell^2(\widetilde{G})} + \langle u, \widetilde{\Delta}\, u\rangle_{\ell^2(\widetilde{G})}\right).$$

We first note that since $\widetilde{J} = J^*$, we have $\|(J - \widetilde{J}^*)f\|_{\ell^2(\widetilde{G})} = 0$. Next we note

$$\|Jf\|^2_{\ell^2(\widetilde{G})} = \sum_{i=1}^7 |(Jf)_i|^2 = |f_6|^2 + \sum_{i=1}^6 |f_i|^2 = \sum_{i=1}^6 \mu_i = \|f\|^2_{\ell^2(G)}.$$

Furthermore we note

$$(\widetilde{J}Jf)_i = \sum_{k=1}^6 f_k \underbrace{\frac{1}{\mu_i}\langle \psi_i, \psi_k\rangle_{\ell^2(\widetilde{G})}}_{=\delta_{ik}} = f_i$$

and hence $\|f - \widetilde{J}Jf\|^2_{\ell^2(G)} = 0$. It remains to control $\|u - J\widetilde{J}u\|^2_{\ell^2(\widetilde{G})}$. We note

$$\widetilde{J}u = (u_1, u_2, u_3, u_4, u_5, (u_5 + u_6)/2)^\top$$

and thus

$$J\widetilde{J}u = (u_1, u_2, u_3, u_4, u_5, (u_6 + u_7)/2, (u_6 + u_7)/2)^\top,$$

Which implies

$$u - J\tilde{J}u = (0, 0, 0, 0, 0, (u_7 - u_6)/2, (u_6 - u_7)/2)^\top,$$

and thus

$$\|u - J\tilde{J}u\|_{\ell^2(\tilde{G})}^2 = 2\frac{|u_6 - u_7|^2}{4} = \frac{|u_6 - u_7|^2}{2}.$$

We have

$$\langle u, \tilde{\Delta}\, u\rangle_{\ell^2(\tilde{G})} = \frac{1}{2} \sum_{i,j=1}^d \widetilde{W}_{ij}|u_i - u_j|^2.$$

Since $\widetilde{W}_{67} = 1/\delta^2$ by assumption, we have

$$\|u - J\tilde{J}u\|_{\ell^2(\tilde{G})}^2 = \frac{1}{2}|u_6 - u_7|^2 = \frac{1}{2}\frac{\delta^2}{\delta^2}|u_6 - u_7|^2 = \frac{1}{2}\delta^2\widetilde{W}_{67}|u_6 - u_7|^2$$

$$\leqslant \frac{1}{2}\delta^2 \sum_{i,j=1}^d \widetilde{W}_{ij}|u_i - u_j|^2 = \delta^2\langle u, \tilde{\Delta}\, u\rangle_{\ell^2(\tilde{G})}$$

$$\leqslant \delta^2 \left(\|u\|_{\ell^2(\tilde{G})}^2 + \langle u, \tilde{\Delta}\, u\rangle_{\ell^2(\tilde{G})}\right).$$

Thus we have proven $\delta$-unitary-equivalence and it remains to establish $(-1)$-$12\delta$ closeness. Combining Proposition 4.4.12. and Theorem 4.4.15 of [29], instead of bounding $\|(\tilde{R}J - JR)f\|_{\ell^2(\tilde{G})} \leqslant 12\delta\|f\|_{\ell^2(G)}$ directly, we may instead establish that there are operators $J^1 : \ell^2(G) \to \ell^2(\tilde{G})$, $\widetilde{J^1} : \ell^2(\tilde{G}) \to \ell^2(G)$ satisfying

$$\|J^1 f - Jf\|_{\ell^2(\tilde{G})} \leqslant \delta^2 \left(\|f\|_{\ell^2(G)} + \langle f, \Delta,\, f\rangle_{\ell^2(G)}\right), \tag{7}$$

$$\|\widetilde{J^1}u - \tilde{J}u\|_{\ell^2(G)} \leqslant \delta^2 \left(\|u\|_{\ell^2(\tilde{G})} + \langle f, \tilde{\Delta},\, u\rangle_{\ell^2(\tilde{G})}\right), \tag{8}$$

and

$$\langle J^1 f, \tilde{\Delta}\, u\rangle_{\ell^2(\tilde{G})} = \langle f, \Delta\, \widetilde{J^1}u\rangle_{\ell^2(G)}. \tag{9}$$

We chose $J^1 = J$ and determine $\widetilde{J^1}$ by setting (for $1 \leqslant i \leqslant 6$)

$$(\widetilde{J^1}u)_i = u_i.$$

Thus (7) is clearly satisfied. For (8) we note that we have

$$(\tilde{J}u - \widetilde{J^1}u) = (0, 0, 0, 0, 0, (u_7 - u_6)/2).$$

Thus we have

$$\|\widetilde{J^1}u - \tilde{J}u\|_{\ell^2(G)} = \frac{1}{2}|u_6 - u_7|^2 \leqslant \delta^2 \left(\|u\|_{\ell^2(\tilde{G})}^2 + \langle u, \tilde{\Delta}\, u\rangle_{\ell^2(\tilde{G})}\right)$$

as before. It remains to establish (9). We have

$$\langle f, \Delta\, \widetilde{J^1}u\rangle_{\ell^2(G)} = \sum_{i,j=1}^6 \overline{f_i}W_{ij}(u_i - u_j),$$

while we have

$$\langle J^1 f, \tilde{\Delta}\, u\rangle_{\ell^2(\tilde{G})} = \sum_{i=1}^6 \overline{f_i} \cdot \langle \psi_i, \tilde{\Delta}\, u\rangle_{\ell^2(\tilde{G})}$$

$$= \sum_{i,j=1}^5 \overline{f_i}W_{ij}(U_j - u_i) + \overline{f_6} \cdot \langle \psi_6, \tilde{\Delta}\, u\rangle_{\ell^2(\tilde{G})}.$$

We have (with all node-weights on $\ell^2(G)$ equal to unity)

$$\langle \psi_6, \widetilde{\Delta}\, u \rangle_{\ell^2(\widetilde{G})} = (\Delta\, u)_6 + (\Delta\, u)_7 = \left( \sum_j W_{6j}(f_6 - f_j) + \frac{1}{\delta^2}(f_6 - f_7) \right) + \left( \frac{1}{\delta^2}(f_7 - f_6) \right)$$

$$= \left( \sum_j W_{6j}(f_6 - f_j) + \frac{1}{\delta^2}(f_6 - f_7) \right)$$

And thus

$$\langle J^1 f, \widetilde{\Delta}\, u \rangle_{\ell^2(\widetilde{G})} = \sum_{i,j=1}^{6} \overline{f_i} W_{ij}(u_i - u_j) = \langle f, \Delta\, \widetilde{J^1} u \rangle_{\ell^2(G)}$$

which proves the claim.

# F   Proof of Lemma 4.4

**Lemma F.1.** In the setting of Definition 4.3 let $\Delta$ and $\widetilde{\Delta}$ be $\omega$-$\delta$-close and satisfy $\|\Delta\|_{op}, \|\widetilde{\Delta}\|_{op} \leqslant K$ for some $K > 0$. If $g : \mathbb{C} \to \mathbb{C}$ is holomorphic on the disk $B_{K+1}(0)$ of radius $(K + 1)$, there is a constant $C_g \geqslant 0$ so that

$$\|g(\widetilde{\Delta})J - Jg(\Delta)\|_{\mathrm{op}} \leqslant C_g \cdot \delta$$

with $C_g$ depending on $g$, $\omega$ and $K$.

*Proof.* Without loss of generality, let us assume that $K > |\omega|$. Let us denote the circle of radius $r$ in $\mathbb{C}$ by $S_r$. For any holomorphic function $g$ and (normal) operator $\Delta$ whose spectrum is enclosed by the circle $S_r$, we can express the operator $g(\Delta)$ as

$$g(\Delta) = -\frac{1}{2\pi i} \oint_{S_r} \frac{g(z)}{\Delta - z} dz$$

as discussed in Appendix A (see also [7] for more details). Note that in our case the resolvent $R(z, \Delta) = (\Delta - z)^{-1}$ is well defined for $|z| \geqslant K$, since with our assumptions all eigenvalues are within the circle of radius $K$. Additionally note that we have

$$dist(z, \sigma(\Delta)) \geqslant dist(z, S_K) = |z| - K$$

if $|z| \geqslant K$. The same holds true after replacing $\Delta$ with $\widetilde{\Delta}$. Since for any normal operator $\Delta$ we have

$$\|R(z, \Delta)\|_{op} = 1/dist(z, \sigma(\Delta)),$$

we find

$$|R(z, \widetilde{\Delta})\|_{op}, \|R(z, \Delta)\|_{op} \leqslant 1/(|z| - K).$$

To quantify the difference $\|R(z, \widetilde{\Delta})J - JR(z, \Delta)\|_{op}$ in terms of the difference $\|\widetilde{R}(\omega)J - JR(\omega)\|_{op} \leqslant \delta$, we define the function

$$\gamma_0(z) := 1 + \frac{|z - \omega|}{|z| - K},$$

for which

$$\|R(z, \widetilde{\Delta})J - JR(z, \Delta)\|_{op} \leqslant \gamma_0(z)^2 \|R(\omega, \widetilde{\Delta})J - JR(\omega, \widetilde{\Delta})\|_{op}$$

holds, as proved (in more general form) in Lemma 4.5.9 in [29]. Since on $S_{K+1}$ we have and $|z - \omega| \leqslant 2K + 1$ hence $\gamma_0(z) \leqslant 2(K + 1)$, we find

$$\|g(\widetilde{\Delta})J - Jg(\Delta)\|_{\mathrm{op}} = \left\| \frac{1}{2\pi i} \oint_{S_{K+1}} g(z) \left( R(z, \widetilde{\Delta}) - R(z, \Delta) \right) dz \right\|_{op}$$

$$\leqslant \frac{1}{2\pi} \oint_{S_{K+1}} |g(z)| \left\| R(z, \widetilde{\Delta}) - R(z, \Delta) \right\|_{op} dz$$

$$\leqslant 2\frac{(K + 1)^2}{\pi} \left( \oint_{S_{K+1}} |g(z)| dz \right) \cdot \|R(\omega, \widetilde{\Delta})J - JR(\omega, \widetilde{\Delta})\|_{op}.$$

Thus we may set

$$C_g := 2\frac{(K+1)^2}{\pi} \oint_{S_{K+1}} |g(z)|dz.$$

$\square$

# G   Proof of Theorem 4.5

We state and prove a somewhat more general theorem, incorporating also the case where the identification operators only almost commute with connecting operators or non-linearities. We also would like to point out that the constant 2 in Definition 4.3 is arbitrary and any constant larger than one would suffice. Much more details are provided in Chapter IV of [29].

**Theorem G.1.** Let $\Phi_N, \widetilde{\Phi}_N$ be scattering transforms based on a common module sequence $\Omega_N$ and differing operator sequences $\mathscr{D}_N, \widetilde{\mathscr{D}}_N$. Assume $R_n^+, L_n^+ \leqslant 1$ and $B_n \leqslant B$ for some $B$ and $n \geqslant 0$. Assume that there are identification operators $J_n : \ell^2(G_n) \to \ell^2(\widetilde{G}_n)$, $\widetilde{J}_n : \ell^2(\widetilde{G}_n) \to \ell^2(G_n)$ $(0 \leqslant n \leqslant N)$ so that the respective signal spaces are $\delta$-unitarily equivalent, the respective normal operators $\Delta_n, \widetilde{\Delta}_n$ are $\omega$-$\delta$-close as well as bounded (in norm) by $K > 0$ and the connecting operators satisfy $\|\widetilde{P}_n J_{n-1}f - J_n P_n f\|_{\ell^2(\widetilde{G}_n)} \leqslant \delta\|f\|_{\ell^2(G_{n-1})}$. For the common module sequence $\Omega_N$ assume that the non-linearities satisfy $\|\rho_n(J_n f) - J_n\rho_n(f)\|_{\ell^2(\widetilde{G}_n)} \leqslant \delta\|f\|_{\ell^2(G_n)}$ and that the constants $C_{\chi_n}$ and $\{C_{g_{\gamma_n}}\}_{\gamma_n \in \Gamma_N}$ associated through Lemma 4.4 to the functions of the generalized frames in each layer satisfy $C_{\chi_n}^2 + \sum_{\gamma_n \in \Gamma_N} C_{g_{\gamma_n}}^2 \leqslant D^2$ for some $D > 0$. Denote the operator that the family $\{J_n\}_n$ of identification operators induce on $\mathscr{F}_N$ through concatenation by $\mathscr{J}_N : \mathscr{F}_N \to \widetilde{\mathscr{F}}_N$. Then we have with $K_N = \sqrt{(8^N - 1)(2D^2 + 12B)/7 \cdot B^{N-1}}$ if $B > 1/8$ and $K_N = \sqrt{(2D^2 + 12B) \cdot (1 - B^N)/(1 - B)}$ if $B \leqslant 1/8$ that

$$\|\widetilde{\Phi}_N(J_0 f) - \mathscr{J}_N \Phi_N(f)\|_{\widetilde{\mathscr{F}}_N} \leqslant K_N \cdot \delta \cdot \|f\|_{\ell^2(G)}, \quad \forall f \in \ell^2(G).$$

If additionally $\|\widetilde{P}_n J_{n-1}f - J_n P_n f\|_{\ell^2(\widetilde{G}_n)} = 0$ or $\|\rho_n(J_n f) - J_n\rho_n(f)\|_{\ell^2(\widetilde{G}_n)} = 0$ holds in each layer, then we have $K_N = \sqrt{(4^N - 1)(2D^2 + 4B)/3 \cdot B^{N-1}}$ if $B > 1/4$ and $K_N = \sqrt{(2D^2 + 4B) \cdot (1 - B^N)/(1 - B)}$ if $B \leqslant 1/4$. If both additional equations hold, we have $K_N = \sqrt{(2^N - 1)2D^2 \cdot B^{N-1}}$ if $B > 1/2$ and $K_N = \sqrt{2D^2 \cdot (1 - B^N)/(1 - B)}$ if $B \leqslant 1/2$.

**Notation G.2.** Let us denote scattering propagators based on operators $\Delta_n$ and $P_n$ by $U_n$ and scattering propagators based on operators $\widetilde{\Delta}_n$ and $\widetilde{P}_n$ by $\widetilde{U}_n$. Similarly, to Notation D.2 and , let us then write (with $q = (\gamma_N, ..., \gamma_1)$)

$$\widetilde{f}_q := \widetilde{U}_n[\gamma_n] \circ ... \circ \widetilde{U}_1[\gamma_1](J_0 f).$$

*Proof.* By definition we have

$$\|\mathscr{J}\Phi_N(f) - \widetilde{\Phi}_N(J_0 f)\|_{\widetilde{\mathscr{F}}_N}^2 = \sum_{n=1}^N \underbrace{\left(\sum_{q \in \Gamma^{n-1}} \|J_n\chi_n(\Delta_n)\rho_n(P_n(f_q)) - \chi_n(\widetilde{\Delta}_n)\rho_n(P_n(\widetilde{f}_q))\|_{\ell^2(\widetilde{G}_n)}^2\right)}_{=:a_n}.$$

We define $b_n := \sum_{q \in \Gamma^n} \|J_n f_q - \widetilde{f}_q\|_{\ell^2(\widetilde{G}_n)}^2$, with $b_0 = \|J_0 f - J_0 f\|_{\ell^2(\widetilde{G})}^2 = 0$ and note

$$a_n + b_n = \sum_{q \in \Gamma^{n-1}} \left(\|J_n\chi_n(\Delta_n)\rho_n(P_n(f_q) - \chi_n(\widetilde{\Delta}_n)\rho_n(P_n(\widetilde{f}_q))\|_{\ell^2(\widetilde{G}_n)}^2 \right.$$

$$\left. + \sum_{\gamma_n \in \Gamma_n} \|J_n g_{\gamma_n}(\Delta_n)\rho_n(P_n((f_q))) - g_{\gamma_n}(\widetilde{\Delta}_n)\rho_n(P_n(\widetilde{f}_q))\|_{\ell^2(\widetilde{G}_n)}^2\right).$$

Using

$$\frac{1}{2}\|J_n g_{\gamma_n}(\Delta_n)\rho_n(P_n((f_q))) - g_{\gamma_n}(\widetilde{\Delta}_n)\rho_n(P_n(\widetilde{f}_q))\|^2_{\ell^2(\widetilde{G}_n)}$$

$$\leqslant \|[J_n g_{\gamma_n}(\Delta_n) - g_{\gamma_n}(\widetilde{\Delta}_n)J_n]\rho_n(P_n(f_q))\|^2_{\ell^2(\widetilde{G}_n)}$$

$$+ \|g_{\gamma_n}(\widetilde{\Delta}_n)[J_n\rho_n(P_n((f_q))) - \rho_n(P_n(\widetilde{f}_q))]\|^2_{\ell^2(\widetilde{G}_n)}$$

$$\leqslant \|[J_n g_{\gamma_n}(\Delta_n) - g_{\gamma_n}(\widetilde{\Delta}_n)J_n]\|_{op} \cdot \|\rho_n(P_n(f_q))\|^2_{\ell^2(\widetilde{G}_n)}$$

$$+ \|g_{\gamma_n}(\widetilde{\Delta}_n)[J_n\rho_n(P_n((f_q))) - \rho_n(P_n(\widetilde{f}_q))]\|^2_{\ell^2(\widetilde{G}_n)},$$

and $\|[g_{\gamma_n}(\Delta_n) - g_{\gamma_n}(\widetilde{\Delta}_n)]\|_\infty \leqslant C^2_{g_\gamma} \cdot \delta^2$ (c.f. Lemma 4.4), we find

$$a_n + b_n \leqslant 2 \sum_{q\in\Gamma^{n-1}} \left( C^2_{\chi_n} + \sum_{\gamma_n\in\Gamma_n} C_{g^2_{\gamma_n}} \right) (L^+_n R^+_n)^2 \delta^2 ||\rho_n(P_n(\widetilde{f}_q))||^2_{\ell^2(\widetilde{G}_n)}$$

$$+ 2 \sum_{q\in\Gamma^{n-1}} B_n ||J_n\rho_n(P_n(f_q)) - \rho_n(P_n(\widetilde{f}_q))||^2_{\ell^2(\widetilde{G}_n)}$$

$$\leqslant 2 \sum_{q\in\Gamma^{n-1}} \delta^2 \left( C^2_{\chi_n} + \sum_{\gamma_n\in\Gamma_n} C_{g^2_{\gamma_n}} \right) (L^+_n R^+_n)^2 ||\rho_n(P_n(\widetilde{f}_q))||^2_{\ell^2(\widetilde{G}_n)}$$

$$+ 4B \cdot B^{n-1}||f||^2_{\ell^2(G)}\delta^2 + 8B \cdot B^{n-1}||f||^2_{\ell^2(G)}\delta^2 + 8Bb_{n-1},$$

where the second inequality arises from permuting the identification operator $J_n$ through non-linearity and connecting operator. Using $C^2_{\chi_n} + \sum_{\gamma_n\in\Gamma_n} C^2_{\gamma_n} \leqslant D^2$, we then infer

$$a_n \leqslant (b_{n-1} - b_n) + [8B - 1]b_{n-1} + (2D^2 + 12B)B^{n-1}\delta^2||f||^2_{\ell^2(G)}.$$

If $B \leqslant \frac{1}{8}$, summing over $n$ and using a geometric sum argument yields the desired stability constant. Hence let us assume $B > \frac{1}{8}$. Using similar arguments as before, we find

$$b_{n-1} \leqslant (2D^2 + 12B)\delta^2 B^{n-2}||f||^2_{\ell^2(G)} + 8Bb_{n-2}$$

$$\leqslant \left( \sum_{k=1}^{n-1} 8^{k-1} \right) B^{n-2}(2D^2 + 12B)\delta^2||f||^2_{\ell^2(G)} = \frac{1}{56}(8^n - 8)(2D^2 + 12)\delta^2||f||^2_{\ell^2(G)}.$$

In total we find

$$\sum_{n=1}^{N} a_n$$

$$\leqslant \underbrace{(b_0 - b_N)}_{\leqslant 0} + (2D^2 + 12B)B^{n-1}\delta^2||f||^2_{\ell^2(G)} + (8B-1)(8^{n-1}-1)/7B^{n-2} \cdot (2D^2 + 12B)\delta^2||f||^2_{\ell^2(G)}$$

$$\leqslant (8^N - 1)(2D^2 + 12B)/7 \cdot B^{N-1}||f||^2_{\ell^2(G)}.$$

If one of the additional equations holds, we find

$$a_n + b_n \leqslant (b_{n-1} - b_n) + [4B - 1]b_{n-1} + (2D^2 + 4B)\delta^2||f||^2_{\ell^2(G)}.$$

and

$$b_{n-1} \leqslant (2D^2 + 4B)\delta^2 B^{n-2}||f||^2_{\ell^2(G)} + 4Bb_{n-2}$$

$$\leqslant \left( \sum_{k=1}^{n-1} 4^{k-1} \right) B^{n-2}(2D^2 + 4)\delta^2||f||^2_{\ell^2(G)} = \frac{1}{12}(4^n - 4)B^{n-2}(2D^2 + 4)\delta^2||f||^2_{\ell^2(G)}.$$

Arguing as previously yields the desired stability bounds.
If both additional equations are satisfied the proof is virtually the same as the one for Theorem 4.2. $\qquad\square$

# H   Details on Energy Decay and Truncation Stability

We first prove the statement made about the relation between truncation stability and energy:

**Lemma H.1.** Given the energy $W_N := \sum_{(\gamma_N,...,\gamma_1)\in\Gamma^N} \|U[\gamma_N]\circ...\circ U[\gamma_1](f)\|^2_{\ell^2(G_N)}$ stored in the network at layer $N$, we have after extending $\Phi_N(f)$ by zero to match dimensions with $\Phi_{N+1}(f)$ that

$$\|\Phi_N(f) - \Phi_{N+1}(f)\|^2_{\mathscr{F}_{N+1}} \leqslant \left(R^+_{N+1}L^+_{N+1}\right)^2 B_{N+1} \cdot W_N.$$

*Proof.* We note

$$\|\Phi_N(f) - \Phi_{N+1}(f)\|^2_{\mathscr{F}_{N+1}} = \sum_{(\gamma_{N-1},...,\gamma_1)\in\Gamma^N} \|V_{N+1}\circ U[\gamma_N]\circ...\circ U[\gamma_1](f)\|^2_{\ell^2(G_{N+1})}$$

$$\leqslant \left(R^+_{N+1}L^+_{N+1}\right)^2 B_{N+1} \sum_{(\gamma_{N-1},...,\gamma_1)\in\Gamma^{N-1}} \|U[\gamma_N]\circ...\circ U[\gamma_1](f)\|^2_{\ell^2(G_{N+1})}.$$

$\square$

In fact one can prove even more:

**Lemma H.2.** The energy $W_N$ stored in layer $N$ satisfies

$$C^-_N\|f\|^2_{\ell^2(G)} \leqslant \|\Phi_N(f)\|_{\mathscr{F}_N} + W_N(f) \leqslant C^+_N\|f\|^2_{\ell^2(G)},$$

with constants $C^-_N := \prod_{i=1}^N \min\left\{1, A_i(L^-_i R^-_i)^2\right\}$ and $C^+_N := \prod_{i=1}^N \max\left\{1, B_i(L^+_i R^+_i)^2\right\}$.

*Proof.*

$$\min\left\{1, A_1(L^-_1 R^-_1)^2\right\}\|f\|^2_{\ell^2(G)}$$
$$=A_1(L^-_1 R^-_1)^2\|f\|^2_{\ell^2(G)}$$
$$=A_1\|\rho_1(P_1(f))\|^2_{\ell^2(G_1)}$$
$$\leqslant \sum_{\gamma_1\in\Gamma_1} \|g_{\gamma_1}(\Delta_1)\rho_1(P_1(f))\|^2_{\ell^2(G_1)} + \|\chi_1(\Delta_1)\rho_1(P_1(f))\|^2_{\ell^2(G_1)}$$
$$= \sum_{q\in\Gamma^1} \|U[q](f)\|^2_{\ell^2(G_1)} + \|\chi_1(\Delta_1)\rho_1(P_1(f))\|^2_{\ell^2(G_1)}$$
$$=\|\chi_1(\Delta_1)\rho_1(P_1(f))\|^2_{\ell^2(G_1)} + W_1(f),$$

and similarly

$$\|\chi_1(\Delta_1)\rho_1(P_1(f))\|^2_{\ell^2(G_1)} + W_1(f)$$
$$= \sum_{q\in\Gamma^1} \|U[q](f)\|^2_{\ell^2(G_1)} + \|\chi_1(\Delta_1)\rho_1(P_1(f))\|^2_{\ell^2(G_1)}$$
$$\leqslant B_1(L^+_1 R^+_1)^2\|f\|^2_{\ell^2(G)}.$$

This yields the starting point for our induction. Now for the inductive step assume the claim holds up until layer $N-1$. Then we have

$$C^-_{N-1}\|f\|^2_{\ell^2(G)} \leqslant \sum_{n=1}^{N-1}\left(\sum_{q\in\Gamma^{n-1}} \|\chi_n(\Delta_n)f_q\|^2_{\ell^2(G_n)}\right) + W_{N-1}(f) \leqslant C^+_{N-1}\|f\|^2_{\ell^2(G)}.$$

using Notation C.2. We note

$$\sum_{n=1}^{N}\left(\sum_{q\in\Gamma^{n-1}}||\chi_n(\Delta_n)\rho_n(P_n(f_q))||^2_{\ell^2(G_n)}\right)+W_N$$

$$=\sum_{n=1}^{N-1}\left(\sum_{q\in\Gamma^{n-1}}||\chi_n(\Delta_n)\rho_n(P_n(f_q))||^2_{\ell^2(G_n)}\right)+\sum_{q\in\Gamma^{N-1}}||\chi_N(\Delta_N)\rho_N(P_N(f_q))||^2_{\ell^2(G_N)}$$

$$+\sum_{q\in\Gamma^N}||f_q||^2_{\ell^2(G_N)}.$$

Every path $\widetilde{q}\in\Gamma^N$ may be written as $q=(\gamma_n,q)$, for some $\gamma_n\in\Gamma_n$ and $q\in\Gamma^{N-1}$. Thus we have

$$\sum_{q\in\Gamma^N}||f_q||^2_{\ell^2(G_N)}=\sum_{q\in\Gamma^{N-1}}\sum_{\gamma_N\in\Gamma_N}||g_{\gamma_N}(\Delta_N)P_N(\rho_N(f_q))||^2_{\ell^2(G_N)}$$

Inserting this in the above equation yields

$$\sum_{n=1}^{N}\left(\sum_{q\in\Gamma^{n-1}}||\chi_n(\Delta_n)\rho_n(P_n(f_q))||^2_{\ell^2(G_n)}\right)+W_N$$

$$=\sum_{n=1}^{N-1}\left(\sum_{q\in\Gamma^{n-1}}||\chi_n(\Delta_n)\rho_n(P_n(f_q))||^2_{\ell^2(G_n)}\right)$$

$$+\sum_{q\in\Gamma^{N-1}}\underbrace{\left(||\chi_N(\Delta_N)\rho_N(P_N(f_q))||^2_{\ell^2(G_{n-1})}+\sum_{\gamma_n\in\Gamma_N}||g_{\gamma_N}(\Delta_N)P_N(\rho_N(f_q)||^2_{\ell^2(G_N)}\right)}_{=:\beta(f_q)}.$$

We have

$$(L_N^-R_N^-)^2A_N||f_q||^2_{\ell^2(G_{n-1})}\leqslant\beta(f_q)\leqslant(L_N^+R_N^+)^2B_N||f_q||^2_{\ell^2(G_{n-1})},$$

by the operator frame property. With this we find:

$$\min\{1,(L_N^-R_N^-)^2A_N\}\left(\sum_{n=1}^{N-1}\left(\sum_{q\in\Gamma^{n-1}}||\chi_n(\Delta_n)\rho_n(P_n(f_q))||^2_{\ell^2(G_n)}\right)+W_{N-1}\right)$$

$$\leqslant\sum_{n=1}^{N}\sum_{q\in\Gamma^{n-1}}||\chi_n(\Delta_n)\rho_n(P_n(f_q))||^2_{\ell^2(G_n)}+W_N$$

$$\leqslant\max\{1,(L_N^-R_N^-)^2B_N\}\left(\sum_{n=1}^{N-1}\left(\sum_{q\in\Gamma^n}||\chi_n(\Delta_n)U[q](f)||^2_{\ell^2(G_n)}\right)+W_{N-1}\right),$$

after unravelling the definition

$$W_{N-1}(f)\equiv\sum_{q\in\Gamma^N}||f_q||^2_{\ell^2(G_{n-1})}.$$

The induction hypothesis together with the definition of $C_N^\pm$ now yields the claim.

$\square$

With this we now prove our main theorem concerning energy decay.

**Theorem H.3.** Let $\Phi_\infty$ be a generalized graph scattering transform based on a an operator sequence $\mathscr{D}_\infty=(P_n,\Delta_n)_{n=1}^{\infty}$ and a module sequence $\Omega_\infty$ with each $\rho_n(\cdot)\geqslant 0$. Assume in each layer $n\geqslant 1$ that there is an eigenvector $\psi_n$ of $\Delta_n$ with solely positive entries; denote the smallest entry by $m_n:=\min_{i\in G_n}\psi_n[i]$. Denote the eigenvalue corresponding to $\psi_n$ by $\lambda_n$. Quantify the

'spectral-gap' opened up at this eigenvalue through neglecting the output-generating function by $\eta_n := \sum_{\gamma_n \in \Gamma_n} |g_{\gamma_n}(\lambda_n)|^2$ and assume $B_n m_n \geqslant \eta_n$. We then have

$$W_N(f) \leqslant C_N^+ \cdot \left[ \prod_{n=1}^N \left( 1 - \left( m_n^2 - \frac{\eta_n}{B_n} \right) \right) \right] \cdot \|f\|_{\ell^2(G)}^2.$$

*Proof.* Denote the spectral projection (i.e. the orthogonal projection projecting to the space of eigenvectors) onto the eigenspace corresponding to $\lambda_n$ by $P_c^n$.

Then we have

$$W_N(f) = \sum_{q \in \Gamma^{N-1}} \sum_{\gamma_N \in \Gamma_N} \|g_{\gamma_N}(\Delta_N) \rho_N(P_N(f_q))\|_{\ell^2(G_N)}^2$$

$$= \sum_{q \in \Gamma^{N-1}} \sum_{\gamma_N \in \Gamma_N} \|g_{\gamma_N}(\Delta_N)(1 - P_c^N)\rho_N(P_N(f_q))\|_{\ell^2(G_N)}^2$$

$$+ \sum_{q \in \Gamma^{N-1}} \sum_{\gamma_N \in \Gamma_N} \|g_{\gamma_N}(\Delta_N)P_c^N \rho_N(P_N(f_q))\|_{\ell^2(G_N)}^2$$

$$\leqslant \sum_{q \in \Gamma^{N-1}} B_N \|(1 - P_c^N)\rho_N(P_N(f_q))\|_{\ell^2(G_N)}^2$$

$$+ \sum_{q \in \Gamma^{N-1}} \eta_N \|P_c^N \rho_N(P_N(f_q))\|_{\ell^2(G_N)}^2$$

$$\leqslant \sum_{q \in \Gamma^{N-1}} B_N \|(1 - P_c^N)\rho_N(P_N(f_q))\|_{\ell^2(G_N)}^2$$

$$+ \sum_{q \in \Gamma^{N-1}} \eta_N \|\rho_N(P_N(f_q))\|_{\ell^2(G_N)}^2.$$

By orthogonality of the spectral projection, we then have

$$\|(1 - P_c^N)\rho_N(P_n(f_q))\|_{\ell^2(G_N)}^2 = \|\rho_N(P_N(f_q))\|_{\ell^2(G_N)}^2 - \|P_c^N \rho_N(P_n(f_q))\|_{\ell^2(G_N)}^2.$$

Furthermore, we have

$$|\langle \psi^N, \rho_N(P_n(f_q)) \rangle_{\ell^2(G_N)}|^2 \leqslant \|P_c^N \rho_N(P_n(f_q))\|_{\ell^2(G_N)}^2$$

with equality if the multiplicity of $\lambda^N$ is exactly one. With this we find

$$\|(1 - P_c^N)\rho_N(P_N(f_q))\|_{\ell^2(G_N)}^2 = \|\rho_N(P_N(f_q))\|_{\ell^2(G_N)}^2 - \|P_c^N \rho_N(P_N(f_q))\|_{\ell^2(G_N)}^2$$

$$\leqslant \|\rho_N(P_N(f_q))\|_{\ell^2(G_N)}^2 - |\langle \psi^N, \rho_N(P_N(f_q)) \rangle_{\ell^2(G_N)}|^2.$$

Since the image of $\rho_N$ is contained in $\mathbb{R}_+$ by assumption, we have

$$|\langle \psi^N, \rho_N(P_N(f_q)) \rangle_{\ell^2(G_N)}|^2 = \left| \sum_{i=1}^{|G_N|} \rho_N(P_N(f_q))_i (\psi_N)_i \mu_i \right|^2$$

$$\geqslant \left| \sum_{i=1}^{|G_N|} |\rho_N(P_N(f_q))_i| \mu_i \right|^2 \cdot m_N^2$$

$$\geqslant \left| \sum_{i=1}^{|G_N|} |\rho_N(P_N(f_q))_i|^2 \mu_i^2 \right| \cdot m_N^2$$

$$\geqslant \left| \sum_{i=1}^{|G_N|} |\rho_N(P_N(f_q))_i|^2 \mu_i \right| \cdot m_N^2$$

$$\geqslant \|\rho_N(P_N(f_q))\|_{\ell^2(G_N)}^2 \cdot m_N^2$$

Here the second to last inequality follows since in any finite dimensional vector space, the 1-norm is larger than the 2-norm ($||f||_1 \geqslant ||f||_2$) and all weights are assumed to satisfy $\mu_i \geqslant 1$. Thus we now know

$$||(1 - P_c^N)\rho_N(P_N(f_q))||_{\ell^2(G_N)}^2 \leqslant (1 - m_N^2) \, ||\rho_N(P_N(f_q))||_{\ell^2(G_N)}^2.$$

Inserting this in our estimate for $W_N(f)$ we find

$$W_N(f) \leqslant \left(1 - \left(m_N^2 - \frac{\eta_n}{B_n}\right)\right) L_N^+ R_N^+ B_N \cdot W_{N-1}(f)$$

$$\leqslant C_N^+ \prod_{n=1}^{N} \left(1 - \left(m_N^2 - \frac{\eta_n}{B_n}\right)\right) ||f||_{\ell^2(G)}^2.$$

$\square$

Taking $N$ to infinity, we know that $C_N^+$ converges to something larger than zero by assumption. For products of the form $\prod_{n=0}^{N} (1 - q_n)$ with $0 \leqslant q_n < 1$ it is a standard exercise to prove that the limit is non-zero precisely if the sum over the $q_n$ converges. Combining the above result with Lemma H.2, we obtain as an immediate Corollary:

**Corollary H.4.** In the setting of Theorem 4.6, the generalized scattering transform satisfies $\Phi_\infty^{-1}(0) = \{0\}$ if $C_N^\pm \to C^\pm$ for some positive constants $C^\pm$ and $\sum_{n=1}^{N}(m_n - \eta_n/B_n) \to \infty$ as $N \to \infty$.

# I  Stability of Graph Level Feature Aggregation

## I.1  General non-linear feature aggregation:

Our main stability theorem for non-linear feature aggregation is as follows:

**Theorem I.1.** We have

$$\|\Psi_N(f) - \Psi_N(g)\|_{\mathscr{R}_N} \leqslant \left(1 + \sum_{n=1}^{N} \max\{[B_n - 1], [B_n(L_n^+ R_n^+)^2 - 1], 0\} \prod_{k=1}^{n-1} B_k\right)^{\frac{1}{2}} \|f - h\|_{\ell^2(G)}.$$

With the conditions and notation of Theorem 4.2 we have

$$\|\Psi_N(f) - \widetilde{\Psi}_N(f)\|_{\mathscr{R}_N} \leqslant \sqrt{2(2^N - 1)} \cdot \sqrt{(\max\{B, 1/2\})^{N-1}} \cdot D \cdot \delta \cdot \|f\|_{\ell^2(G)}.$$

Additionally, in the setting of Theorem 4.5, assuming that for each $n \leqslant N$ the identification operator $J_n$ satisfies $\left|\|J_n f\|_{\ell^1(\widetilde{G}_n)}/\sqrt{\mu_{\widetilde{G}_n}} - \|f\|_{\ell^1(G_n)}/\sqrt{\mu_{G_n}}\right|, \left|\|J_n f\|_{\ell^k(\widetilde{G}_n)} - \|f\|_{\ell^k(G_n)}\right| \leqslant \delta \cdot K \cdot \|f\|_{\ell^2(G_n)}$ ($2 \leqslant k \leqslant p_n$) implies ($\forall f \in \ell^2(G)$)

$$\|\widetilde{\Psi}_N(J_0 f) - \Psi_N(f)\|_{\mathscr{R}_N} \leqslant \sqrt{2} \cdot \sqrt{K_N^2 \cdot + K^2} \cdot \delta \cdot \|f\|_{\ell^2(G)}.$$

Furhermore, under the assumptions of Corollary H.4 $\Psi_\infty(f) = 0$ implies $f = 0$.

*Proof.* Let $f, h \in \ell^2(G)$. To prove the first two claims, it suffices to prove

$$\|\Psi_N(f) - \Psi_N(h)\|_{\mathscr{R}_N} \leqslant \|\Phi_N(f) - \Phi_N(h)\|_{\mathscr{F}_N},$$

and

$$\|\Psi_N(f) - \widetilde{\Psi}_N(f)\|_{\mathscr{R}_N} \leqslant \|\Phi_N(f) - \widetilde{\Phi}_N(f)\|_{\mathscr{F}_N}.$$

Both statements follow immediately, as soon as we have proved

$$\|N_p^G(f) - N_p^G(h)\|_{\mathbb{R}^p} \leqslant \|f - h\|_{\ell^2(G)}$$

for arbitrary choices of $p$ and $G$. To this end we note that for $p \geqslant 2$ we have $\|f\|_{\ell^p(G)} \leqslant \|f\|_{\ell^2(G)}$ by the monotonicity of $p$-norms, while we have $\|f\|_{\ell^1(G)} \leqslant \sqrt{\mu_G} \cdot \|f\|_{\ell^2(G)}$ by Hölder's inequality. With this we find

$$
\begin{aligned}
\|N_p^G(f) - N_p^G(h)\|_{\mathbb{R}^p}^2 &= \frac{1}{p}\left(\frac{1}{\mu_G}\big|\|f\|_{\ell^1(G)} - \|h\|_{\ell^1(G)}\big|^2 + \sum_{i=2}^{p}\big|\|f\|_{\ell^i(G)} - \|h\|_{\ell^i(G)}\big|^2\right) \\
&\leqslant \frac{1}{p}\left(\frac{1}{\mu_G}\|f-h\|_{\ell^1(G)}^2 + \sum_{i=2}^{p}\|f-h\|_{\ell^i(G)}^2\right) \\
&\leqslant \frac{1}{p}\cdot p\cdot\|f-h\|_{\ell^2(G)}^2 \\
&= \|f-h\|_{\ell^2(G)}.
\end{aligned}
$$

where we have employed the reverse triangle inequality in the first step.

To prove the second claim, we note that we have

$$
\begin{aligned}
&\|\Psi_N(f) - \widetilde{\Psi}_N(J_0 f)\|_{\mathscr{R}_N}^2 \\
&= \sum_{n=1}^{N}\left(\sum_{q\in\Gamma^{n-1}}\|N_{p_n}^{G_n}(\underbrace{\chi_n(\Delta_n)\rho_n(P_n((f_q)))}_{=:x_q}) - N_{p_n}^{\widetilde{G}_n}(\underbrace{\chi_n(\widetilde{\Delta}_n)\rho_n(P_n(\widetilde{f}_q))}_{=:\widetilde{x}_q})\|_{\mathbb{R}^{p_n}}^2\right) \\
&\leqslant 2\sum_{n=1}^{N}\left(\sum_{q\in\Gamma^{n-1}}\|N_{p_n}^{\widetilde{G}_n}(J_n x_q) - N_{p_n}^{\widetilde{G}_n}(\widetilde{x}_q)\|_{\mathbb{R}^{p_n}}\right) \\
&\quad + 2\sum_{n=1}^{N}\left(\sum_{q\in\Gamma^{n-1}}\|N_{p_n}^{\widetilde{G}_n}(J_n x_q) - N_{p_n}^{G_n}(x_q)\|_{\mathbb{R}^{p_n}}\right) \\
&= 2\|\mathscr{J}\Phi_N(f) - \widetilde{\Phi}_N(J_0 f)\|_{\mathscr{F}_N}^2 \\
&\quad + 2\sum_{n=1}^{N}\left(\sum_{q\in\Gamma^{n-1}}\|N_{p_n}^{\widetilde{G}_n}(J_n x_q) - N_{p_n}^{G_n}(x_q)\|_{\mathbb{R}^{p_n}}\right).
\end{aligned}
$$

Thus it remains to bound the last expression. We have

$$
\begin{aligned}
&\|N_p^{\widetilde{G}_n}(J_n x_q) - N_{p_n}^{G_n}(x_q)\|_{\mathbb{R}^{p_n}} \\
&= \frac{1}{p_n}\left(\left|\frac{1}{\sqrt{\mu_{G_n}}}\|f\|_{\ell^1(G)} - \frac{1}{\sqrt{\mu_{\widetilde{G}_n}}}\|J_n f\|_{\ell^1(\widetilde{G})}\right|^2 + \sum_{i=2}^{p_n}\big|\|f\|_{\ell^i(G)} - \|J_n f\|_{\ell^i(\widetilde{G})}\big|^2\right) \\
&\leqslant K^2\cdot\delta^2\cdot\|x_q\|_{\ell^2(G_n)}^2.
\end{aligned}
$$

By our results of Appendix C and since we assume admissibility, we have

$$
\sum_{n=1}^{N}\sum_{q\in\Gamma^{n-1}}\|x_q\|_{\ell^2(G_n)}^2 \leqslant \|f\|_{\ell^2(G)}^2.
$$

Thus in total

$$
\|\Psi_N(f) - \widetilde{\Psi}_N(J_0 f)\|_{\mathscr{F}_N}^2 \leqslant 2\|\mathscr{J}\Phi_N(f) - \widetilde{\Phi}_N(J_0 f)\|_{\mathscr{F}_N}^2 + 2K\delta\|f\|_{\ell^2(G)},
$$

from which our stability claim follows.

It remains to prove that the assumptions of Corollary H.4 $\Psi_\infty(f) = 0$ imply $f = 0$. But since $N_p^G(f) = 0$ implies $f = 0$, this is clear. $\qquad\square$

## I.2  Low-Pass feature Aggregation

The main assumption we have in this section is that each operator $\Delta_n$ (and $\widetilde{\Delta}_n$) has a simple lowest lying eigenvalue equal to zero. We denote the associated eigenvector (determined up to a complex phase) by $\psi_{\Delta_n}$ and the associated spectral projection to the lowest lying eigenvalue by $P_{\Delta_n}$. It acts as

$$P_{\Delta_n} f \equiv \psi_{\Delta_n} \langle \psi_{\Delta_n}, f \rangle_{\ell^2(G_n)}.$$

Now we are ready to state our main stability result under these circumstances:

**Theorem I.2.** We have

$$\|\Psi_N^{|\langle \cdot, \cdot \rangle|}(f) - \Psi_N^{|\langle \cdot, \cdot \rangle|}(g)\|_{\mathscr{C}_N} \leqslant \left( 1 + \sum_{n=1}^{N} \max\{[B_n - 1], [B_n (L_n^+ R_n^+)^2 - 1], 0\} \prod_{k=1}^{n-1} B_k \right)^{\frac{1}{2}} \|f - h\|_{\ell^2(G)}.$$

With the conditions and notation of Theorem 4.2 and under the additional assumption $\|(P_{\Delta_n} - P_{\widetilde{\Delta}_n})\|_{op} \leqslant K \cdot \delta$ for $n \leqslant N$ and some $K \geqslant 0$, we have

$$\|\Psi_N^{|\langle \cdot, \cdot \rangle|}(f) - \widetilde{\Psi}_N^{|\langle \cdot, \cdot \rangle|}(f)\|_{\mathscr{C}_N} \leqslant \sqrt{2} \cdot \sqrt{2(2^N - 1)(\max\{B, 1/2\})^{N-1} + K^2} \cdot \delta \cdot \|f\|_{\ell^2(G)}.$$

In the setting of Theorem 4.5 and under the additional assumption $\|P_{\Delta_n} f\|_{\ell^2(G_n)} - \|P_{\widetilde{\Delta}_n} J_n f\|_{\ell^2(\widetilde{G}_n)}| \leqslant K\delta \|f\|_{\ell^2(G_n)}$ for all $f \in \ell^2(G_n)$ ($n \leqslant N$), we have

$$\|\widetilde{\Psi}_N^{|\langle \cdot, \cdot \rangle|}(J_0 f) - \Psi_N^{|\langle \cdot, \cdot \rangle|}(f)\|_{\mathscr{C}_N} \leqslant \sqrt{2} \cdot \sqrt{K_N^2 \cdot + K^2} \cdot \delta \cdot \|f\|_{\ell^2(G)}.$$

*Proof.* Let $f, h \in \ell^2(G)$. To prove the first claim, it suffices to prove

$$\|\Psi_N^{|\langle \cdot, \cdot \rangle|}(f) - \Psi_N^{|\langle \cdot, \cdot \rangle|}(h)\|_{\mathscr{C}_N} \leqslant \|\Phi_N(f) - \Phi_N(h)\|_{\mathscr{F}_N}.$$

This immediately follows from the fact that for all $f \in \ell^2(G_n)$

$$|\langle \psi_{\Delta_n}, f \rangle_{\ell^2(G_n)}|^2 \leqslant \|\psi_{\Delta_n}\|_{\ell^2(G_n)}^2 \cdot \|f\|_{\ell^2(G_n)}^2$$

by Hölder's inequality.

The next claim we want to prove is that we have for all $f \in \ell^2(G)$

$$\|\Psi_N^{|\langle \cdot, \cdot \rangle|}(f) - \widetilde{\Psi}_N^{|\langle \cdot, \cdot \rangle|}(f)\|_{\mathscr{C}_N} \leqslant \sqrt{2} \cdot \sqrt{2(2^N - 1) + K^2} \cdot \delta \cdot \|f\|_{\ell^2(G)}.$$

We note

$$\|\Psi_N^{|\langle \cdot, \cdot \rangle|}(f) - \widetilde{\Psi}_N^{|\langle \cdot, \cdot \rangle|}(f)\|_{\mathscr{C}_N}^2$$

$$= \sum_{n=1}^{N} \left( \sum_{q \in \Gamma^{n-1}} \left| |\langle \psi_{\Delta_n}, \underbrace{\chi_n(\Delta_n) \rho_n(P_n(f_q))}_{=:x_q} \rangle_{\ell^2(G_n)}| - |\langle \psi_{\widetilde{\Delta}_n}, \underbrace{\chi_n(\widetilde{\Delta}_n) \rho_n(P_n(\widetilde{f}_q))}_{\widetilde{x}_q} \rangle_{\ell^2(G_n)}| \right|^2 \right)$$

$$= \sum_{n=1}^{N} \left( \sum_{q \in \Gamma^{n-1}} \left| \|P_{\Delta_n} x_q\|_{\ell^2(G_n)} - \|P_{\widetilde{\Delta}_n} \widetilde{x}_q\|_{\ell^2(G_n)} \right|^2 \right)$$

$$\leqslant \sum_{n=1}^{N} \left( \sum_{q \in \Gamma^{n-1}} \|P_{\Delta_n} x_q - P_{\widetilde{\Delta}_n} \widetilde{x}_q\|_{\ell^2(G_n)}^2 \right)$$

$$\leqslant 2 \sum_{n=1}^{N} \left( \sum_{q \in \Gamma^{n-1}} \|P_{\widetilde{\Delta}_n}(x_q - \widetilde{x}_q)\|_{\ell^2(G_n)}^2 \right) + 2 \sum_{n=1}^{N} \left( \sum_{q \in \Gamma^{n-1}} \|(P_{\Delta_n} - P_{\widetilde{\Delta}_n}) x_q\|_{\ell^2(G_n)}^2 \right)$$

$$\leqslant 2 \|\Phi_N(f) - \Phi_N(h)\|_{\mathscr{F}_N}^2 + 2 \sum_{n=1}^{N} \left( \sum_{q \in \Gamma^{n-1}} \|(P_{\Delta_n} - P_{\widetilde{\Delta}_n}) x_q\|_{\ell^2(G_n)}^2 \right)$$

Hence we need to bound the expression "$\|(P_{\Delta_n} - P_{\tilde{\Delta}_n})x_q\|^2_{\ell^2(G_n)}$". We note

$$\|(P_{\Delta_n} - P_{\tilde{\Delta}_n})x_q\|^2_{\ell^2(G_n)} \leqslant \|(P_{\Delta_n} - P_{\tilde{\Delta}_n})\|_{op} \cdot \|x_q\|^2_{\ell^2(G_n)}$$
$$\leqslant K^2 \cdot \delta^2 \cdot \|x_q\|^2_{\ell^2(G_n)}$$

and thus

$$\|\Psi_N^{|\langle \cdot,\cdot\rangle|}(f) - \tilde{\Psi}_N^{|\langle \cdot,\cdot\rangle|}(f)\|^2_{\mathscr{C}_N}$$

$$\leqslant 2\|\Phi_N(f) - \Phi_N(h)\|^2_{\mathscr{F}_N} + 2K^2 \cdot \delta^2 \cdot \sum_{n=1}^{N}\left(\sum_{q\in\Gamma^{n-1}} \|\chi_n(\Delta_n)\rho_n(P_n((f_q)))\|^2_{\ell^2(G_n)}\right)$$

$$\leqslant 2\|\Phi_N(f) - \Phi_N(h)\|^2_{\mathscr{F}_N} + 2K^2 \cdot \delta^2 \cdot \|f\|^2_{\ell^2(G)}$$

and the claim follows.

Finally we want to prove

$$\|\tilde{\Psi}_N^{|\langle \cdot,\cdot\rangle|}(J_0 f) - \Psi_N^{|\langle \cdot,\cdot\rangle|}(f)\|_{\mathscr{C}_N} \leqslant \sqrt{2}\cdot\sqrt{K_N^2 \cdot + K^2}\cdot\delta\cdot\|f\|_{\ell^2(G)}.$$

We note

$$\|\Psi_N^{|\langle \cdot,\cdot\rangle|^2}(f) - \tilde{\Psi}_N^{|\langle \cdot,\cdot\rangle|}(f)\|_{\mathscr{C}_N}$$

$$= \sum_{n=1}^{N}\left(\sum_{q\in\Gamma^{n-1}}\left||\langle\psi_{\Delta_n}, \underbrace{\chi_n(\Delta_n)\rho_n(P_n((f_q)))}_{=:x_q}\rangle_{\ell^2(G_n)}| - |\langle\psi_{\tilde{\Delta}_n}, \underbrace{\chi_n(\tilde{\Delta}_n)\rho_n(P_n(\tilde{f}_q))}_{\tilde{x}_q}\rangle_{\ell^2(\tilde{G}_n)}|\right|^2\right)$$

$$= \sum_{n=1}^{N}\left(\sum_{q\in\Gamma^{n-1}}\left|\|P_{\Delta_n}x_q\|_{\ell^2(G_n)} - \|P_{\tilde{\Delta}_n}\tilde{x}_q\|_{\ell^2(\tilde{G}_n)}\right|^2\right)$$

$$\leqslant \sum_{n=1}^{N}\left(\sum_{q\in\Gamma^{n-1}}\left|\|P_{\Delta_n}x_q\|_{\ell^2(G_n)} - \|P_{\tilde{\Delta}_n}J_n x_q\|_{\ell^2(\tilde{G}_n)} + \|P_{\tilde{\Delta}_n}J_n x_q\|_{\ell^2(\tilde{G}_n)} - \|P_{\tilde{\Delta}_n}\tilde{x}_q\|_{\ell^2(\tilde{G}_n)}\right|^2\right)$$

$$\leqslant 2\sum_{n=1}^{N}\left(\sum_{q\in\Gamma^{n-1}}\left|\|P_{\tilde{\Delta}_n}J_n x_q\|_{\ell^2(\tilde{G}_n)} - \|P_{\tilde{\Delta}_n}\tilde{x}_q\|_{\ell^2(\tilde{G}_n)}\right|^2\right)$$

$$+2\sum_{n=1}^{N}\left(\sum_{q\in\Gamma^{n-1}}\left|\|P_{\Delta_n}x_q\|_{\ell^2(G_n)} - \|P_{\tilde{\Delta}_n}J_n x_q\|_{\ell^2(\tilde{G}_n)}\right|^2\right)$$

$$\leqslant 2\sum_{n=1}^{N}\left(\sum_{q\in\Gamma^{n-1}}\left|\|P_{\tilde{\Delta}_n}J_n x_q\|_{\ell^2(\tilde{G}_n)} - \|P_{\tilde{\Delta}_n}\tilde{x}_q\|_{\ell^2(\tilde{G}_n)}\right|^2\right)$$

$$+2\sum_{n=1}^{N}\left(\sum_{q\in\Gamma^{n-1}}\left|\|P_{\Delta_n}x_q\|_{\ell^2(G_n)} - \|P_{\tilde{\Delta}_n}J_n x_q\|_{\ell^2(\tilde{G}_n)}\right|^2\right)$$

$$\leqslant 2\|\mathscr{J}\Phi_N(f) - \tilde{\Phi}_N(J_0 f)\|^2_{\widetilde{\mathscr{F}}_N} + 2\sum_{n=1}^{N}\left(\sum_{q\in\Gamma^{n-1}}\left|\|P_{\Delta_n}x_q\|_{\ell^2(G_n)} - \|P_{\tilde{\Delta}_n}J_n x_q\|_{\ell^2(\tilde{G}_n)}\right|^2\right)$$

$$\leqslant 2\|\mathscr{J}\Phi_N(f) - \tilde{\Phi}_N(J_0 f)\|^2_{\widetilde{\mathscr{F}}_N} + 2\sum_{n=1}^{N}\left(\sum_{q\in\Gamma^{n-1}} K^2 \cdot \delta^2 \|x_q\|^2_{\ell^2(G_n)}\right)$$

$$\leqslant 2\|\mathscr{J}\Phi_N(f) - \tilde{\Phi}_N(J_0 f)\|^2_{\widetilde{\mathscr{F}}_N} + 2K^2 \cdot \delta^2\|f\|^2_{\ell^2(G)}.$$

which proves the claim. $\qquad\square$

In establishing triviality of the 'kernel', we have to be a tiny bit more careful:

**Theorem I.3.** In the setting of of Corollary H.4, assume that in each layer $n$, the output generating function $\chi_n$ of the underlying scattering transform satisfies $\chi_n(0) \neq 0$ and $\chi_n(\lambda_i) = 0$ for ordered non-zero eigenvalues $\lambda_2 \leqslant ... \leqslant \lambda_{|G_n|}$ of the operator $\Delta_n$. Then $\Psi_\infty^{|\langle \cdot, \cdot \rangle|}(f) = 0$ implies $f = 0$.

*Proof.* Under these assumptions, we do not lose any information by projecting to $\psi_{\Delta_n}$ in each $\ell^2(G_n)$, since the image of $\chi_n(\Delta_n)$ is already contained in the one-dimensional space generated by the lowest lying eigenvector $\psi_{\Delta_n}$. $\square$

## J   Details on Higher Order Scattering

Node signals capture information about nodes in isolation. However, one might also want to analyse or incorporate information about binary, ternary or even higher order relations between nodes, such as distances or angles between nodes representing atoms in a molecule. This can be formalized by considering tensorial input signals:

**Tensorial input:**   A 2-tensor on a graph $G$, as it was already utilized in Section 6, is simply an element of $\mathbb{C}^{|G| \times |G|}$ or – equivalently – a map from $G \times G$ to $\mathbb{C}$, since it associates a complex number to each element $(g_1, g_2) \in G \times G$. Since $G \times G$ is precisely the set of (possible) edges $E$, we can equivalently think of 2-tensors edge-signals. A 3-tensor an element of $\mathbb{C}^{|G| \times |G| \times |G|}$ or equivalently a map from $G \times G \times G \equiv G^3$ to $\mathbb{C}$. A 4-tensor then is a map from $G^4 \equiv G \times G \times G \times G$ to $\mathbb{C}$ or equivalenlty an element of $\mathbb{C}^{|G| \times |G| \times |G| \times |G|}$ and so forth. Clearly the space of $k$-tensors forms a linear vector space. Addition and scalar multiplication by $\lambda \in \mathbb{C}$ are given by

$$(f + \lambda g)_{i_1,...,i_k} := f_{i_1,...,i_k} + \lambda g_{i_1,...,i_k}$$

with $f$ and $g$ being $k$-tensors. For fixed $k$, we equip the space of $k$-tensors with an inner product according to

$$\langle f, g \rangle = \sum_{i_1,...,i_k=1}^{|G|} \overline{f_{i_1,...,i_k}} g_{i_1,...,i_k} \mu_{i_1,...,i_k}$$

and denote the resulting inner-product space by $\ell^2(G^k)$.

**Operators on Spaces of Tensors:**   Since for fixed $k$ the space $\ell^2(G^k)$ is simply a $|G|^k$-dimensional complex inner product space, there are exist normal operators $\Delta^k : \ell^2(G^k) \to \ell^2(G^k)$ on this space. Note that the $k$ in $\Delta^k$ signifies on which space this operator acts. It does not signify that an operator is raised to the $k^{th}$ power. Setting for example node-weights $\mu_i$ and edge weights $\mu_{ik}$ to one, the adjacency matrix $W$ as well as normalized or un-normalized graph Laplacians constitute self-adjoint operators on $\ell^2(G^2)$, where they act by matrix multiplication.

**Higher order Scattering Transforms:**   We can then follow the recipe laid out Section 3 in constructing $k^{th}$-order scattering transforms; all that we need are a module sequence $\Omega_N$ and an operator sequence $\mathscr{D}_N^k := (P_n^k, \Delta_n^k)_{n=1}^N$, where now $P_n^k : \ell^2(G_{n-1}^k) \to \ell^2(G_n^k)$ and $\Delta_n^k : \ell^2(G_n^k) \to \ell^2(G_n^k)$.

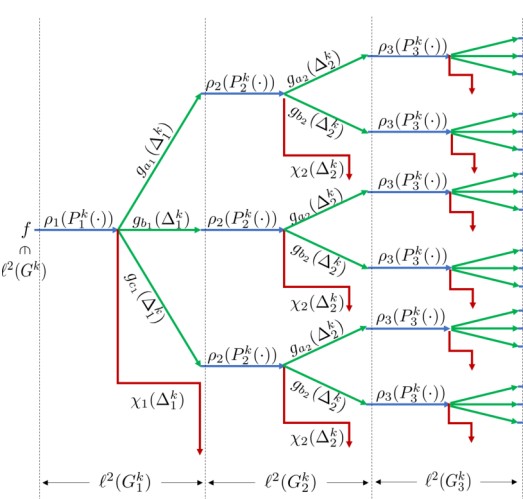

To our initial signal $f \in \ell^2(G^k)$ we first apply the connecting operator $P_1^k$, yielding a signal representation in $\ell^2(G_1^k)$. Subsequently, we apply the pointwise non-linearity $\rho_1$. Then we apply our graph filters $\{\chi_1(\Delta_1^k)\} \bigcup \{g_{\gamma_1}(\Delta_1^k)\}_{\gamma_1 \in \Gamma_1}$ to $\rho_1(P_1^k(f))$ yielding the output $V_1(f) := \chi_1(\Delta_1^k)\rho_1(P_1^k(f))$ as well as the intermediate hidden representations $\{U_1[\gamma_1](f) := g_{\gamma_1}(\Delta_1^k)\rho_1(P_1^k(f))\}_{\gamma_1 \in \Gamma_1}$ obtained in the first layer. Here we have introduced the **one-step scattering propagator** $U_n[\gamma_n] : \ell^2(G_{n-1}^k) \to \ell^2(G_n^k)$ mapping $f \mapsto g_{\gamma_n}(\Delta_n)\rho_n(P_n(f))$ as well as the **output generating operator** $V_n : \ell^2(G_{n-1}^k) \to \ell^2(G_n^k)$ mapping $f$ to $\chi_n(\Delta_n^k)\rho_n(P_n^k(f))$. Upon defining the **set** $\Gamma^{N-1} := \Gamma_{N-1} \times ... \times \Gamma_1$ **of paths** of length $(N-1)$ terminating in layer $N-1$ (with $\Gamma^0$ taken to be the one-element set) and iterating the above procedure, we see that the outputs generated in the $N^{\text{th}}$-layer are indexed by paths $\Gamma^{N-1}$ terminating in the previous layer.

Figure 7: Schematic Higher Order Scattering Architecture

We denote the resulting feature map by $\Phi_N^k$ and write $\mathscr{F}_N^k$ for the corresponding feature space. The node-level stability results of the preceding sections then readily translate to higher order scattering transforms.

As the respective proofs are identical to the corresponding results for the node setting, we do not repeat them here.

**Theorem J.1.** With the notation of Section 4, we have for all $f, h \in \ell^2(G^k)$:

$$\|\Phi_N^k(f) - \Phi_N^k(h)\|_{\mathscr{F}_N^k}^2 \leqslant \left(1 + \sum_{n=1}^N \max\{[B_n - 1], [B_n(L_n^+ R_n^+)^2 - 1], 0\} \prod_{\ell=1}^{n-1} B_\ell\right) \|f - h\|_{\ell^2(G^k)}^2$$

**Theorem J.2.** Let $\Phi_N$ and $\widetilde{\Phi}_N$ be two scattering transforms based on the same module sequence $\Omega_N$ and operator sequences $\mathscr{D}_N^k, \widetilde{\mathscr{D}}_N^k$ with the same connecting operators $(P_n^k = \widetilde{P}_n^k)$ in each layer. Assume $R_n^+, L_n^+ \leqslant 1$ and $B_n \leqslant B$ for some $B$ and $n \leqslant N$. Assume that the respective normal operators satisfy $\|\Delta_n^k - \widetilde{\Delta}_n^k\|_F \leqslant \delta$ for some $\delta > 0$. Further assume that the functions $\{g_{\gamma_n}\}_{\gamma_n \in \Gamma_n}$ and $\chi_n$ in each layer are Lipschitz continuous with associated Lipschitz constants satisfying $L_{\chi_n}^2 + \sum_{\gamma_n \in \Gamma_n} L_{g_{\gamma_n}}^2 \leqslant D^2$ for all $n \leqslant N$ and some $D > 0$. Then we have for all $f \in \ell^2(G^k)$

$$\|\widetilde{\Phi}_N^k(f) - \Phi_N^k(f)\|_{\mathscr{F}_N} \leqslant \sqrt{2(2^N - 1)} \cdot \sqrt{(\max\{B, 1/2\})^{N-1}} \cdot D \cdot \delta \cdot \|f\|_{\ell^2(G^k)}$$

**Theorem J.3.** Let $\Phi_N^k, \widetilde{\Phi}_N^k$ be higher order scattering transforms based on a common module sequence $\Omega_N$ and differing operator sequences $\mathscr{D}_N^k, \widetilde{\mathscr{D}}_N^k$. Assume $R_n^+, L_n^+ \leqslant 1$ and $B_n \leqslant B$ for some $B$ and $n \geqslant 0$. Assume that there are identification operators $J_n : \ell^2(G_n^k) \to \ell^2(\widetilde{G}_n^k)$, $\widetilde{J}_n : \ell^2(\widetilde{G}_n^k) \to \ell^2(G_n^k)$ $(0 \leqslant n \leqslant N)$ so that the respective signal spaces are $\delta$-unitarily equivalent, the respective normal operators $\Delta_n^k, \widetilde{\Delta}_n^k$ are $\omega$-$\delta$-close as well as bounded (in norm) by $K > 0$ and the connecting operators satisfy $\|\widetilde{P}_n^k J_{n-1} f - J_n P_n^k f\|_{\ell^2(\widetilde{G}_n^k)} \leqslant \delta \|f\|_{\ell^2(G_{n-1}^k)}$. For the common module sequence $\Omega_N$ assume that the non-linearities satisfy $\|\rho_n(J_n f) - J_n \rho_n(f)\|_{\ell^2(\widetilde{G}_n^k)} \leqslant \delta \|f\|_{\ell^2(G_n^k)}$ and that the constants $C_{\chi_n}$ and $\{C_{g_{\gamma_n}}\}_{\gamma_n \in \Gamma_N}$ associated through Lemma 4.4 to the functions of the generalized frames in each layer satisfy $C_{\chi_n}^2 + \sum_{\gamma_n \in \Gamma_N} C_{g_{\gamma_n}}^2 \leqslant D^2$ for some $D > 0$. Denote the operator that the family $\{J_n\}_n$ of identification operators induce on $\mathscr{F}_N^k$ through concatenation by $\mathscr{J}_N : \mathscr{F}_N^k \to \widetilde{\mathscr{F}}_N^k$. Then we have with $K_N = \sqrt{(8^N - 1)(2D^2 + 12B)/7} \cdot B^{N-1}$ if $B > 1/8$ and

$K_N = \sqrt{(2D^2 + 12B) \cdot (1 - B^N)/(1 - B)}$ if $B \leqslant 1/8$ that

$$\|\widetilde{\Phi}_N^k(J_0 f) - \mathscr{J}_N \Phi_N^k(f)\|_{\widetilde{\mathscr{F}}_N^k} \leqslant K_N \cdot \delta \cdot \|f\|_{\ell^2(G}, \quad \forall f \in \ell^2(G^k).$$

If additionally $\|\widetilde{P}_n^k J_{n-1} f - J_n P_n^k f\|_{\ell^2(\widetilde{G}_n)} = 0$ or $\|\rho_n(J_n f) - J_n \rho_n(f)\|_{\ell^2(\widetilde{G}_n^k)} = 0$ holds in each layer, then we have $K_N = \sqrt{(4^N - 1)(2D^2 + 4B)/3 \cdot B^{N-1}}$ if $B > 1/4$ and $K_N = \sqrt{(2D^2 + 4B) \cdot (1 - B^N)/(1 - B)}$ if $B \leqslant 1/4$. If both additional equations hold, we have $K_N = \sqrt{(2^N - 1)2D^2 \cdot B^{N-1}}$ if $B > 1/2$ and $K_N = \sqrt{2D^2 \cdot (1 - B^N)/(1 - B)}$ if $B \leqslant 1/2$.

The map $N_p^G$ introduced in (4) can also be adapted to aggregate higher-order tensorial features into graph level features: With

$$\|f\|_q := \left( \sum_{i_1,\ldots,i_k \in G} |f_{i_1,\ldots,i_k}|^q \mu_{i_1,\ldots,i_k} \right)^{1/q}$$

and $\mu_{G^k} := \sum_{i_1 \ldots i_k = 1}^{|G|} \mu_{i_1,\ldots,i_k}$, we define

$$N_p^{G^k}(f) = (\|f\|_{\ell^1(G^k)}/\sqrt{\mu_{G^k}}, \|f\|_{\ell^2(G^k)}, \|f\|_{\ell^3(G^k)}, \ldots, \|f\|_{\ell^p(G^k)})^\top / \sqrt{p}.$$

Given a feature map $\Phi_N^k$ with feature space

$$\mathscr{F}_N = \oplus_{n=1}^N \left( \ell^2(G_n^k) \right)^{|\Gamma^{n-1}|},$$

we obtain a corresponding map $\Psi_N^k$ mapping from $\ell^2(G^k)$ to

$$\mathscr{R}_N = \oplus_{n=1}^N \left( \mathbb{R}^{p_n} \right)^{|\Gamma^{n-1}|}$$

by concatenating $\Phi_N^k$ with the map that the family of non-linear maps $\{N_{G_n^k}^{p_n}\}_{n=1}^N$ induces on $\mathscr{F}_N$ by concatenation. The resulting map $\Psi_N^k$ again has stability properties analogous to the node level case:

**Theorem J.4.** Assuming admissibility, we have

$$\|\Psi_N^k(f) - \Psi_N^k(h)\|_{\mathscr{R}_N} \leqslant \left( 1 + \sum_{n=1}^N \max\{[B_n - 1], [B_n (L_n^+ R_n^+)^2 - 1], 0\} \prod_{\ell=1}^{n-1} B_\ell \right) \|f - h\|_{\ell^2(G^k)}^2$$

for all $f, h \in \ell^2(G)$. With the conditions and notation of Theorem J.2 we have

$$\|\Psi_N^k(f) - \widetilde{\Psi}_N^k(f)\|_{\mathscr{R}_N} \leqslant \sqrt{2(2^N - 1)} \cdot \sqrt{(\max\{B, 1/2\})^{N-1}} \cdot D \cdot \delta \cdot \|f\|_{\ell^2(G^k)}.$$

Additionally, in the setting of Theorem J.3, assuming that for each $n \leqslant N$ the identification operator $J_n$ satisfies $\left|\|J_n f\|_{\ell^1(\widetilde{G}_n^k)}/\sqrt{\mu_{\widetilde{G}_n^k}} - \|f\|_{\ell^1(G_n^k)}/\sqrt{\mu_{G_n^k}}\right|, \left|\|J_n f\|_{\ell^r(\widetilde{G}_n^k)} - \|f\|_{\ell^r(G_n^k)}\right| \leqslant \delta \cdot K \cdot \|f\|_{\ell^2(G_n^k)}$ for $2 \leqslant r \leqslant p_n$ implies ($\forall f \in \ell^2(G^k)$)

$$\|\widetilde{\Psi}_N(J_0 f) - \Psi_N(f)\|_{\mathscr{R}_N} \leqslant \sqrt{2} \cdot \sqrt{K_N^2 + K^2} \cdot \delta \cdot \|f\|_{\ell^2(G^k)}.$$

As the proofs here are virtually the same as for the corresponding results in previous sections – essentially only replacing $G$ by $G^k$, we omit a repetition of them here.

# K  Additional Details on Experiments

Here we provide additional details on utilized scattering architectures, training procedures, datasets and (performance of) other methods our approach is being compared to. Irrespective of task, our models are trained on an NVIDIA DGX A100 architecture utilizing between two and eight NVIDIA Tesla A100 GPUs with 80GB memory each. Running 10-fold cross validation for the respective experiments took at most 71 hours (which was needed for social network graph classification on REDDIT-12K).

### K.1 Social Network Graph Classification

**Datasets:** The data we are working with is taken from [43]. In particular this work introduced six social network datasets extracted from from scientific collaborations (COLLAB), movie collaborations (IMDB-B, IMDB-M) and Reddit discussion threads (REDDIT-B, REDDIT-5K, REDDIT-12K). Data is anonymised and contains no content that might be considered offensive. Each graph carries a class label, and the goal is to predict this label. Some basic properties of these datasets are listed in Table 3 below.

Table 3: Social Network Dataset Characteristics

| Attributes: | COLLAB | IMDB- B | IMDB-M | REDDIT-B | REDDIT-5K | REDDIT-12K |
|---|---|---|---|---|---|---|
| Graphs | 5K | 1K | 1.5K | 2K | 5K | 12K |
| Nodes | 372.5K | 19.8K | 19.5K | 859.2K | 2.5M | 4.7M |
| Edges | 49.1M | 386.1K | 395.6 | 4M | 11.9M | 21.8M |
| Maximum Degree | 2k | 540 | 352 | 12.2K | 8K | 12.2K |
| Minimum Degree | 4 | 4 | 4 | 4 | 4 | 4 |
| Average Degree | 263 | 39 | 40 | 9 | 9 | 9 |
| Target Labels | 3 | 2 | 3 | 2 | 5 | 11 |
| Disconnected Graphs | No | No | No | Yes | Yes | Yes |

These datasets contain graph structures, however they don't contain associated weights or graph signals. Having unspecified weights simply means that the adjacency matrix $W$ from which we construct the graph Laplacian

$$\mathcal{L} = D - W$$

on which our operator $\Delta$ is based simply has each entry corresponding to an edge set to unity. If no edge is present between vertices $i$ and $j$, the entry $W_{ij}$ is set to zero. It remains to solve the problem of the missing input signals. Our strategy is to generate signals reflecting the geometry of the underlying graph. We do this by utilizing features that associate to each node a number that characterizes its role or importance within its local environment or within the entire graph. We briefly describe them here:

1. **Degree**: The degree of a node is the number of edges incident at this node.

2. **Eccentricity**: For a connected graph, the eccentricity of a node is the maximum distance from this node to all other nodes. On a disconnected graph it is not defined.

3. **Clustering**: For unweighted graphs the clustering $c(u)$ of a node $u$ is the fraction of possible triangles through that node that actually exist. It is calculated as

$$c(u) = \frac{2T(u)}{\deg(u)(\deg(u) - 1)}.$$

4. **Number of triangles**: The number of triangles containing the given node as a vertex.

5. **Core number**: A k-core is a maximal subgraph that only contains nodes of degree k or more. The core number of a node is the largest value k of a k-core containing that node.

6. **Clique number**: A clique is a subset of vertices of an undirected graph such that every two distinct vertices in the clique are adjacent. This input assigns the number of cliques the nodes participates in to each node.

7. **Pagerank**: This returns the PageRank of the respective nodes in the graph. PageRank computes a ranking of the nodes in the graph based on the structure of the edges. Originally it was designed as an algorithm to rank web pages.

For the first three datasets listed in Table 3 we utilize all listed input features. For the latter three datasets we have to refrain from using eccentricity as an input signal, as these datasets contain graphs that have multiple non-connected graph components.

**Scattering Architecture:** We chose a generalized scattering architecture of depth $N = 4$. As normal-operators, we utilize in each layer the un-normalized graph Laplacian $\mathcal{L} = D - W$ scaled by its largest eigenvalue ($\Delta = \mathcal{L}/\lambda_{max}(\mathcal{L})$). Filters are chosen as $\frac{1}{2}(\sin(\pi/2 \cdot \Delta), [\cos(\pi/2 \cdot \Delta) -$

$\psi_\Delta \psi_\Delta^\top], \sin(\pi \cdot \Delta), [\cos(\pi \cdot \Delta) - \psi_\Delta \psi_\Delta^\top])$, which allows to specify the output generating function solely by demanding $\chi(0) = 1$ and $\chi(\lambda) = 0$ on all other eigenvalues of $\Delta$. Here $\psi_\Delta$ is the normalized vector of all ones (satisfying $\Delta \psi_\Delta = 0$). Connecting operators are chosen as the identity, while we set $\rho_{n \geqslant 1}(\cdot) = |\cdot|$. We note that for connected graphs, this recovers Architecture I of Fig. 2. On disconnected graphs (as they can appear in the REDDIT datasets), we however do not account for vectors other than $\psi_\Delta$ in the lowest-lying eigenspace of the graph Laplacian. This scattering architecture is then applied to each of these input signal individually. For each input signal, this returns a feature vector with $1 + 4 + 16 + 64 = 85$ entries. These individual feature vectors are then concatenated into one final composite feature vector for each graph. Concerning applicable theoretical results, we note the following:

**Training Procedure:** We train RBF kernel support vector classifiers on our composite scattering features. We fix $\epsilon = 0.1$. The hyperparameter $\gamma$ scaling the exponent is chosen from

$$G_{\text{pool}} := \{0.00001, 0.0001, 0.001, 0.01, 0.1, 1, 10, 100\},$$

while we pick the $C$ that controls the error our slack variables introduce among

$$C_{\text{pool}} := \{0.001, 0.01, 0.1, 1, 10, 25, 50, 100, 1000\}.$$

We chose these parameters in agreement with the choices of [12] to facilitate comparison between the two works.

We could simply implement the training of the RBF-classifier on our composite scattering features by dividing each social-network dataset into 10 folds, then iteratively choosing one fold for testing and among the remaining 9 folds randomly choosing one for validation (i.e. for tuning the hyperparameters). Instead, following [12] (whose code is released under an Apache license and on which we partially built), we take a slightly different approach: We still randomly split our dataset into 10 folds. Among the 10 folds, we iteratively pick one for testing. Say we have picked the $n^{th}$ fold for testing. Then there are 9 remaining folds. We iteratively pick the $m_n^{th}$ (with $1 \leqslant m_n \leqslant 9$) of the remaining 9 folds for choosing hyperparameters. This leaves 8 folds on which we train our model for each choice of hyper parameter in $C_{\text{pool}} \times G_{\text{pool}}$. The resulting classifiers are all evaluated on the $m_n^{th}$ fold. The one that performs best is retained as classifier $m_n$. As $m_n$ varies between 1 and 9 (still for fixed $n$), this yields a set $\{f_{m_n} : 1 \leqslant m_n \leqslant 9\}$ of nine classifiers. From these we build the classifier $f_n$, whose classification result is obtained from a majority vote among the nine classifiers in $\{f_{m_n} : 1 \leqslant m_n \leqslant 9\}$. Then we evaluate the performance of $f_n$ on the $n^{th}$ fold to obtain the $n^{th}$ estimation of how well our model performs. As $n$ varies from one to ten, we built the mean and variance of the performances of the classifiers $f_n$ on the $n^{th}$ fold expressed as the percentage of correct classifications.

**Reference Methods:** To allow for a comparison of our results to the literature, typical classification accuracies for graph algorithms on social network datasets are displayed in Table 1. Following the standard format of reporting classification accuracies, they are presented in the format (Accuracy $\pm$ standard deviation). If results are not reported for a dataset, we denote this as not available (N/A). The first three rows of Table 1 display results for graph kernel methods; namely Weisfeiler-Lehman graph kernels (WL, [33]), Graphlet kernels (Graphlet, [34]) and deep graph kernels (DGK, [42]). The subsequent rows display results for geometric deep learning algorithms: Deep graph convolutional neural networks (DGCNN,[46]), Patchy-san (PSCN (with k=10), [26]), recurrent neural network autoencoders (S2S-N2N-PP, [16]) and graph isomorphism networks (GIN [41]). These results are taken from [12]. Additionally we compare with P-Poinc [19], which embeds nodes into a hyperbolic space (the Poincare ball, to be precise), GSN-e [3] which combines message passing with structural features extracted via subgraph isomorphism and WKPI-kC [47] which utilizes a weighted kernel within its metric learning framework. The second to last row (GS-SVM [12]) provides a result that is also based on a method that combines a static scattering architecture with a support vector machine. Its filters are based on graph wavelets built from differences between lazy random walks that have propagated at different time scales.

## K.2 Regression of Quantum Chemical Energies

**Dataset:** The dataset we consider is the QM7 dataset, introduced in [2, 31]. This dataset contains descriptions of 7165 organic molecules, each with up to seven heavy atoms, with all non-hydrogen

atoms being considered heavy. A molecule is represented by its Coulomb matrix $C^{\text{Clmb}}$, whose off-diagonal elements

$$C_{ij}^{\text{Clmb}} = \frac{Z_i Z_j}{|R_i - R_j|} \tag{10}$$

correspond to the Coulomb-repulsion between atoms $i$ and $j$, while diagonal elements encode a polynomial fit of atomic energies to nuclear charge [31]:

$$C_{ii}^{\text{Clmb}} = \frac{1}{2} Z_i^{2.4}$$

For each atom in any given molecular graph, the individual Cartesian coordinates $R_i$ and the atomic charge $Z_i$ are also accessible individually. To each molecule an atomization energy - calculated via density functional theory - is associated. The objective is to predict this quantity, the performance metric is mean absolute error. Numerically, atomization energies are negative numbers in the range $-600$ to $-2200$. The associated unit is [*kcal/mol*].

**Scattering Architecture:** Off-diagonal entries in the Coulomb Matrix clearly represent an inverse distance. A weight of zero can then heuristically be thought of as the inverse distance between two infinitely separated atoms. After calculating the degree matrix $D$ associated to $C$, we obtain the graph Laplacian once more as $\mathcal{L} = D - C$ and set our normal operator to

$$\Delta = \frac{\mathcal{L}}{\lambda_{\max}(\mathcal{L})}.$$

If we continuously vary the distances in (10), staying clear of zero, then the adjacency matrix and hence the graph Laplacian $\mathcal{L}$ varies continuously. As long as we avoid complete degeneracy, the largest eigenvalue $\lambda_{\max}(\mathcal{L})$ will remain positive. This implies that our normal operator $\Delta$ varies continuously under changes of the inter-atomic distances, which implies that our feature vector also varies continuously, as distances are changed. Connecting operators are set to the identity, while non-linearities are fixed to $\rho_{n \geqslant 1}(\cdot) = |\cdot|$. Filters are chosen as $(\sin(\pi/2 \cdot \Delta),$ $\cos(\pi/2 \cdot \Delta), \sin(\pi \cdot \Delta), \cos(\pi \cdot \Delta))$ acting through matrix multiplication. The output generating functions are set to the identity as well. Graph level features are aggregated via the map $N_5^E(\cdot)$ of Section 6; slightly modified to neglect the normalizing factor in the first entry for improved convenience in numerical implementability. As weights $\mu_{ij}$ for our second-order feature space are set to unity and molecular graphs in QM7 contain at most 23-molecules, we note that $\sqrt{\mu_{G^2}} \leqslant \sqrt{23^2} = 23$. Going through the proofs of our graph-level stability results, we see that they remain valid after multiplying each stability constant by 23. The Coulomb matrix (divided by a factor of 10 as this empirically improved performance) is then also utilized as an edge level input signal. Node level features are obtained by applying the above architecture to the node level information provided by the respective atomic charges $\{Z_i\}$ on each graph. We aggregate to graph level features using $N_5^G$ (cf. Section 5), again neglecting the normalizing factor in the first entry for improved convenience in implementing. The network depth is set to $N = 4$ in both cases. We then concatenate graph level features obtained from node- and edge level input into a composite scattering feature vector.

**Training Procedure:** The QM7 dataset comes with a precomputed partition into five subsets; each containing a representative amount of heavy and light molecules covering the entire complexity range of QM7. To allow for 10-fold cross validation, we further dissect each of these subsets into two smaller datasets, one containing graphs indexed by an even number, one containing graphs indexed by an odd number. On these 10-subsets, we then perform 10-fold cross validation. Among the 10 folds, we iteratively pick one for testing. Say we have picked the $n^{th}$ fold for testing. Then there are 9 remaining folds. We iteratively pick the $m_n^{th}$ (with $1 \leqslant m_n \leqslant 9$) of the remaining 9 folds for choosing hyperparameters. This leaves 8 folds on which we train our model for each choice of hyper parameter in $C_{\text{pool}} \times G_{\text{pool}}$. This yields 8 regression models, which we average to built our final predictor for the $n^{th}$ run. This mean absolute error of this predictor is then evaluated on the $n^{th}$ fold which was retained for testing. As $n$ varies from one to ten, we built the mean and variance of the performances of the generated regression models. We chose $\log$-linear equidistant hyperparameters from

$$G_{\text{pool}} := \{0.00003, 0.0003, 0.003, 0.03, 0.3, 3, 30\},$$

and
$$C_{\text{pool}} := \{400000, 40000, 4000, 400, 40, 4, 0.4\}.$$

**Reference Methods:**   We comprehensively evaluate our method against 11 popular baselines and state of the art approaches. Among these methods are graph convolutional methods such as GraphConv [18], Weave [17] or SchNet [32]. MPNN [13] and its variant DMPNN [44] are models considering edge features during message passing. AttentiveFP [40] is an extension of the graph attention framework, while N-Gram [21] is a pretrained method. Results for these methods as well as for GROVER are taken from [30]. PhysChem [45] learns molecular representations via fusing physical and chemical information. Deep Tensor Neural Networks (DTNN [39]) are adaptable extensions of the Coulomb Matrix featurizer mapping atom numbers to trainable embeddings which are then updated based on distance information and other (node-level) atomic features. Finally Path-Augmented Graph Transformer Networks (PATGN, [6]) exploit the connectivity structure of the data in a global attention mechanism.