# OpenReview forum: "Graph Scattering beyond Wavelet Shackles"
_NeurIPS.cc/2022/Conference — NeurIPS 2022 Accept_

### Official Review · Reviewer_oeim · 2022-07-12

**Rating:** 7
**Confidence:** 3
**Soundness:** 3 good
**Presentation:** 3 good
**Contribution:** 2 fair

**Summary:**

In this paper, the authors define a very general notion of scattering transform, with an application on graphs. They (re)define each building blocks of the scattering transform: spectral filters, output (low-pass) functions, non-linearities, etc., as well as proper projection operators when the domain changes between layers. Under appropriate assumption of Lipschitzness of each of these elements, they show stability of the resulting transform, as well as energy preservation, generalizing the classical Euclidean results. Some variants on graphs are presented: an aggregation strategy for graph classification and higher-order scattering. Experiments on real data show the effectiveness of the approach.

**Questions:**

-The approach is interesting, but I would suggest either to reformulate the title and/or abstract to be less specialized on graphs, or on the contrary focusing far more the description of the architecture on graphs by giving minimal examples satifying all the formulated hypothesis along the way.

- if the higher-order architecture used in the regression experiment ? It is not clear.

- is the constant "$2$" in definition 4.3 arbitrary ?

- in theorem 4.2 (and related results), could the proximity of the normal operators be expressed in terms of spectral norm rather than Frobenius ? This is more often satisfied (eg by random graphs, or spectral graph coarsening, etc)

**Limitations:**

The authors are quite honest about the limitations in their first experiment, where they are often not state-of-the-art.

**Strengths And Weaknesses:**

Strengths:

- a general approach, than can take into account many variants, domain changes, etc.
- all classical theoretical results on scattering hold
- a very complete supplementary material
- the experiments are convincing, especially for graph regression

Weaknesses:

- a bit paradoxically, the approach suffers from too much generality. The authors define very abstract operators and elements, and in fact nothing in particular is about graphs at all, until the discussions of sections 6 and 7 which are not the core of the approach. Furthermore, the actual choice of the filters, some combination of sin and cos, is quite hidden within the experiment section, and may seem a tad arbitrary. As a result, the reader is somewhat left wondering all along the paper what the actual architecture is, if this is just an abstract formulation of previous architecture or if there is something fundamentally new here. Examples of implementation on graphs along the abstract description could really help the understanding of the approach.
- many variants are described but, it seems, not tested in experiments (changing graphs, higher-order tensors...)
- the theorems are valid under many assumptions, but a minimal examples satisfying all of them is not given

---

> ### Author Response · Authors · 2022-08-02
> **Introduced particular examples along with general results much sooner, clarified which variants are tested, gave (minimal) example satisfying conditions**
>
> We would like to thank the reviewer for the careful review and the poignant comments on our paper. We were especially happy to read that the approach was considered interesting, the experiments (especially for regression), were able to convince the reviewer and that the work that went into presenting the mathematical background as well as detailed explanations in the Appendix/Supplementary material was appreciated by her or him.
>
>
> We followed the advice that was given to us and incorporated the suggested changes in our revised manuscript, as detailed point-by-point below:
>
> 1) "The authors define very abstract operators and elements, […], the actual choice of the filters, […] is quite hidden within the experiment section […]"
>
> We thank the reviewer for this important feedback.
> A discussion of the operators we are utilizing together with a discussion of why we elected to utilize precisely these operators is now already presented in ‘Section 2: Graph Signal Processing’. The topic is picked up again in ‘Section 3: The Generalized Graph Scattering Transform’:
> In this section we now also already describe the filters that we utilize in our numerical experiments and how they harmonize well with our choices of normal operators.
> This section now also describes the scattering transforms corresponding to these filter- and operator choices, so that readers have handy examples at their hands while reading through the theoretical results and have ample time to familiarize themselves with the specific architectures we utilize in computations, before encountering the corresponding numerical results.
>
> 2) "[…]  [the choice of filters]  may seem a tad arbitrary. […]"
>
> Together with our newly written discussion of filter choices in ‘Section 3: The Generalized Graph Scattering Transform’, we have included a discussion of why we chose precisely these filters. In short, two of these filters provide high and low pass filters on the spectrum of the operators we are utilizing, while the other two are spectral refinements of the former two.
>
> 3) "Examples of implementation on graphs along the abstract description could really help the understanding of the approach."
>
> We thank the reviewer for this observation! We now introduce our choices of parameters, functions, operators, etc.  utilized in experiments already in ‘Section 3: The Generalized Graph Scattering Transform’ as particular examples of the general theory. Theoretical results are then illuminated using this example, as we progress through the theoretical sections up to the experimental results.
>
> 4) "many variants are described but, it seems, not tested in experiments (changing graphs, higher-order tensors...)"
>
> We thank the reviewer for this feedback; however we have to slightly and respectfully disagree:
> The higher order architecture is in fact heavily utilized in the regression experiment. To draw attention to this fact, we had already written  ‘In order to showcase the prowess of both our higher order scattering scheme and […] we combine these building blocks into a hybrid architecture’ and subsequently described the higher order scattering architecture, for which, as written in our paper ‘we consider a Coulomb matrix as an edge-level input signal on a given  graph. ‘.
> The generated feature, corresponding to second order (i.e. edge level information/ binary relations between nodes)  were then‘combined with node level scattering features (based on atomic charge) into composite feature vectors; plotted in Figure 4.’
> To make this point even more clear, we now already mention in ‘Section 6: Higher order scattering’ that we test second order scattering experimentally in our regression experiment.
> Beyond that we would be amiss not to mention that the supplementary material submitted with our original manuscript contains a section comparing results of regression on first- and second order scattering vectors (as described in Section 7: Experimental results; Regression of Quantum Chemical Energies) against regression of scattering vectors obtained solely from first order scattering. We obtain the result that the inclusion of second order scattering vectors significantly improves performance.
> Following the received feedback, we have now included the results from solely utilizing first order scattering features in our Table 2.
> We now also discuss the effect of including higher order scattering feature vectors in our main text of ‘Section 7: Experimental Results’
>
> [[continued below]]

---

> > ### Author Response · Authors · 2022-08-02
> > **Adressing further raised points**
> >
> > [[continuation]]
> >
> > As far as changing input graphs are concerned, this is the main focus of our numerical experiments.
> >
> > Beyond that, we did indeed not change graphs as we progress through the layers of our scattering transform in experiments. Changing graphs within an architecture in the sense of graph pooling – while for a time being a persistent approach in the community-- has been shown to be not particularly helpful in many settings [A], while decoupling the input graph from the graph on which message-passing/graph-convolution is performed is a fairly new idea that currently continues to be investigated [1, 37]. While we wanted to state that the scattering setting can indeed accommodate such approaches as well, we thought it best to not widen our focus too much and disentangle the effects of going beyond the wavelet setting from any added benefits of performing convolutions on graphs other than the input graph. We do however plan to investigate this in further works.
> > Considering flexibility in the choices of operators, we would like to point out that while our described theory provides a framework incorporating all previously investigated scattering transforms (based on various choices of graph shift operators), the shift operators we utilize are different from all previously utilized ones; hence showcasing that previous reliance on specific choices on Laplacians (e.g. related to diffusion processes [10] ) are not necessary.
> > Both newly introduced methods of feature aggregation are tested experimentally; one in the classification experiment and one in the regression experiment. We have now made this point clearer by highlighting more which aggregation method is utilized in the respective experiments.
> > We believe that introducing more variation in the experimentally utilized architectures would yield diminishing returns as it pertains to the readability of the paper.
> > However, should the reviewer disagree, we would of course be happy to oblige any further requests!
> >
> >
> > 5) "the theorems are valid under many assumptions, but a minimal examples satisfying all of them is not given
> > The approach is interesting, but I would suggest either to […] [focus ] far more the description of the architecture on graphs by giving minimal examples satifying all the formulated hypothesis along the way."
> >
> > We thank the reviewer for this great idea. In fact, we have chosen the experimental setup for our classification experiment precisely so that it provides a minimal example satisfying the conditions for (almost) all stated theoretical results. We have now made this a lot clearer, explaining after each theorem in ‘Section 4: Stability Results’ how the requirements of the respective theorems are fulfilled by our two numerically tested Architectures.

---

> > > ### Author Response · Authors · 2022-08-02
> > > **Now answering raised questions:**
> > >
> > > Having explained how we implemented the received feedback, we now answer the raised questions:
> > >
> > > 1) if the higher-order architecture used in the regression experiment ?
> > >
> > > As detailed above,  we have now highlighted that second-order scattering is utilized in our regression experiment in our revised manuscript at various points in the paper.
> > > We also rewrote ‘Section 6: Higher order scattering’ to focus solely on second-order scattering, as this is the architecture we analyse experimentally with our regression experiment.
> > >  The discussion of scattering transforms applied to input beyond binary relations is now deferred to the appendix, which -- we hope -- also aids clarity and flow of the paper.
> > >
> > >
> > > 2) Is the constant " 2 " in definition 4.3 arbitrary ?
> > >
> > > Any constant larger than 1 would work as well (see Chapter IV of [30] for a very detailed discussion). We have chosen 2 for simplicity.
> > > Following this question, our revised Manuscript, now contains a comment addressing this point, to eliminate any confusion.
> > >
> > > 3) In theorem 4.2 (and related results), could the proximity of the normal operators be expressed in terms of spectral norm rather than Frobenius?
> > >
> > > Unfortunately, this is impossible without making the constant in the inequality of Theorem 4.2 dependent on the cardinality of the vertex sets of the utilized graphs (for more details see e.g. [B]  or [38]).
> > > If a cardinality-dependent constant is acceptable in  an application, one might use the inequality
> > > $||\cdot||_F  \leq ||\cdot||_s \leq \sqrt{| G|} \cdot ||\cdot||_F$ to facilitate contact between the two norms (with $|G|$ the corresponding vertex set cardinality and $||\cdot||_s$ denoting spectral norm).
> > >
> > > References:
> > >
> > > [A] Rethinking pooling in graph neural networks, Proceedings of the 34th International Conference on Neural Information Processing Systems, December 2020 Article No.: 187Pages 2220–2231
> > >
> > > [1] Uri Alon and Eran Yahav. On the bottleneck of graph neural networks and its practical
> > >  implications. In International Conference on Learning Representations, 2021.
> > >
> > > [37] Jake Topping, Francesco Di Giovanni, Benjamin Paul Chamberlain, Xiaowen Dong, and
> > >  Michael M. Bronstein. Understanding over-squashing and bottlenecks on graphs via curvature,
> > >  2021
> > >
> > > [10] Fernando Gama, Alejandro Ribeiro, and Joan Bruna. Diffusion scattering transforms on graphs.
> > >  In 7th International Conference on Learning Representations, ICLR 2019, New Orleans, LA,
> > >  USA, May 6-9, 2019. OpenReview.net, 2019.
> > >
> > > [30] Olaf. Post. Spectral Analysis on Graph-like Spaces / by Olaf Post. Lecture Notes inMathematics,
> > >  2039. Springer Berlin Heidelberg, Berlin, Heidelberg, 1st ed. 2012. edition, 2012.
> > >
> > > [B]   Operator Lipschitz functions, Aleksandrov, Alexei and Peller, Vladimir,
> > > Russian Mathematical Surveys, Volume 71, Number 4, https://arxiv.org/abs/1611.01593
> > >
> > > [38] Wihler T.P. On the Hölder continuity of matrix functions for normal matrices. Journal of
> > >  inequalities in pure and applied mathematics, 10(4), Dec 2009.

---

> > > > ### Comment · Reviewer_oeim · 2022-08-08
> > > > **Response to rebuttal**
> > > >
> > > > I thank the authors for their very detailed rebuttal.
> > > >
> > > > I am not going to go over each bullet point again, but I am quite satisfied with the changes provided; especially in section 3 and after each theorem, the paper is much clearer that way.
> > > >
> > > > I have no real other concern, I increased my initial score.

---

> > > > > ### Author Response · Authors · 2022-08-08
> > > > > **We are glad our rebuttal was able to convince and thank the reviewer for the increased score.**
> > > > >
> > > > > It was a pleasure implementing suggestions and providing answers and explanations for questions!

---

### Official Review · Reviewer_5wnJ · 2022-07-12

**Rating:** 6
**Confidence:** 4
**Soundness:** 3 good
**Presentation:** 1 poor
**Contribution:** 3 good

**Summary:**

The paper proposes a generalization of graph scattering networks that goes beyond the graph wavelets setting. The authors provide stability guarantees for their generalized scattering transform, as well as layer-wise energy decay bounds. The authors propose a simple feature aggregation method to transform graphs into Euclidean space and briefly discussed taking into account higher-order scattering. The authors conducted experiments of their methods on a graph classification task and a graph regression task.

**Questions:**

Questions on the quality front:

1. The lipschitz-type bounds in theorems 4.1 and 4.2 appear to be just iterative applications of a layer-wise lipschitz type condition. While this is certainly a valid bound, the constants in the upper-bound involves a product of N terms/terms raised to the Nth power (this exponential term also appears in theorem 4.5). I would argue that on a practical level, these exponential terms render the bounds rather impractical, unless matching lower bounds/some sharpness result can be shown. In particular, say N is 10. Then if I perturb say the input by some small constant, the output could have a order C^10 change to it, which can be astronomical.

2. In the expressivity/energy section, the analysis was conducted in the limit of infinite network depth. The idea is quite neat, but it raises several important questions that the authors have to address:

-  the authors make the assumption that one can always choose an eigenvector of strictly positive entries. This of course follows from results in spectral graph theory. However, for the connected graph case, which I argue is probably the most important case, it is my understanding that the only eigenvector that satisfies this will be the eigenvector corresponding to the smallest eigenvalue, in which case the eigenvector has constant entries (they are all the same), and the corresponding eigenvalue is just 0, in which case the notation of defining m_n as the minimum and lambda_n etc becomes somewhat redundant. On the other hand, if the graph is disconnected, then the eigenvector that you pick will, to the best of my knowledge, have positive entries in one component and 0's in some other component, which would violate the strictly positive entry assumption. in either case, I think some change/clarification has to be made here.

- While I think the energy bounds are interesting, it is unclear to me how useful/related this is to link to the expressivity of the network. The fact that the mapping only maps 0 to 0, when N goes to infinity, seems like a property that is only marginally related to expressivity in some bare-minimum way. One can probably come up with some kind of invertible linear-type transformation that also only maps 0 to 0. This property to me at first sight seems to just mean that the mapping is not contracting, but going from "no contraction" to "expressivity" seems a bit of a stretch claim to me. I think the authors could modify the wording of their conclusion here.

In terms of the clarity of exposition, I think the general ML audience will find this paper difficult to read and apply, for the following reasons:

1. There is too much material for a 9 page conference paper. Important aspects of the paper are delegated to the appendix, and there is not enough room for the authors to give the necessary treatment for background knowledge and definitions. As a result, only those that already have very substantial backgrounds in graph wavelets/graph networks and spectral graph theory will be able to understand it.

2. The authors have the style of defining things in the broadest, most abstract and general version first, and then in the experimental section just make some very specific choices in their model that conform to their general theoretical results, but without justifying those experimental choices at all. I understand that this is a theoretical paper, but I think having one or two tables of pseudocodes on particular instantiations of your architecture, and providing more justification for why certain parameter or modeling choices are made (such as cross-validation etc) would help the users understand and adopt their method much more readily. In particular, the experimental results for the regression application is great, but for the classification is not very good. I wonder if the performance on the classification task could be improved with an alternative instantiation of their model, such as using other functions than sines and cosines, or changing the layer parameter, or using a different operator than the Laplacian etc.



**Limitations:**

Main limitations are outlined in section above.
No negative societal impact.


**Strengths And Weaknesses:**

- Originality: the idea, methods etc are original. The novelty/originality of the paper definitely meets the bar for publication at NeurIPS.
- Quality: the background and tools used are somewhat technical and motivated quite abstractly. The quality of the mathematics/derivations, even though I didn't check proofs line by line, are, to the best of my knowledge, sound.
- Clarity: the organization and exposition of the article leaves much room for improvement. Details are given below.
- Significance: Nowadays it is certainly interesting and significant to consider problems involving GCNs and graph networks. The topic of this paper is of sufficient significance to meet the bar for publication at NeurIPS.

Overall, I think the originality, significance and mathematical soundness of the paper are fine, whereas the clarity of the paper leaves room for improvement, which I will detail in sections below.

---

> ### Author Response · Authors · 2022-08-02
> **Followed the advice on Clarity and Presentation dilligently**
>
> We immensely thank the reviewer for the careful read of our paper. We are especially happy that in her or his opinion the novelty/originality of the paper definetely meets the bar for publication at NeurIPS. We were also delighted to read that in her or his opinion   the topic of this paper is of sufficient significance for NeurIPS.
>
> We are also very grateful, for the detailed comments and advice we received, which we have followed, as we detail below point for point:
>
>
>
> 1) "The authors have the style of defining things in the broadest, most abstract and general version first,"
>
> We thank the reviewer for this critical observation. Following this comment, we now present a discussion of the operators we are utilizing in numerical experiments  together with a discussion of why we elected to utilize precisely these operators  already in ‘Section 2: Graph Signal Processing’.
> The topic is picked up again in ‘Section 3: The Generalized Graph Scattering Transform’:
> In this section we now also already describe the filters that we utilize in our numerical experiments and how they harmonize well with our choices of normal operators.
> This section now also describes the scattering transforms corresponding to these filter- and operator choices, so that readers have handy examples at their hands while reading through the theoretical results and have ample time to familiarize themselves with the specific architectures we utilize in computations, before encountering the corresponding numerical results.
> Additionally in ‘Section 4: Stability Results’, we pick these Example-Architectures up again and  explain after each Theorem how the requirements of the Theorem are fulfilled by our two numerically tested Architectures.
>
>
> 2) "and then in the experimental section just make some very specific choices in their model that conform to their general theoretical results, but without justifying those experimental choices at all. "
>
> We have now explained our parameter choices much sooner and a lot clearer:
> ‘Section 2: Graph Signal Processing’ now includes a discussion of why we pick the normal operators that we utilize for our numerical investigations. In short, the reason is that the spectrum of these operators is contained in the interval [0,1], with the values 0 and 1 being attained. This control over the spectrum aids greatly in selecting filters.
> In particular, as ‘Section 3: The Generalized Graph Scattering Transform’ now explains in more detail, our experimentally utilized filters essentially provide high- and-low pass filters on this spectrum contained in [0,1]. Thus input-signals are dissected corresponding to their high and low frequency components.
> Connecting operators and non-linearities are chosen in the standard way to facilitate comparability with existing wavelet-scattering architectures.
> Depth of our generalized scattering transform as well as the parameter ‘p’ for the non-linear graph level aggregation method are chosen to avoid overfitting and keep computation-time palatable.
> Split size for cross validation was chosen to follow the standards on the respective datasets.
>
>
> 3) "I understand that this is a theoretical paper, but I think having one or two tables of pseudocodes on particular instantiations of your architecture, and providing more justification for why certain parameter or modeling choices are made (such as cross-validation etc) would help the users understand and adopt their method much more readily. "
>
> As discussed in our point above, we completely agree with this comment and have now diligently provided justifications and heuristics for our parameter choices. We have opted for a graphical representation of our specific Architectures (c.f. Fig. 2 ) utilized in experiments, and have included this graphical representation together with a description of these architectures already in ‘Section 3: Generalized Scattering transforms’.

---

> > ### Author Response · Authors · 2022-08-02
> > **Continuation of Comment I**
> >
> > 4) "In particular, the experimental results for the regression application is great, but for the classification is not very good. I wonder if the performance on the classification task could be improved with an alternative instantiation of their model, such as using other functions than sines and cosines, or changing the layer parameter, or using a different operator than the Laplacian etc."
> >
> > We have run the classification experiments with different choices of filters (polynomials of varying degrees, exponentials), different network depths, different aggregation methods (low pass vs. general non-linear), different branching ratios (2 vs. the presented 4) and different non-linearities; albeit not on all datasets but only on the IMDB datasets. The architecture presented in the paper provided the best results. Additionally, [12]  ran this very classification experiment with geometric wavelets (based on a different operator and different functional calculus filters), as can be read from the row entitled GS-SVM in Table 1 of our paper. Results of [12]  were not better than ours. We believe that the presented architecture is at the apex of scattering transforms applied to classification objectives.
> > We believe that the much better performance of our scattering architecture (when compared to other leading approaches) on the regression task can be explained by the fact that inputs in the classification setting can heuristically be considered as discrete (a node has a property or it does not; an edge exists or it does not) in the classification setting, while inputs can heuristically be considered continuous in the regression setting (e.g. interatomic distances can be varied continuously). Scattering is particularly adapted to this continuous setting as the results of ‘Section 4: Stability Results’ and Figure 5 illustrate.
> >
> >
> >
> >
> > 5a) "There is too much material for a 9 page conference paper."
> >
> > While we do agree that our paper does contain many new, and – we believe – interesting as well as widely applicable results, we would argue that we do not present too much material: It is our firm believe that a conference paper should be self-contained and should provide the reader with deep insight into the topic of the work as well as detailed explanations of any novel material.
> > We firmly believe that presenting experiments that display superior numerical performance of non-wavelet filters together with a general theory that also allows the application of scattering transforms to new domains (inaccessible to standard wavelet based scattering transforms) within graph signal processing
> > is the best way to convince the community to transcend the graph-wavelet setting and embrace our newly developed general scattering transforms.
> > We have made sure to only include the minimal necessary information to develop this topic, scrapping many additional results we would have ideally liked to present.  We do however welcome further advice by the reviewers on how to streamline our paper even more, should they deem it necessary. Should we be given information on which parts seem irrelevant, we would be more than happy to follow such recommendations to scrap!
> >
> >
> >
> >
> >
> >
> > 5b) "Important aspects of the paper are delegated to the appendix, and there is not enough room for the authors to give the necessary treatment for background knowledge and definitions. "
> >
> > We are sorry to hear that the reviewer feels this way.
> > To ease reading and facilitate the uptake of our ideas by the community, we took great care to organize the material in a manner most accessible to the reader. In fact, our goal is to keep the main aspects and novel conceptual ideas within the main body of the paper and outsource, for instance, the proofs which are not key for understanding the novel ideas to the appendix.
> > We agree that we wrote a very comprehensive appendix, but this was mainly done for the sake of completeness, in the interest of being self-contained and to aid readers that are not completely familiar with the subject matter.
> > We would like to stress that it is by no means necessary to work through the entire appendix to appreciate the main points of the paper and do sincerely hope that we did not give this impression to the reviewer.
> > Should the reviewer still feel that important aspects are missing from the main body of the work, we would be curious to know to which aspects she or he is referring to. We would be happy to do our best to transport them from the appendix to the main body of the paper.

---

> > > ### Author Response · Authors · 2022-08-02
> > > **Continuation of Comment II**
> > >
> > > 6) "As a result, only those that already have very substantial backgrounds in graph wavelets/graph networks and spectral graph theory will be able to understand it. "
> > >
> > > We would like to respectfully state that all necessary definitions and notions are contained in the main body of the paper alongside a description of the necessary intuition of the field and our novel conceptual ideas and results. In fact, we start our discussion by reviewing the very basics of the signal processing framework and only afterwards build up steam and introduce our new scattering transforms, while always making sure to properly introduce new or maybe not too well-known concepts. For readers unfamiliar with the field or even parts of it we have written a comprehensive appendix reviewing concepts from fundamental topics in linear algebra to tensorial inputs and functional calculus. Along the way, we provide ample resources and references containing even more detailed explanations of concepts and topics touched upon.
> > > It would help us immensely if the reviewer could make clear what she/he feels is missing in our introductions or where precisely we could extend our writing or make it more precise.
> > >
> > >
> > >
> > > 7) "The lipschitz-type bounds in theorems 4.1 and 4.2 appear to be just iterative applications of a layer-wise lipschitz type condition."
> > >
> > > While Lipschitz continuity plays an important part in deriving the stability bounds, equally important parts are played by applications of Cauchy-Schwartz and Cauchy-Young inequalities and maybe most importantly the generalized frame condition in various guises as well as careful applications of combinations of these inequalities. The importance of  the generalized frame property above all else is elucidated further in our next comment below.
> > >
> > > 8) "While this is certainly a valid bound, the constants in the upper-bound involves a product of N terms/terms raised to the Nth power (this exponential term also appears in theorem 4.5). I would argue that on a practical level, these exponential terms render the bounds rather impractical, unless matching lower bounds/some sharpness result can be shown. In particular, say N is 10. Then if I perturb say the input by some small constant, the output could have a order C^10 change to it, which can be astronomical. "
> > >
> > > We empathise with the reviewer's initial reaction to the dependence of the bounds on layer depth; we had the same initial reaction when we first derived these results. However the situation is much much better than it would initially seem.
> > > A first mitigating factor is that in real world applications, scattering networks are rarely deeper than N = 5, which already controls any exponential behaviour well.
> > >
> > > In greater specificity, let us now address the stability constants of the two theorems individually:
> > >
> > > For Theorem 4.1 we may note that in front of the product of n terms, there is a factor that is zero if $B_n \leq 1$ and the product $B_n(L_n^+R_n^+)^2 \leq 1$.
> > > If this demand is met, the product of n terms disappears and no longer contributes to the stability constant. What is more, since filters, connecting operators and non-linearities are static parts of the architecture, one can always meet the demand $B_n \leq 1$ and $B_n(L_n^+R_n^+)^2 \leq 1$ by a simple rescaling operation.
> > > If the demand is met in each layer of the scattering architecture, the resulting scattering transform is 1-Lipschitz irrespective of depth.
> > > We have emphasized this in our discussion immediately after Theorem 4.1 (c.f. also equation (2)).
> > >
> > > As for Theorem 4.2 as it stood in our original submission, we note that even upon choosing N= 10, as the reviewer suggested, the contribution to the stability bound by the exponential term will only be $ \sqrt{2(2^{10} - 1)} \approx 45$, which is far from being astronomical.
> > > However there is even more that can be said. In our original submission, we had fixed the upper frame bound to be equal to one for simplicity in presentation. Following the reviewers concerns about exponential increase of stability constants with the depth N, we have now kept the upper frame constant B variable. The reason for this is that there is a sort of phase transition going on: For $B\leq\frac12$ in each layer, we can prove that the exponential increase with the depth does not persist, and the stability constant can be chosen as $2\cdot D$ independently of network depth if $B\leq\frac12$. Again, this can always be achieved through a rescaling operation. Here D accounts for the Lipschitz constants of the individual filters. More details are provided in our updated Theorem 4.2.
> > > As this behaviour is a consequence of the generalized frame condition, this also exemplifies that the derived bounds are not merely a consequence of a repeated Lipschitz condition.
> > >
> > > The discussion for Theorem 4.5 proceeds analogously to the one of Theorem 4.2.

---

> > > > ### Author Response · Authors · 2022-08-02
> > > > **Adressing raised Questions and further points:**
> > > >
> > > > 9) "In the expressivity/energy section, the analysis was conducted in the limit of infinite network depth. The idea is quite neat, but it raises several important questions that the authors have to address:"
> > > >
> > > > •	"the authors make the assumption that one can always choose an eigenvector of strictly positive entries. This of course follows from results in spectral graph theory. However, for the connected graph case, which I argue is probably the most important case, it is my understanding that the only eigenvector that satisfies this will be the eigenvector corresponding to the smallest eigenvalue, in which case the eigenvector has constant entries (they are all the same), and the corresponding eigenvalue is just 0, in which case the notation of defining m_n as the minimum and lambda_n etc becomes somewhat redundant. "
> > > >
> > > > On a connected graph, it is indeed correct that for the graph Laplacian $ L = D – W$  the lowest lying eigenvalue is zero and that the only eigenvector that can be chosen to have purely positive entries is the eigenvector corresponding to the eigenvalue zero.
> > > > In that case, this eigenvector has indeed constant entries.
> > > > However, one might also consider the normalized graph Laplacian $ Id – D^{-\frac12}W D^{-\frac12}$ as e.g. [10] does in a wavelet-scattering setting. In this case, the entries of the lowest lying eigenvector (of the normalized graph Laplacian) are given by the square-roots of the degrees of the corresponding nodes. Thus we believe it has merit to keep $m_n$ as the minimum entry of such a vector as a variable in the formulation of our theorem.
> > > > We introduced $\lambda_n$ as the corresponding eigenvalue and did not fix it to equal zero, to emphasize that nothing in particular is dependent on the eigenvalue under consideration being zero. We might equally well base our architecture on $ 3Id – D^{-\frac12}W D^{-\frac12}$ instead of $ Id – D^{-\frac12}W D^{-\frac12}$, in which case this lowest lying eigenvalue would be equal to $2$.
> > > >
> > > > •	"On the other hand, if the graph is disconnected, then the eigenvector that you pick will, to the best of my knowledge, have positive entries in one component and 0's in some other component, which would violate the strictly positive entry assumption. "
> > > >
> > > > Let us assume that the graph has K disconnected components. Let us further make use of the un-normalized graph Laplacian $ L = D – W$.
> > > > Then the lowest lying eigenspace (corresponding to eigenvalue 0) is $K$-dimensional. An orthogonal basis of this space is given by the $K$ vectors whose entries are equal to one on a specific connected component and zero on all others.
> > > > Any linear combination with positive coefficients of these vectors will yield a vector with only positive entries.
> > > > As these K vectors form a basis of the lowest lying eigenspace, any linear combination of these vectors will still lie in this eigenspace and hence will be an eigenvector to the eigenvalue zero.
> > > > Thus a vector as is desired by our theorem also exists for the disconnected case. In fact, we can simply chose it to be the normalized constant vector with positive entries again.
> > > > Following this question by the reviewer, we have included a comment emphasizing this in our revised manuscript.
> > > >
> > > >
> > > >
> > > > •	"While I think the energy bounds are interesting, it is unclear to me how useful/related this is to link to the expressivity of the network. The fact that the mapping only maps 0 to 0, when N goes to infinity, seems like a property that is only marginally related to expressivity in some bare-minimum way. One can probably come up with some kind of invertible linear-type transformation that also only maps 0 to 0. This property to me at first sight seems to just mean that the mapping is not contracting, but going from "no contraction" to "expressivity" seems a bit of a stretch claim to me. I think the authors could modify the wording of their conclusion here."
> > > >
> > > > We do agree that it is a stretch to go from a trivial ‘kernel’ to talking about expressivity. The term ‘Expressivity’ and the surrounding discussion stems from analysis of Euclidean Scattering Networks [A]. We derived the corresponding results for the graph setting in our paper.
> > > > To decrease the emphasis on expressivity, we struck this word from Abstract and Introduction and demoted the discussion of the trivial ‘Kernel’ property to a side-note in the appendix. In our revised manuscript, we now focus much more on the relation between Energy decay and truncation stability. As this discussion pertains to stability, we have now incorporated it into ‘Section 4: Stability Results’.

---

> > > > > ### Author Response · Authors · 2022-08-02
> > > > > **References:**
> > > > >
> > > > > References:
> > > > >
> > > > > [12] Feng Gao, Guy Wolf, and Matthew J. Hirn. Geometric scattering for graph data analysis. In
> > > > >  Kamalika Chaudhuri and Ruslan Salakhutdinov, editors, Proceedings of the 36th International
> > > > >  Conference on Machine Learning, ICML 2019, 9-15 June 2019, Long Beach, California, USA,
> > > > >  volume 97 of Proceedings of Machine Learning Research, pages 2122–2131. PMLR, 2019.
> > > > >
> > > > > [10] Fernando Gama, Alejandro Ribeiro, and Joan Bruna. Diffusion scattering transforms on graphs.
> > > > >  In 7th International Conference on Learning Representations, ICLR 2019, New Orleans, LA,
> > > > >  USA, May 6-9, 2019. OpenReview.net, 2019.
> > > > >
> > > > > [A] Wiatowski, Thomas, Harmonic Analysis of Deep Convolutional Neural Networks, 2018, Doctoral Thesis, ETH Research Collection

---

### Official Review · Reviewer_697M · 2022-07-14

**Rating:** 7
**Confidence:** 2
**Soundness:** 3 good
**Presentation:** 3 good
**Contribution:** 3 good

**Summary:**

In this paper, the authors focus on the design and analysis of graph scattering networks with variable branching ratios and generic functional calculus filters. Spectrally-agnostic stability guarantees for node- and graph-level perturbations are established.



**Questions:**

1. The authors should add comments on the optimality of the upper bounds established in Section 4.

2. More background and details are needed for the section on higher order scattering.

**Limitations:**

Yes.

**Strengths And Weaknesses:**

Strengths:
This paper is well-organized. The theoretical results on stability guarantees and the experimental results on quantum chemical energies are interesting.

Weaknesses:
1. The authors present several upper bounds in Section 4 for stability guarantees. However, it is unknown the optimality of these bounds.

2. The section on higher order scattering needs more background knowledge to follow for readers.

---

> ### Author Response · Authors · 2022-08-02
> **Followed Advice: Added comment on optimality and added much more details on higher order scattering**
>
> We thank the reviewer for her or his careful evaluation, appreciation of the paper and kind comments. We were very happy to read that the paper was considered to be well organized and that theoretical- as well as experimental results were thought to be interesting.
>
> Let us address the raised questions and given advice individually:
>
> 1) "The authors present several upper bounds in Section 4 for stability guarantees. However, it is unknown the optimality of these bounds."
>
> The reviewer raised the question of the optimality of bounds in Section 4:
> To obtain these bounds, we have developed a proof-framework that combines the triangle inequality with the Cauchy Young inequality and the generalized frame condition in various disguises.
> This allowed us to achieve significantly better and more general bounds than previous works focusing on graph scattering ( e.g. [11]) and recover bounds of the Euclidean setting (see e.g. [40]).
> Given the desired generality (in terms of utilized filters, connecting operators, non-linearity, graph-sizes,…) in the statement of the respective inequalities; we are unaware of approaches that lead to even better bounds. However, since we cannot rule out their existence, we have added a comment, stating that the derived bounds are not necessarily optimal at the beginning of Section 4.
>
> 2) "More background and details are needed for the section on higher order scattering."
>
> Following this feedback, we have significantly expanded the section titled ‘Details on Higher Order Scattering’; i.e. Appendix  J. It now includes a recap of the notion of higher order (tensorial) input, a precise and very detailed formulation of higher order scattering transforms and a discussion of the corresponding feature aggregation map.
> We have also reduced the scope  of ‘Section 6: Higher Order Scattering’ which now focuses solely on the familiar case of edge-inputs. We made this choice since edge inputs (i.e. binary relations or equivalently 2-tensors) constitute the higher-order input that is actually utilized in our regression experiment in ‘Section 7: Experimental Results’.
> We also believe that focusing on the familiar Edge level setting first and deferring a full discussion of higher order scattering to the appendix, helps build intuition first and prevents the reader from being overwhelmed from the theory of higher order inputs in full generality before developing an appreciation of the topic.
>
>
> References:
>
> [11] Fernando Gama, Alejandro Ribeiro, and Joan Bruna. Stability of graph scattering transforms. In
>  Hanna M. Wallach, Hugo Larochelle, Alina Beygelzimer, Florence d’Alché-Buc, Emily B. Fox,
>  and Roman Garnett, editors, Advances in Neural Information Processing Systems 32: Annual
>  Conference on Neural Information Processing Systems 2019, NeurIPS 2019, December 8-14,
>  2019, Vancouver, BC, Canada, pages 8036–8046, 2019.
>
> [40] Thomas Wiatowski and Helmut Bölcskei. A mathematical theory of deep convolutional neural
>  networks for feature extraction. IEEE Transactions on Information Theory, 64:1845–1866,
>  2018.

---

### Meta-Review · Area_Chair_nfDE · 2022-08-23

**Recommendation:** Accept
**Confidence:** Certain

**Metareview:**

In the discussion, we reached a clear consensus that this paper is interesting for the NeurIPS community and should be accepted. The author's rebuttal and subsequent discussion were very useful and we are looking forward to the final version of the paper with the promised improvements implemented.

**Award:**

No

---

### Decision · Program_Chairs · 2022-09-14

Accept